# The Role of Determinism in the Prediction of Corrosion Damage

Digby D. Macdonald 

Departments of Nuclear Engineering and Materials Science and Engineering, University of California at Berkeley, Berkeley, CA 94720, USA; macdonald@berkeley.edu

**Abstract:** This paper explores the roles of empiricism and determinism in science and concludes that the intellectual exercise that we call "science" is best described as the transition from empiricism (i.e., observation) to determinism, which is the philosophy that the future can be predicted from the past based on the natural laws that are condensations of all previous scientific knowledge. This transition (i.e., "science") is accomplished by formulating theories to explain the observations and models that are based on those theories to predict new phenomena. Thus, models are the computational arms of theories, and all models must possess a theoretical basis, but not all theories need to predict. The structure of a deterministic model is reviewed, and it is emphasized that all models must contain an input, a model engine, and an output, together with a feedback loop that permits the continual updating of the model parameters and a means of assessing predictions against new observations. This latter feature facilitates the application of the "scientific method" of cyclical prediction/assessment that continues until the model can no longer account for new observations. At that point, the model (and possibly the theory, too) has been "falsified" and must be discarded and a new theory/model constructed. In this regard, it is important to stress that no amount of successful prediction can prove a theory/model to be "correct", because theories and models are merely the figments of our imagination as developed through imperfect senses and imperfect intellect and, hence, are invariably wrong at some level of detail. Contrariwise, a single failure of a model to predict an observation invalidates ("falsifies") the theory/model. The impediment to model building is complexity and its impact on model building is discussed. Thus, we employ instruments such as microscopes and telescopes to extend our senses to examining smaller and larger objects, respectively, just as we now employ computers to extend our intellects as reflected in our computational prowess. The process of model building is illustrated with reference to the deterministic Coupled Environment Fracture Model (CEFM) that has proven to be highly successful in predicting crack growth rate in metals and alloys in contact with high-temperature aqueous environments of the type that exist in water-cooled nuclear power reactor primary coolant circuits.

**Keywords:** theories; models; determinism; empiricism; prediction; science; CEFM; BWR; stress corrosion cracking





## 1. Introduction

Complex industrial systems are unique, even when they are of the same design, often because of unique operating conditions and histories. Failures are rare events [1], and hence it is generally impossible to develop an effective empirical database covering the range of independent variables that characterize complex, industrial systems, which is required for the accurate prediction of damage. Furthermore, empirical models are generally expensive because of the need for large, labor-intensive and, hence, expensive calibrating databases. Empirical models also fail to capture the mechanism of failure, and they generally fail to yield the accuracy of prediction to make them useful for maintenance and life extension analyses. Of great importance is that empirical models have limited prediction factors, PF (defined as the time of prediction/calibrating data record). Commonly, for an empirical model, 1 < PF < 5, whereas for deterministic models 1 < PF < 1000 or more. Indeed, this

feature of determinism is exploited in modeling the fate of metallic canisters for the disposal of high-level nuclear waste (HLNW), where a PF of >100,000 is required to ensure that the waste can be isolated from the biosphere for sufficient time for the fission product nuclides to have decayed to harmless levels.

Why is accurate prediction important? Corrosion damage is responsible for huge economic losses in industrialized societies (3–4.5% of GDP per year [2] or about USD 630 billion to USD 945 billion in the US in 2020, based on a GDP of USD 21 trillion). The worldwide cost is three to four times greater, making corrosion one of the costliest of all-natural phenomena. For comparison, the annual USA costs of hurricanes and earthquakes are USD 28 billion [3] and USD 6.1 billion [4], respectively. Approximately 30% of the cost of corrosion could be avoided by the better application of existing corrosion control technology, if only we knew in advance where and when corrosion damage will occur. Thus, if we knew, then systems might be serviced during scheduled outages, thereby avoiding costly, unscheduled downtime. For example, the cost of downtime for a 980 MWe nuclear power plant is USD 1–2 million/day [5], depending upon the cost of purchasing power from the grid to replace that lost by an unscheduled outage. The total cost is determined by the length of the outage. A failed low-pressure steam turbine (LPST) or a steam generator (SG) can keep a plant offline for a year or more, resulting in a total loss of USD 360 million to USD 720 million per event. These enormous costs are eventually paid for by the consumers of the electricity and/or the taxpayers. Accordingly, a considerable incentive exists in proactively managing the development of damage, but this requires the use of models that can accurately predict the progression of corrosion damage. This paper describes one such model, the Coupled Environment Fracture Model (CEFM) that was developed by the author and his colleagues to model the progression of stress corrosion cracking damage in the heat-transport circuits of water-cooled nuclear power reactors [1]. The CEFM is used here to illustrate the development of a deterministic model.

The most insidious forms of corrosion are localized corrosion processes, such as pitting, stress corrosion cracking, corrosion fatigue, hydrogen embrittlement, and crevice corrosion, because they often produce failures with little outward sign of accumulated damage. To date, predictions have been made largely based on empirical statistical models, e.g., extreme value statistics (EVS), which generally have failed to produce the required accuracy of prediction, although they often bring satisfaction in the ordering of observations. The principal problem is that the EVS distribution in damage depth within a large population is determined by two parameters in the Gumbel Type 1 distribution function: the shape parameter and the location parameter [1]. In the empirical form of EVS both must be determined by calibration, which requires a large database of crack depth (for example) at various times in the past, and because in the original EVS model no deterministic method was available for predicting the time dependencies of the two parameters. Thus, in essence, the predicted result had to be known in advance of the prediction being made. This limitation has been largely addressed by the development of Deterministic Extreme Value Statistics (DEVS) and Deterministic Monte Carlo Simulation (DMCS) [1], but to the author's knowledge these models have only been applied to the SCC failure of LPST blades and natural gas pipelines [1].

## 2. Philosophical Basis of Determinism vs. Empiricism

Much has been written about the philosophical basis of science, extending all the way back 2870 years when Aristotle (384–322 BC) published his treatise, Physics [6], in 350 BC. Although he is often credited with defining the concept of causality, upon which modern scientific philosophy is based, in the opinion of the author this attribution is perhaps a little overstated. Although he did discuss at some length the nature of "cause" and the resulting "effect", he did not do so in terms of quantifiable concepts, such as "force" or "displacement", respectively. Nevertheless, Aristotle, for his time, displayed great insight into the philosophical basis of the natural world, as is displayed by his statement: "It is plain then that nature is a cause, a cause that operates for a purpose". Indeed, in his

writings it is possible to detect the foundation of Newton's Laws of Motion, which are generally regarded as being the foundation of modern physics but which were formulated about 2200 years later. Undoubtedly, Newton was conversant with the writings of Aristotle, as were most natural philosophers at the time. However, a comprehensive discussion of the philosophical basis of science is well beyond the scope of this paper and the reader will find many outstanding treatises on the subject identified on the web. Below, the views on "science" are strictly those of the author, and no pretense is made that the views represent those of mainstream scientists or scientific philosophers.

Extensive inquiry by the author on the nature of science and the role of determinism in the scientific process has led him to conclude that the fabric of science is based upon the natural laws, which are generalizations and condensations of all scientific knowledge. The five natural laws of relevance in this discussion are Proust's Law of Definite Proportions, Lavoisier's Law of the Conservation of Mass, the Law of Multiple Proportions (Dalton), the Law of the Conservation of Charge, and Faraday's Law of Mass-Charge Equivalency. The first three are the fundamental laws of chemistry, while the latter two are fundamental laws of physics and electrochemistry, respectively. A particularly important feature of the natural laws is that they are time- and space-invariant. Accordingly, a useful definition of "science" is that it is a process resulting in the transition from empiricism (observation) to determinism (deduction or prediction) via the formulation of the natural laws, which are condensations of all scientific knowledge, as noted above. Note that a theory and the resulting model may be based upon one or more (often multiple) natural laws, and it is important to recognize that all laws must be compatible in that any given law cannot violate other laws that may be only peripherally related to the subject at hand. If such a conflict exists, one or both are not "natural laws", and the conflict must be resolved before proceeding further. The transition and, hence, "science", involves the development of theories and models that are based upon compatible natural laws, with the "scientific method" being used to nudge the models toward reality, recognizing that "reality" is a figment of the observer's imagination. These concepts are discussed in greater detail below.

In this paper, I will outline my views on the subject by discussing and addressing important issues, including:

- The definition of determinism versus empiricism;
- Why determinism is so important;
- The structures of deterministic vs. empirical models;
- The concept of the corrosion evolutionary path (CEP).

I will illustrate these concepts with respect to the development of the Coupled Environment Fracture Model (CEFM), which, as noted above, has been remarkably successful in predicting the stress corrosion crack growth rate in a variety of metals and alloys in industrial systems, including the coolant circuits of water-cooled, nuclear power reactors [7–21].

Finally, it is important to note that pure "determinism" in any model is strictly an idealized, unachievable state because all models contain data and concepts that are empirically based, and the CEFM is no exception. As we will see below, the CEFM must be calibrated by two crack growth rate (CGR) data at different temperatures for specified values of the other independent values. Nevertheless, with that minimal calibration, the CEFM accurately predicts the response of the dependent variable (CGR) on the various independent variables with accuracies within experimental observation and, indeed, has predicted the previously unreported dependence of the CGR on the electrochemical crack length (ECL).

### 3. The Structure of a Deterministic Model

The development of models in human intellectual pursuits is a very complex subject that extends well beyond this paper, but an excellent review is given by Frig and Hartmann [22]. Only those aspects of the subject that impact the topic of this paper will be discussed here. All deterministic models have a common structure, either explicitly or implicitly, as displayed in Figure 1. Thus, all deterministic models must have a theoretical

basis that is, itself, based upon observation [23]. These observations may be presented as postulates or assumptions, with postulates being based directly on observation (e.g., the sun emits heat) while assumptions are statements that are accepted initially without necessarily being supported by direct observation/knowledge (e.g., the sun's heat must come from combustion, a clearly incorrect statement, but which might account for the observation in the absence of more detailed evidence). Clearly, a theory/model based upon this postulate and assumption might explain some of the sun's impact on its surroundings, but would fail once inquiry was made concerning the fuel and oxidant. Thus, it is important to note that a theory can be no more valid than the postulates and assumptions upon which it is based. It is also important that the postulates should not foretell the desired result. Thus, most climate change models developed under the auspices of the Intergovernmental Panel on Climate Change (IPCC) employ the Anthropogenic Global Warming (AGW) hypothesis that presupposes that "global warming" is due to humans releasing carbon dioxide into the atmosphere via the burning of fossil fuels. It has been recently shown by the author that the AGWH violates the Causality Principle in that ice-core records show that the change in temperature precedes the change in $[CO_2]$. Accordingly, the AGWH lacks a valid scientific basis [24]. In adopting the AGW hypothesis, the global warming models invariably predict that human activity is responsible for global warming, but they are fatally flawed, and their predictions should not be relied upon. The fault lies in adopting a postulate that foretells the desired result and that also violates causality.

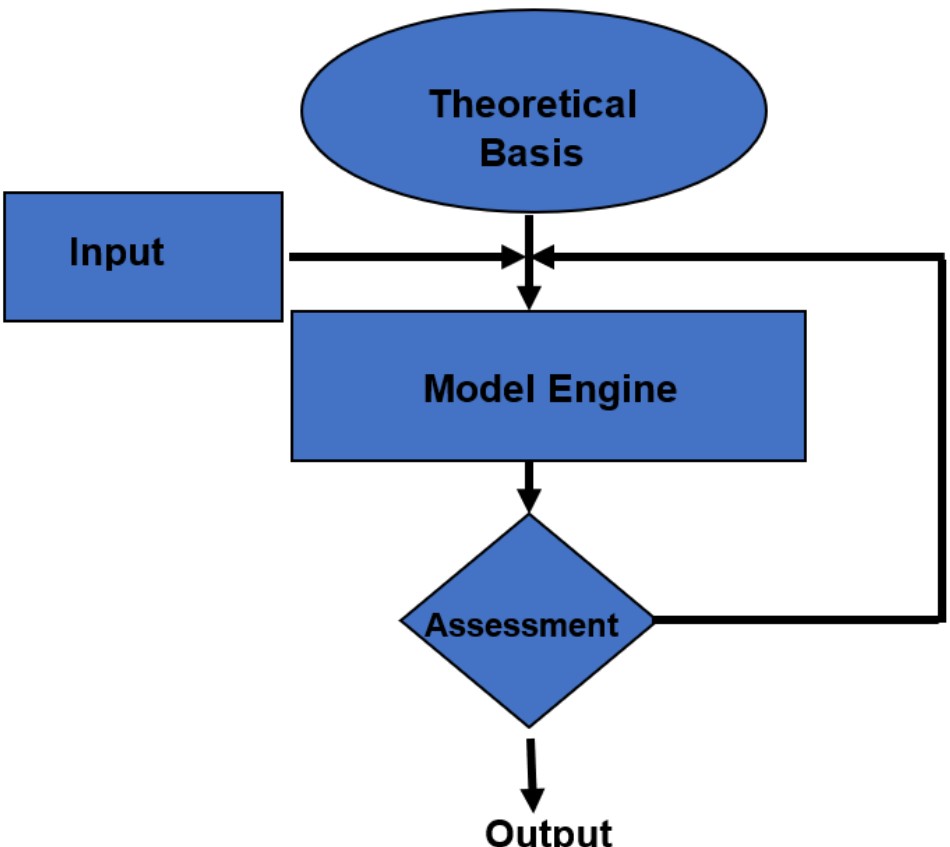

**Figure 1.** Structure of a deterministic model.

Thus, in essence, the important characteristics of a valid theory and the resulting deterministic model may be summarized as follows:

- A theory must be based upon experimental observation [23].
- The model based on that theory contains N "constitutive" equations that describe the relationships between various components and M "constraints", which are statements

of the natural laws (typically the conservation conditions) and which constrain the output to that which is "physically real".

- M + N must be at least equal to the number of unknowns in the model. If it is not, the system is said to be mathematically underdetermined and deterministic prediction is not possible. If N + M is greater than the number of unknowns, the model is said to be mathematically "overdetermined" and deterministic prediction is unimpeded.
- All equations must be mathematically independent.
- *Ad hoc* relations cannot be added simply to "make the model work" (Einstein's famous admonishment to the scientific community!) [23].

The other essential component of a deterministic model is the assessment loop that is used to determine how accurately the model predicts new phenomena (i.e., phenomena that were not included in the postulates and assumptions upon which the model is based). This loop is shown in Figure 1 as the feedback loop from the assessment module (the diamond) to the input and is, in essence, the "scientific method", in which the parameters of the model engine are modified to nudge the prediction closer to reality.

"Reality" is defined by the user, but it generally requires the prediction to fall within an uncertainty band that is consistent with empirical observation. This process (the "scientific method") is continued until no further valid adjustments of the model improve the prediction or until the prediction fails to agree with observation, resulting in the rejection of the theory and model. In this regard, it is important to stress again that no amount of agreement between model prediction and observation can "prove" that a model is correct, but only one disagreement is required to disprove a model and the theory upon which the model is based. Regrettably, models that have been proven to be invalid often are continued to be used merely because they have some convenient feature, such as an equation that is easily manipulated. Such an example is the high field model for metal oxidation [25], which has been invalidated on several levels and yet still enjoys widespread use. The continued use does not advance the science of metal oxidation but simply diverts intellectual resources from the development of valid alternatives.

On the other hand, the path of scientific progress is littered with the corpses of well-formulated models that were discarded because they, seemingly, made predictions that did not agree with observation. The disagreement can often be traced to the fact that the model and the observed system are not in "confluence", in that the model did not accurately describe the observed system in the detail that matters and vice versa. This often arises because the observer is not conversant with the postulates and assumptions upon which the model is based and, hence, unintentionally violates a critical condition in the experiment or chooses to ignore the conflict between the two anyway.

Once a model has been demonstrated to fail in the prediction of observations, the model must be discarded and a new theory/model must be devised. This often requires careful assessment of the validity of the postulates and assumptions upon which the model is based, often in the light of new experimental observation and evidence, but under no circumstances can a model be made to "work" by the ad hoc introduction of information that is not based upon observation or accepted theory, as stated above. This admonishment to the scientific community was delivered by Albert Einstein, who, paradoxically, added the cosmological constant in an ad hoc manner to the Special Theory of Relativity, which he reportedly later termed his "greatest blunder" [26].

Not all experiments must be physicochemical in nature. Thus, "thought experiments" have played a prominent role in the development of scientific theory, the most famous being Einstein's riding of a light beam in the development of the Theory of General Relativity. Unlike physicochemical experiments, thought experiments do not introduce new empirical data. Nevertheless, they often result in conclusions via inductive/deductive reasoning from their starting postulates, particularly when a physicochemical experiment under the relevant conditions is impossible. Thus, Einstein had no practical means of observing the natural world at velocities of the order of that of light. It is the invocation of these particulars that gives thought experiments their experiment-like appearance. When

properly constructed, thought experiments often enlighten the consequences of various postulates and assumptions and can lead to identifying critical experiments that may be performed to test a theoretical prediction.

Finally, it is necessary to briefly introduce the role of complexity in the development of models. Complexity is a state in which the behavior of a system is obscured by the interaction of constituent components in multiple ways that follow local rules, the impact of which are not immediately resolvable using the tools currently at hand [27]. Colloquially, it may be likened to driving along a road on a foggy night; your view of distant objects is obscured by the opacity of the fog and the limited ability of your headlights on low beam to penetrate the fog. However, upon switching to a high beam, previously obscured objects now become visible. The "low beam headlight" in the scientific case is our limited intellectual and sensory ability to observe the issues before us. We extend our senses using instruments (e.g., electron microscopes, telescopes, and so forth) and we extend our intellect using computers. This is necessary because while our brains are capable of discerning relationships between related objects, they are unable to compute rapidly or accurately. Since many of the constitutive equations in complex, deterministic models are highly complex themselves, often being high order, non-linear, coupled differential equations for which analytical solutions are not readily available or possible, the equations can only be solved by using high-speed digital computers. Recall that, in the 1960s, one of the great challenges in physics was the "many-bodied" problem, in which one sought to describe the interactions between three or more particles, all of which mutually interacted. A manifestation of this problem was the inability to accurately describe the electronic energy levels of helium (a three-body problem) and higher atoms. This changed with the introduction of high-speed computers that can extend our intellects to accurately describe the electronic structures of virtually all atoms in the periodic table. Indeed, computers have enabled a greater advance of science over the past three decades than was accomplished in all preceding history.

## 4. Model Building

The art of model building has evolved over centuries in a manner that is still not fully appreciated in the scientific community, primarily because the subject is seldom taught in universities. Indeed, students are commonly expected to "know" model-building skills as though they were part of their genetic code, like having blue eyes! My interest in the subject stemmed from contentious issues that I had with models that were portrayed by their proponents to be capable of predicting the accumulation of stress corrosion cracking damage in the primary coolant circuits of water-cooled nuclear reactors. Close inspection showed that the models either did not possess a general theoretical basis (or even a local one, in many cases) for the phenomenon of interest (i.e., crack growth), or they were simply empirical correlations, albeit sometimes quite sophisticated correlations, despite being claimed to be "mechanistic" in nature. That resulted in my developing a graduate course in the Department of Materials Science and Engineering at the Pennsylvania State University titled "Theories and Models in Science and Engineering", upon which the present paper is partly based. It should be noted that all deterministic models contain a "mechanism", as described by the constitutive equations, but not all mechanistic models are deterministic if they may lack the constraints imposed by the natural laws.

Ideally, model building requires that the database upon which the model is to be based be developed first, and from that database a general theory is derived that subsequently yields a model that is the calculational arm of the theory. The reason for this is that we should always seek to develop a general theory/model that accounts for all known facts about the system. However, this is seldom done in practice because in principle we can never know all the facts about a given system, and further experimentation continually uncovers new facts. The best that can be done is to base a theory on all the known facts, in which case it is referred to as being a "Global Theory". Thus, we term a theory that is based upon facts that do not encompass all that is known about the system a "Local Theory", and

it is usually developed because the researcher seeks, regrettably, to account for only their own experimental findings while ignoring the findings of others. The use of local theories is discouraged because they tend to be less complete than general or global theories and are not as effective in advancing the field.

Stress corrosion cracking (SCC) is a form of localized corrosion that, generally, falls within the differential aeration hypothesis (DAH) that was first advanced by Evans in 1931 [28] (Figure 2), and which forms the theoretical basis upon which the CEFM is based [7–21].

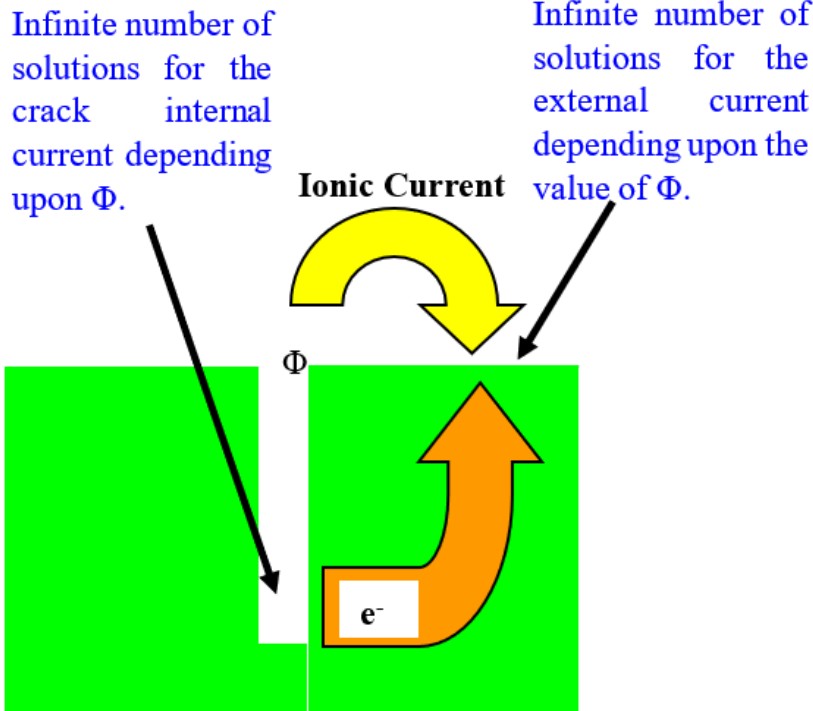

**Figure 2.** Schematic of the origin of the coupling current in stress corrosion cracking, according to the CEFM. The coupling current is required by the differential aeration hypothesis for localized corrosion, and the conservation of charge requires that the electron current flowing from the crack to the external surface must be equal to the positive ionic current flowing through the solution from the crack to the external surface.

The phenomenon that we seek to model is the intergranular stress corrosion cracking (IGSCC) of sensitized Type 304 SS in Boiling Water (Nuclear) Reactor (BWR) primary coolant circuits [29]. Although the Coupled Environment Fracture Model (CEFM), the development of which is the basis of this paper, was initially restricted to Type 304 SS, because of the urgency of mitigating stress corrosion cracking in operating BWRs worldwide that employed that steel in the primary coolant circuit, the model has now been extended to stress corrosion cracking in general in a variety of other alloys [30–34]. The principal innovation brought forth by the CEFM is the recognition that the current generated at the crack tip must be largely consumed on the external surfaces via charge transfer reactions involving redox species in the environment (e.g., $H^+$, $H_2O$, $O_2$, and $H_2O_2$), so that charge is conserved in the entire system. This required that the external environment/surfaces be included in formulating the model, something that had not been done in an analytical manner in previous models.

The morphology of IGSCC in sensitized Type 304 SS is depicted in Figure 3 [35]. IGSCC produces an intercrystalline morphology with evidence of grain boundary separation and with the fracture path being along the grain boundaries (Figure 3b,c), in contrast to the mechanical tearing morphology displayer in Figure 3a [35]. It appears that a crack can nucleate from both pits and at emergent grain boundaries.

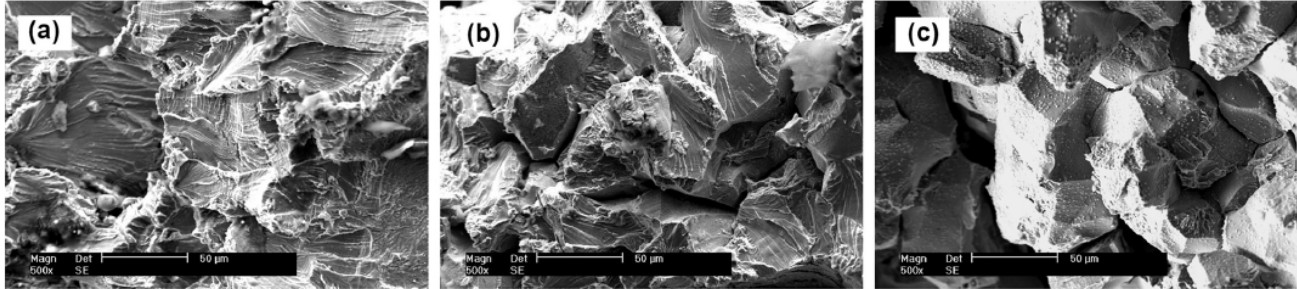

**Figure 3.** Micrographs of IGSCC in Type 304 SS in high-temperature water: (**a**) annealed; (**b,c**) sensitized [35]. This figure was published in Corrosion Science, 146, Z. Zhang, X. Wu, and J. Tan, "In-situ monitoring of stress corrosion cracking of 304 stainless steel in high-temperature water by analyzing acoustic emission waveform", pp. 90–99, Copyright Elsevier 2019.

Some evidence exists showing that, in the latter, the emergent grain boundary is wedged open, most likely by corrosion products formed by the reaction of chromium with water that has penetrated down the emergent boundary, a mechanism that was first proposed by Scott et al. [36] and later developed by Macdonald et al. [34]. In both cases (nucleation at a pit or at an emergent grain boundary), the crack nucleates when the stress intensity factor $K_I > K_{ISCC}$, where $K_{ISCC}$ is the critical value of $K_I$ for crack propagation, which requires the presence of residual or applied tensile stress and/or a defect dimension of a certain minimum magnitudes.

The literature on cracking in the structural alloys employed in water-cooled nuclear power reactors is large, and a comprehensive review of the data is well beyond the scope of the present paper. Some of the data that have been obtained by Ford et al. [37] are summarized in Figure 4, along with data calculated using the CEFM (solid lines). These plots display the characteristic sigmoid form of Log(CGR) vs. ECP in "pure" water at constant $T$ (288 °C), conductivity, $K_I$ (25 MPa·m$^{1/2}$), and DoS (EPR = 15 C/cm$^2$), with the lower limit being defined by the creep crack growth rate ($1.69 \times 10^{-10}$ cm/s) and the upper limit being controlled by the mass transport of the cathodic depolarizer (e.g., O$_2$) to the external surface. Also shown is the positive impact that solution conductivity has on the CGR at a given ECP, a relationship that will be explored later in this paper.

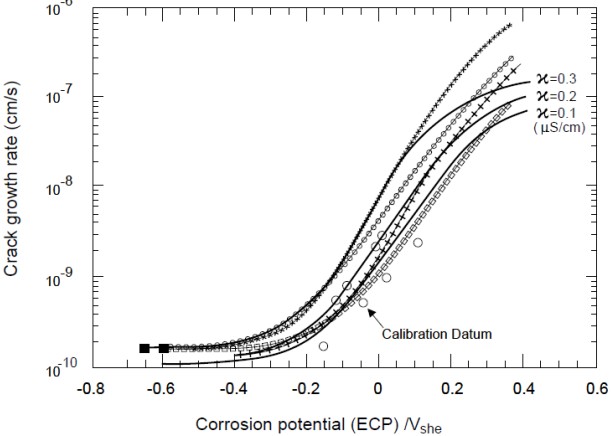

**Figure 4.** Compilation of crack growth rate (CGR) data for intergranular stress corrosion cracking on sensitized Type 304 SS in Boiling Water (Nuclear) Reactor (BWR) primary coolant at 288 °C as a function of the electrochemical potential (ECP) and the system parameters, as follows: 25 mm C(T) specimen, $K_I$ = 27.5 MPa.m$^{1/2}$, ambient (25 °C) solution conductivity = 0.1 to 0.3 μS/cm, DoS (EPR) = 15 C/cm$^2$. Selected data from Ford et al. [37] (open circles). Solid dark lines are CGR calculated using the CEFM, other symbols/lines are from Ford et al. [37].

As noted above, a positive "coupling current" flows through the solution from the crack tip to the external surfaces, while an equal electron current flows through the metal in the same direction (Figure 2). The two currents annihilate quantitatively via a charge transfer reaction (e.g., reduction of oxygen and/or hydrogen evolution). As indicated below, the measurement of the coupling current (CC) provides unprecedented insight into the processes that occur during stress corrosion cracking, and it has always puzzled the author why CC analysis is not more extensively employed.

The specific steps in model building are, therefore:

- Collate property data—a valid "global" theory must account for all the known properties of the system;
- Formulate hypotheses, postulates, and assumptions. These must agree with our empirical knowledge or theoretical expectation of the system;
- Specify the "mechanism", and hence the "constitutive equations";
- Specify the "constraints" (e.g., conservation equations, Faraday's law of mass-charge equivalency);
- Solve the equations and predict the output;
- Compare the output with the experimental data and adjust the model to make new predictions that are in better agreement with the experiment;
- The last step is repeated until no amount of valid adjustment can make the model "work" by accounting for new observations within experimental uncertainty. The theory/model is then rejected, a new theory/model is developed, and the process starts over again;
- Finally, it is important to recognize that modeling is always a compromise between complexity and mathematical tractability. After a certain threshold, the modeler must make a compromise by either simplifying the model (e.g., reducing the number of species considered, and hence the number of independent variables) or by invoking assumptions to simplify the mathematics, or both. This is particularly important in the development of analytical models that may require numerical solutions of coupled high order differential equations for which analytical solutions do not exist.

For the specific cases of the IGSCC in sensitized Type 304 SS in Boiling Water (Nuclear) Reactor (BWR) primary coolant circuits and for IGSCC in milled-annealed Alloy 600 in Pressurized Water Reactor (PWR) primary coolant circuits, the empirical data show that the CGR depends primarily on both electrochemical independent variables (potential, pH, conductivity) and mechanical/metallurgical independent variables (stress intensity, DoS, hardness, cold work, etc.) as described, for example, by Ford et al. [37] for cracking in BWR coolant circuits. Upon compilation of the database [38,39], it was found to be sparse because some parameters were not reported in the original studies. For example, because most SCC studies have been performed by researchers in mechanical or nuclear engineering communities, until recently, electrochemical parameters, such as the ECP, pH, and solution conductivity were often not measured or reported. However, these parameters can be calculated with sufficient accuracy using a Mixed Potential Model (MPM) [40], solution equilibrium theory [41], and ionic conductivity theory [42], respectively. In this way, the databases can be populated with the identified independent variables. In doing so, a database of several hundred CGR ($T$, ECP, $\kappa$, pH, $K_I$, EPR) was established with data being taken from laboratory studies and field observation, provided that the independent variables were clearly defined [38,39]. For example, only CGR data reported from fracture mechanics specimens were employed, while those obtained from CERTs (Constant Extension Rate Tests) were rejected because of the poor definition of $K_I$, the lack of clear differentiation of crack initiation and crack growth, the existence of multiple cracks, and the resultant uncertainty in the CGR.

Possibly, the most efficient method for extracting knowledge from a database of this kind is an Artificial Neural Network (ANN) subjected to supervised learning, as presented in Figure 5 [38,39]. The objective of ANN analysis is to establish quantitatively the relationships that exist between the dependent variable (crack growth rate, CGR) and

each of the independent variables ($T$, ECP, $\kappa$, $K_I$, pH, and DOS) that are often hidden in the database. This is best done by using an artificial neural network (ANN) in the pattern recognition mode, using backpropagation and error minimization, resulting in the optimal weights between neurons. The ANN employed in Refs. [38,39] has one input layer, one output layer and three hidden layers, with each neuron having a sigmoid transfer function, which imbues the net with a certain "fuzziness" for handing data of lower accuracy. The ANN chosen for this work was taken from the MATLAB Neural Network Toolbox software.

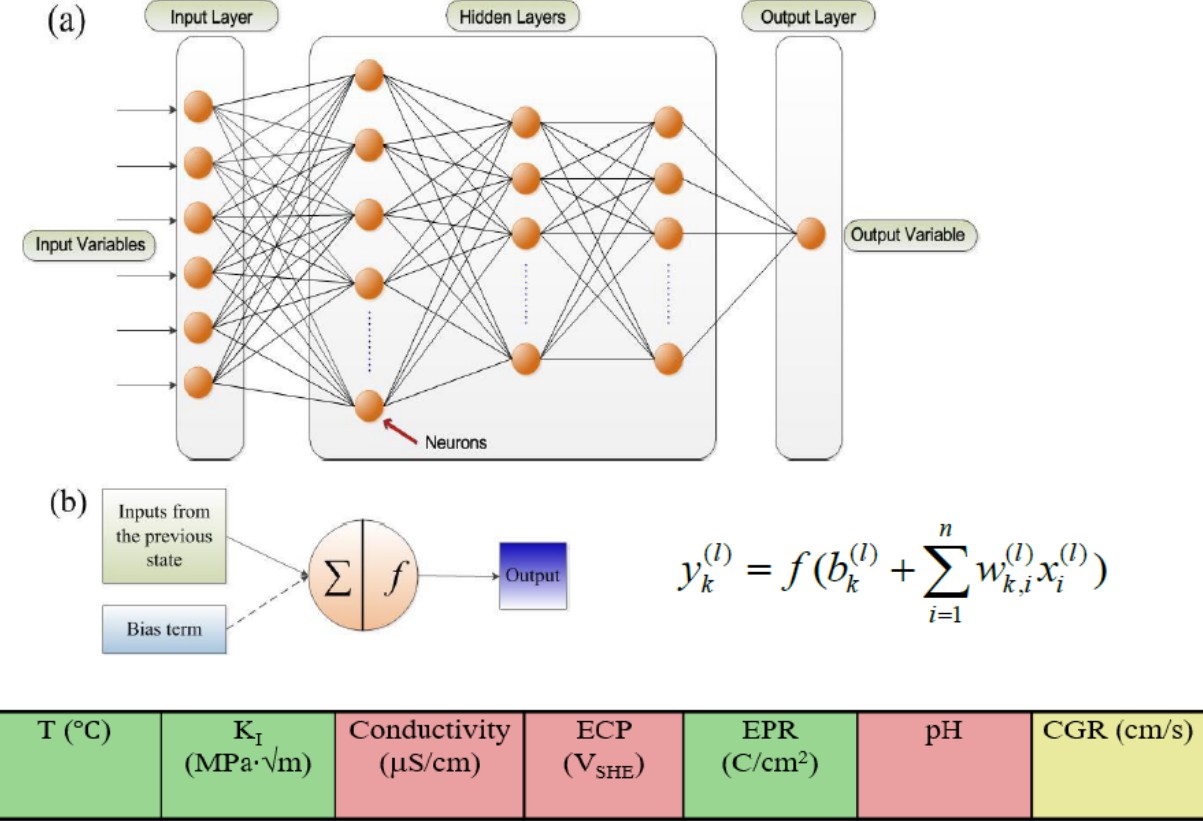

**Figure 5.** (**a**) The topology of the artificial neural networks (ANN) used in this work; (**b**) schematic of each neuron in the network. $\Sigma$ signifies summation, while $f$ represents the transformation (sigmoid function). After [38,39].

The raw data of CGR vs. ECP from the database that was established for that work [38,39] are presented in Figure 6a. The reader will note that, when presented in this format (CGR vs. ECP), the data are scattered over three orders in magnitude, rendering them essentially useless for analytical engineering use (e.g., in predicting the service life of a structure). Part of the problem lies in representing data for a multivariate function in two dimensions vs. a single independent variable (ECP in this case); however, the major issue is that the hidden relationships between the dependent and independent variables cannot be gleaned by inspection. Data sets of this type are ideal candidates for ANN analysis, which seeks to uncover the hidden relationships between the dependent variable (CGR) and the independent variables ($T$, ECP, $\kappa$, pH, $K_I$, EPR). In doing so, the database must contain sufficient content related to each independent variable that the ANN can converge on a solution. Once the database is established, 70% of the data, selected at random, are used to train the net, 15% are employed to validate the prediction of the net (i.e., they act as the "known" cases), and the remaining 15% are used to further test the predictions of the net (the test set contains no known CGR data) (Figure 5b).

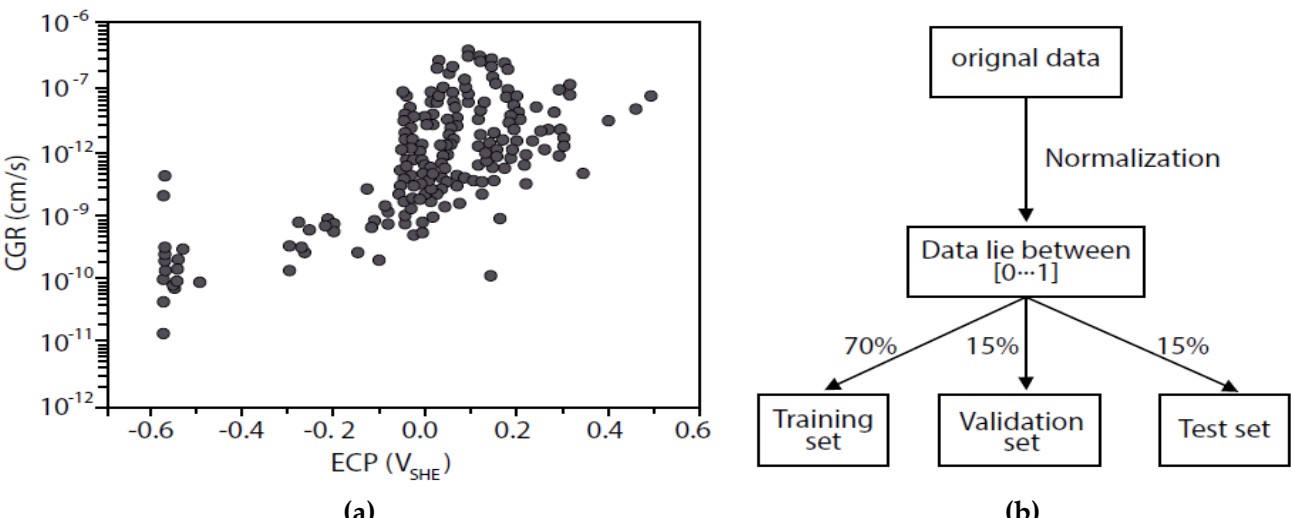

**Figure 6.** Raw normalized CGR for IGSCC in Type 304 SS in BWR primary coolant plotted as a function of ECP (**a**), and the division of the data into training, validation, and testing subsets (**b**) [39].

The training of the net comprises of first arbitrarily choosing the weights of the connections between each neuron in a layer with those in the preceding and following layers, and then inputting the independent variables into the net and calculating the dependent variable (CGR) and comparing it with the known (measured) value for that set. The weights are then re-adjusted cyclically as additional sets are exposed, so that the calculated dependent variable and the known (measured) values are minimized. This process is known as "supervised learning"; in reality, it is best described as "optimization". Once the differences between the ANN-calculated and known (measured) dependent variable values are minimal, as defined by the operator, training is stopped, and validation and testing begin. Further details can be found in Shi et al. [38,39]. An important feature of an ANN is that the net may be continually updated and refined by the inclusion of new data into the empirical database and in this sense, an ANN is a "living model".

The trained net is used to calculate the CGR for each set in the training and validation sets with the measured CGR for each entry in the sets, and the calculated and known CGR are plotted in a log-log format, as shown in Figure 7. If the prediction were perfect, the data would fall upon a straight line of 45° slope with zero scatter. However, the input data contain experimental and, possibly, other errors, so that deviation from the diagonal is expected and is found (Figure 7). The ANN is found to produce a diagonal correlation, with 95% of the data falling within ±0.4 log unit [38,39]. Thus, the net demonstrates that the CGR data measured in various laboratories by different researchers and those taken from the field are quite internally consistent, in contrast to the data being first judged to the contrary as presented in Figure 6a, demonstrating that the data are sufficiently accurate for analytical engineering purposes.

From the weights between the various neurons in the ANN (Figure 5), it is possible to calculate the relative contributions of the various independent variables to the dependent variable (CGR); that is, the weights define the "character" of the crack growth process in the alloy [38,39]. These contributions are given in Table 1. The first column lists the independent variables and the second states the range over which each independent variable varies in the database. The third column presents the contributions for IGSCC in sensitized Type 304 SS [38], while Columns 4 and 5 present the corresponding data for Alloy 600 [39].

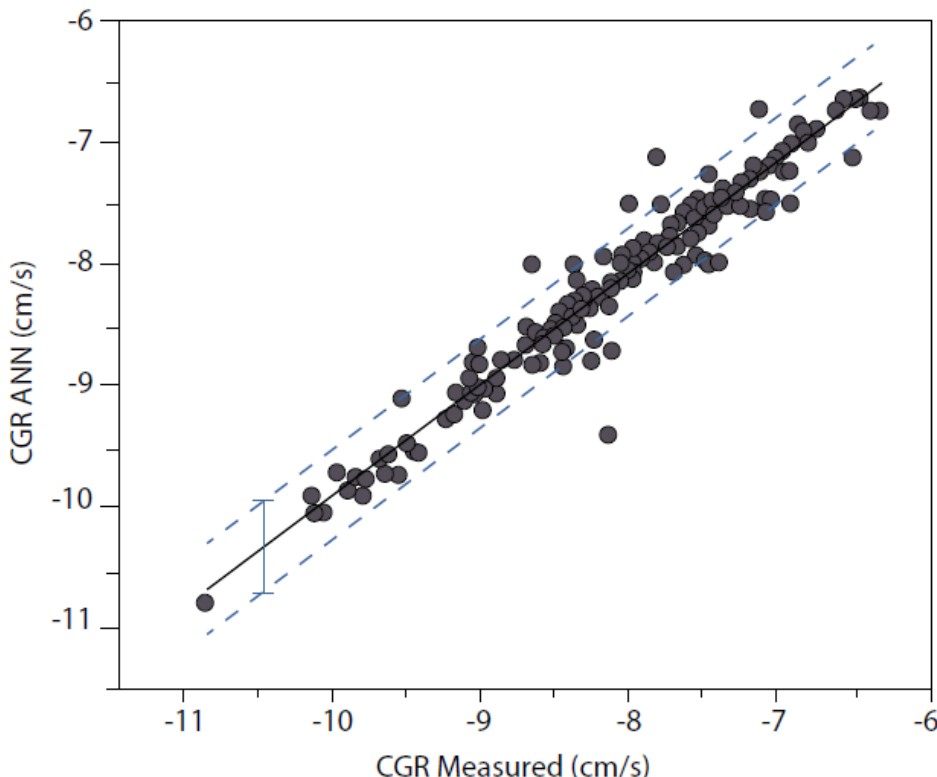

**Figure 7.** Logarithmic plot of ANN predicted CGR against measured CGR in the training and validation subsets of the database. The broken lines define the 95th percentiles of ±0.4log (CGR CEFM). After Shi et al. [38,39].

**Table 1.** Contributions of various independent variables to IGSCC in sensitized Type 304 SS and Alloy 600 in simulated BWR primary coolant and simulated PWR coolant, respectively [38,39].

| Independent Variable | Range | Type 304 SS in BWR Primary Coolant | Range | Alloy 600 in PWR Primary Coolant |
|---|---|---|---|---|
| Temperature (°C) | 25–292 | 17.8 | 290–360 | 18.6 |
| ECP ($V_{she}$) | −0.575–0.496 | 43.6 | −1.096 to −0.610 | 14.1 |
| Stress Intensity Factor (MPa·$\sqrt{m}$) | 10.4–67.78 | 10.8 | 4.6–101 | 15.2 |
| Conductivity (μS/cm) | 0.52–5.72 | 14.0 | 1.7–116 | 14.1 |
| Degree of Sensitization (DoS) (C/cm$^2$) | 0–33.79 | 13.8 | - | - |
| Yield Strength (MPa) | N/I | - | 211–500 | 12.0 |
| [LiOH] ppm | N/A | - | 0–10 | 4.0 |
| [$H_3BO_3$] ppm | N/A | - | 0–1800 | 7.6 |
| pH | N/A | - | 5.52–9.19 | 14.5 |

For sensitized Type 304 SS in simulated BWR primary coolant, the most important contributions are from environmental variables (ECP > temperature > conductivity) followed by the metallurgical variable (DoS), and finally from the mechanical variable (stress intensity factor) [38]. Insufficient data were reported for the carbon content, heat treatment protocols and parameters, yield strength, hardness, extent of cold work, etc., to include those variables in the analysis, even though their impact on IGSCC in sensitized Type 304 SS is well-known. Furthermore, the pH for pure water is determined by the dissociation constant for water, assuming the absence of acidic or basic impurities, and, therefore, is

not an independent variable. The contributions to the character demonstrate that IGSCC in sensitized Type 304 SS in simulated BWR primary coolant is primarily an electrochemical process augmented by metallurgy and mechanics.

A similar analysis of IGSCC in mill-annealed Alloy 600 in simulated PWR primary coolant ($H_3BO_3$/LiOH) was also reported by Shi et al. [39], in which the independent variables gleaned from the database are $T$, ECP, $K_I$, conductivity, yield strength, [LiOH], [$H_3BO_3$] and pH. The concentrations of boric acid and lithium hydroxide typically vary from 2000 ppm to 0 ppm and 0 ppm to 4 ppm, respectively, from the beginning to the end of a fuel cycle during normal power operation. Since the pH is determined by [LiOH] and [$H_3BO_3$], strictly pH is, again, not an independent variable. However, it was included in the analysis because pH does have a discernible effect on the CGR. As shown in Table 1, the character of IGSCC in Alloy 600 is markedly different from that in Type 304 SS, being equally environmental ($T$, ECP, conductivity, pH), metallurgical (yield strength), and mechanical ($K_I$) in character. In both cases, electrochemistry (ECP, conductivity, pH) plays an important role in determining CGR, but electrochemistry was ignored for many years because such studies tended to be carried out in Mechanical Engineering and Nuclear Engineering departments in universities and in research institutes/national laboratories where electrochemical expertise was minimal. In the author's opinion, this reflected the fact that electrochemistry is seldom included in the teaching curricula in those disciplines. Regardless of the tortured path taken to include electrochemistry in such studies, any viable theory and resulting model must account for the characters identified above, including the electrochemical character.

Few techniques have proved to be effective in probing the events that take place at the tip of a stress corrosion crack as it propagates through a metal. Acoustic emission indicates that, frequently, crack advance is a cyclical, intermittent process in which each advance emits an acoustic wave that is detected by a sensor attached to the metal sample [35,43,44]. Another technique that has proven effective in obtaining crack tip information is to monitor the coupling current that flows through the specimen from the crack tip to the external surface (Figure 2) [45]. An early version of this technique was reported by Williams et al. [46], who monitored the current flowing from a cracked, round Type 304 SS tensile specimen under active load to a remote cathode under controlled polarization conditions in high-temperature water. Accordingly, the specimen was not under the natural, open circuit condition that exists in a reactor coolant circuit, and a remote cathode does not effectively simulate the external surface with a few tens of CODs from the crack centerline. Nevertheless, that study also demonstrated that crack advance was intermittent, as evidenced by transients in the current.

Manahan et al. [45] devised a way of monitoring the actual coupling current by coating a pre-cracked C(T) fracture mechanics specimen with PTFE, so as to insulate the external surface from the environment, and then mounting cathodes on the side surfaces, with the cathodes being connected to the specimen via a zero-resistance ammeter that holds the specimen and the cathodes at the same potential as would be the case if the cathodes were part of the actual specimen. Thus, the coupling current is routed through the ZRA, where it is measured as a voltage from the control amplifier. The study involved employing different cathodes (stainless steel, platinized nickel, and titanium) to explore the impact of the external surfaces on the coupling current (CC), and hence on the CGR (see below), and involved different loads to ascertain the impact of the stress intensity factor ($K_I$). The measured CC and load are plotted in Figure 8 as the specimen was loaded and unloaded to/from various $K_I$ values in dilute $Na_2SO_4$ solution at 252–288 °C under static autoclave conditions.

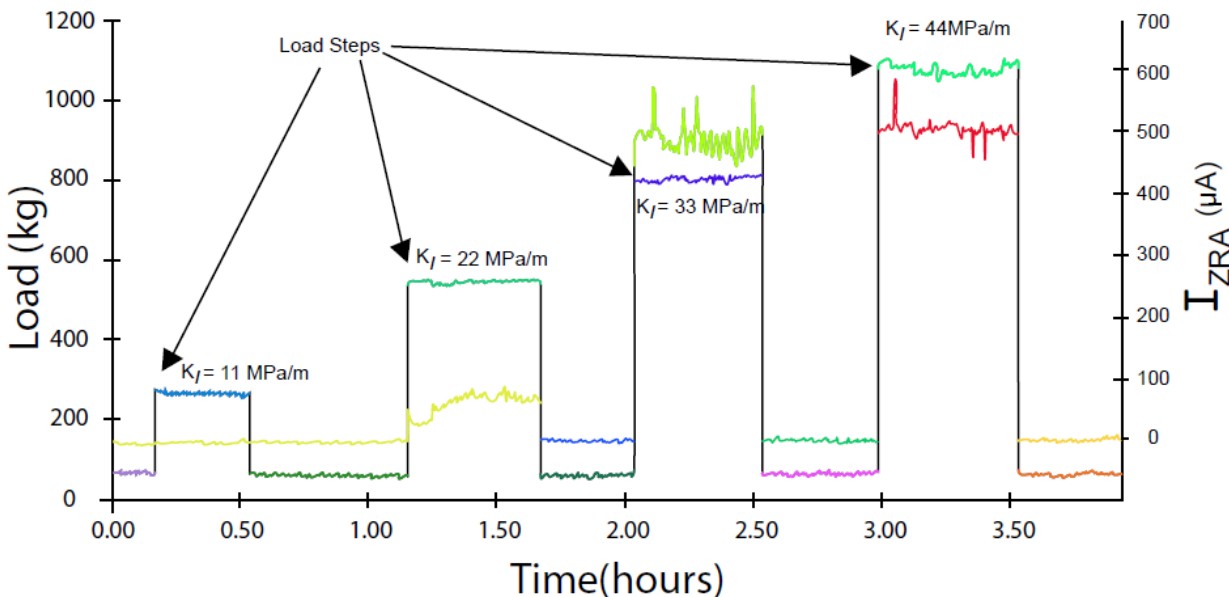

**Figure 8.** Plot CC and load for crack propagation in sensitized Type 304 SS in simulated Boiling Water (Nuclear) Reactor (BWR) primary coolant at 252–288 °C [45].

The cathodes first used in this experiment were platinized nickel, because it was feared that the coupling current might be too small to measure accurately. Instead, the current was found to be surprisingly large; 500 μA at $K_I$ = 44 MPa·m$^{1/2}$. Since the current is easily measured down to the sub-picoamp level with standard equipment, our initial fears of an immeasurably low CC were unfounded. In any event, the first load increment to $K_I$ = 11 MPa·m$^{1/2}$ does not produce a CC response, showing that SCC has not activated sufficiently to be detectable, but an increment to 22 MPa·m$^{1/2}$ does elicit a response. Unloading reduces the CC to zero, which is attributed to crack closure. Further increments in $K_I$ result in larger CC responses, but the response saturates for $K_I$ > 33 MPa·m$^{1/2}$ at about 500 μA. The CC was measured at an acquisition rate of 1 Hz, which proved to be too low to capture the fine structure in the current record, so that the CC appears to be randomly noisy. A second experiment was performed using uncatalyzed Type 304 SS cathodes in which the CC was measured at 100 Hz, and the CC record is given in Figure 9.

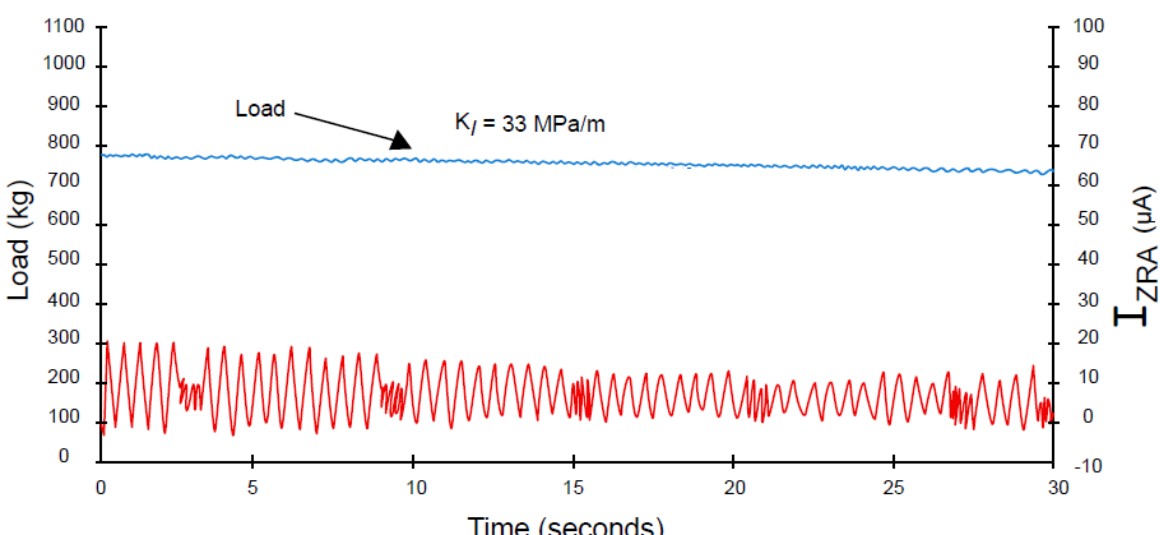

**Figure 9.** CC record for crack propagation in sensitized Type 304 SS in simulated Boiling Water (Nuclear) Reactor (BWR) primary coolant. Data acquisition frequency = 100 Hz. After Manahan et al. [45].

In this case, the CC comprises packages of between four and thirteen oscillations, with each package being separated by a brief period of intense activity that apparently contains a similar number of oscillations but at a much higher frequency. Thus, not only does the crack advance intermittently, but the frequency of the oscillations is bimodal, apparently depending upon the orientation of the grain boundary with respect to the principal stress. The lower frequency oscillations are attributed to the climb of the crack up a grain face that is unfavorably oriented with respect to the principal stress axis (i.e., not normal), such that the resolved stress tensioning the crack tip is sufficiently low that the microfracture frequency is also low due to a correspondingly low crack tip strain rate. Once the crack climbs to the top of that face, it encounters a grain boundary (and hence a grain face) that is more favorably oriented to the principal stress axis, and the microfracture frequency greatly increases and the boundary "unzips" in a succession of rapid microfracture events. This period corresponds to the intense activity observed between each package in Figure 9. Note that, in this case, because uncatalyzed cathodes were employed, the mean CC is only about 10 μA, a factor of about 50 lower than that observed with the catalyzed Ni cathodes. Note, further, that some oscillations result in a negative CC, albeit only momentarily, indicating crevice reversal that has been observed in other systems [47–49].

From the current record in Figure 9 we can calculate the microfracture frequency (MFF) of the slow climb of the crack up a grain face that is not favorably oriented with respect to the principal stress axis and a plot of MFF is plotted in Figure 10 as a function of $K_I$ for two cathodes (Type 304 SS and titanium) [45]. The plot shows that the MFF is initially zero (no microfracture events) for $K_I < 10$ MPa·m$^{1/2}$, but increases sharply with an increase in $K_I$ to 11 MPa·m$^{1/2}$. At higher loads, the MFF increases only modestly corresponding to the Stage II region of the CGR vs. $K_I$ correlation. The MFF is seen to be essentially independent of the type of the cathode (Ti vs. Type 304 SS), which indicates that the kinetics of oxygen reduction on the external surfaces are also little different since it is unlikely that the crack tip strain rate is a function of the kinetics of the oxygen reduction reaction. This seemingly surprising result is understandable when one notes that the exchange current density ($i_0$) and the Tafel constants depend upon the thickness ($L_{ss}$) but not the identity of the barrier oxide layer on the surface [50,51], because electronic charge carriers (electrons and electron holes) must quantum mechanically tunnel through the layer. The exchange current density on a passive surface may be expressed as $i_0 = \hat{i}_0 exp(-\hat{\beta} L_{ss})$, where $\hat{i}_0$ is the exchange current density on the hypothetical bare metal and $\hat{\beta}$ is the tunneling constant that can be estimated from quantum mechanical tunneling theory (QMT) or measured experimentally [50]. Since $\hat{i}_0$ for any given redox reaction appears to be similar on the bare surfaces of many metals of the same group (e.g., transition metals and their alloys), the difference in this parameter on passive surfaces lies in the differences in $L_{ss}$. However, at the equilibrium potential of the oxygen electrode reaction for the same T, pH, and [$O_2$], the barrier layer thicknesses appear to be similar on stainless steels and Ti, although this conclusion is based upon somewhat sparse and equivocal data because no exchange current or Tafel constant information is available for Ti at elevated temperatures. However, modeling by Sutanto and Macdonald [52] of IGSCC in weld-sensitized heat-affected zones in simulated BWR primary coolant circuits suggests that the quantum mechanical correction to the exchange current density is not particularly important, at least in this case.

If the microfracture events that occur at the crack tip are semi-circular in form with a radius of $r$, and noting that, on average, $B/2r$ events must occur on the crack front for the crack to advance by $r$ cm, the crack growth rate may be expressed as [45]

$$\frac{da}{dt} = \frac{2r^2 f}{B} \tag{1}$$

where $f$ is the MFF (~2 s$^{-1}$) and $B$ is the width of specimen (1.27 cm). Rearranging Equation (1) yields

$$r = \left[\frac{B(da/dt)}{2f}\right]^{1/2} \tag{2}$$

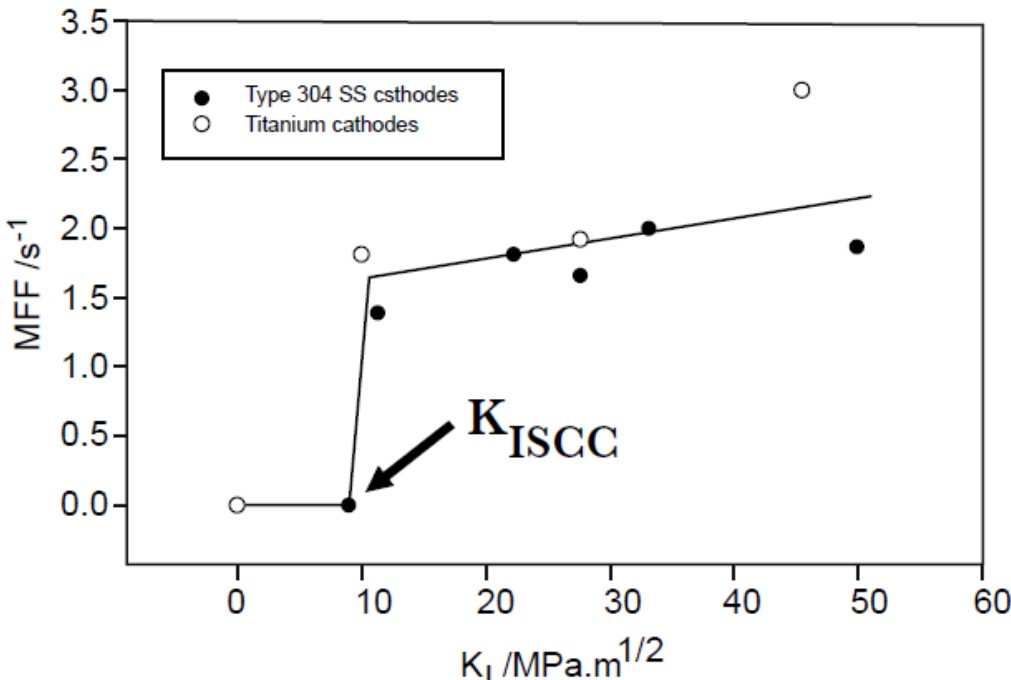

**Figure 10.** Microfracture frequency vas $K_I$ for IGSCC in sensitized Type 304 SS in simulated Boiling Water (Nuclear) Reactor (BWR) primary coolant [45].

Noting that, under the conditions of the experiment, $da/dt \sim 3.1 \times 10^{-7}$ cm/s [12], we find that $r \sim 3$ µm. Given a grain size of 20–50 µm and that the distance traveled by the crack during the "slow" advance stage represents about one half of the grain size, there should be roughly four to ten events in each package, which is in good agreement with the experiment. It is important to note that the value of $r$ is too large to be attributed to slip, for which the microfracture dimension (MFD) should be some small multiple of the Burger's vector or a few nm. We conclude that while slip is responsible for the intermittent events, the MFD is determined by hydrogen-induced cracking (HIC) of the matrix ahead of the crack tip, with that matrix being susceptible to HIC possibly because of the presence of strain-induced martensite. The source of hydrogen is the hydrogen evolution reaction via proton reduction that occurs on the metal at the crack tip in contact with the highly acidic environment that is maintained by differential aeration, assuming a diffusivity for H in Type 304 SS of about $10^{-12}$ cm$^2$/s [53], and noting that, to a first approximation, the diffusion length of hydrogen at 288 °C, $x_d > (D/f)^{1/2} = 7 \times 10^{-6}$ cm. An embrittlement dimension of $100x_d$ is not unreasonable, as the MFD likely reflects the momentum of the microfracture event once the crack has initiated in the embrittled phase ahead of the crack tip due to slip, or it may reflect the spacing of voids (possibly at precipitates, such as $Cr_{23}C_6$) that nucleate on the grain boundary ahead of the crack tip. The resulting dimension (1.4 µm) is of the same order as the MFD calculated above from the coupling current noise. Finally, in this case, the remarkable conclusion is that the crack advances fracture event by fracture event, with minimal overlap between events. However, this is not the case for IGSCC in sensitized Type 304 SS in thiosulfate solution at ambient temperature, where it is found that extensive overlapping occurs due to microfracture events occurring more-or-less simultaneously at different points on the crack front, resulting in a coupling current comprising structured noise [54].

In summary, any viable model for the IGSCC in sensitized Type 304 SS must account for the following observations.

- The crack growth rate (CGR) increases roughly exponentially with the potential of the metal at sufficiently high potentials. At lower potentials, the CGR is potential-independent, corresponding to mechanical creep fracture (Figure 4);
- In the case of IGSCC, the crack propagates intergranularly, giving rise to the intergranular crack pathway (Figure 3). In other cases, the crack propagates across the grain in a process termed transgranular stress corrosion cracking (TGSCC), and in still other cases mixed mode (IDSCC/TGSCC) may be observed [55];
- The CGR increases with the DOS, the yield strength, the hardness, and the extent of cold work [37];
- SCC only occurs if the stress intensity factor ($K_I$) exceeds a lower limit, $K_{ISCC}$. The upper limit of the stress intensity factor, $K_{Ic}$, is defined by unstable, mechanical fracture. Between these limits, cracking occurs via stress corrosion cracking, with the CGR increasing sharply with $K_I$ in a Stage I region and then progressing almost independently of $K_I$ in a Stage II region [55];
- A coupling current is observed to flow through the solution (including that in the crack) from the crack tip to the external surfaces, where it is annihilated by the corresponding electron current flowing through the metal via a charge transfer reaction (e.g., oxygen reduction) [7–21,45,46]. The environmentally mediated CGR is proportional to the magnitude of the coupling current [56];
- Oscillations appear in the coupling current that are attributed to microfracture events at the crack tip. In the case of IGSCC in sensitized Type 304 SS, as described in Ref. [45], the oscillations come in packages that are separated by brief periods of intense activity;
- Coating the external surfaces with an insulator, and hence inhibiting the reduction of oxygen, causes the coupling current to sharply decrease and the crack to be reduced accordingly [57];
- Catalyzing the reduction of oxygen on the external surface results in an increase in the coupling current and, hence, an increase in the CGR [7–21,45];
- The crack growth rate is a positive function of the solution conductivity (i.e., the CGR increases with increasing conductivity) [7–21,37];
- The character of an SCC model contains contributions from electrochemistry, mechanics and metallurgy, as demonstrated by the ANN analyses reported by Shi et al. [38,39]. The model engine must contain mechanistic concepts and relationships that allow the model to predict the character without the input of any additional information;
- Enhanced mass transfer of oxygen to the external surfaces increases the CGR [58]. For a sufficiently short, open crack, increasing the flow rate may destroy the aggressive conditions that develop within the cavity and hence inhibit crack growth;
- CGR vs. temperature commonly follows an Arrhenius relationship, but in the case of the IGSCC in sensitized Type 304 SS in Boiling Water (Nuclear) Reactor (BWR) primary coolant the CGR passes through a maximum at about 175–225 °C [8,13,59] (see below).

It is shown below that the Coupled Environment Fracture [7–21] accounts for all these observations and, additionally, predicts phenomena that had not been previously detected experimentally. Since its initial formulation, the CEFM has been used to successfully account for SCC in a variety of other alloys, including Alloy 600 [30,31,39], AA5083 [32], Alloy 22 [60], and high strength NiCrMoV (A470/471) turbine disk/rotor steel [33].

## 5. The Coupled Environment Fracture Model

The development of the CEFM was initiated in the late 1980s in response to a Swedish regulator's request for a deterministic model that could be used to calculate CGR in stainless steel components in the primary coolant circuits of BWRs operating in that country [7–21]. A review of previously developed models found none in which the predictions were constrained by the relevant natural laws. Furthermore, if electrochemistry was included it was done so in a "handwaving" manner or was incorporated inadvertently, because the (mechanical) model was calibrated on data that were clearly an unrecognized function of the electrochemistry of the system (as are all SCC CGR data above the critical potential (Figure 4)). As noted above, many previous models had been developed in mechanical/nuclear engineering fraternities that ignored electrochemistry and often metallurgy completely, and incorporated only mechanics in the theoretical basis of the model. In essence, all the previous models were empirical correlations, albeit with some being quite sophisticated correlations, as previously noted.

### 5.1. Default Conditions

To minimize the repetitive statement of the values of parameters employed in the calculations reported in this paper, two sets of default parameters are defined, as presented in Tables 2 and 3. Should a calculation be performed for non-default conditions, the actual parameter values employed that are different from the default values will be given in the caption.

**Table 2.** Default system parameter values for calculating CGR in sensitized Type 304 SS in simulated BWR primary coolant.

| Parameter | Value | Comments |
| --- | --- | --- |
| T | 288 °C | Operating temperature of a BWR. |
| COD | 0.001 cm | Typical of a tight crack |
| Crack width | 1.0 cm | Assumed |
| Crack length | 0.5 cm | Assumed for a "standard crack" |
| Pipe hydrodynamic diameter | 50 cm | Typical of BWR recirculation system |
| Flow velocity | 100 cm/s | Assumed |
| Stress intensity factor | 27.5 MPa.m$^{1/2}$ | Assumed |
| $O_2$ concentration | 100 ppb | Typical of BWR under Normal Water Chemistry (NWC) |
| $H_2$ concentration | 1 ppb | Assumed |
| $H_2O_2$ concentration | 1 ppb | Assumed |
| Degree of sensitization (EPR) | 15 C/cm$^2$ | Typical of weld sensitization of Type 304 |

### 5.2. Constitutive Equations and Constraints

Retuning now to Figure 1, and with reference to Figure 10, we note that the objective in modelling SCC is to solve for the current and potential distributions along the path through the solution from the crack tip to the external surface at an effectively infinite distance from the crack mouth; i.e., at a distance at which the crack has no further influence over the processes that occur on the surface, and at which the local net current density is zero, as expected for the open circuit condition. Modeling shows that this distance is a few tens of the COD, but the actual value depends upon the independent variables ($T$, ECP, $K_I$, $\kappa$, flow velocity, etc.) [7]. These distributions are obtained by solving a set of Nernst–Planck equations for the flux ($J_i$) of each ionic species in the system having a concentration $C_i$ and a diffusivity, $D_i$. The first term on the right side of Equation (3) describes diffusional

transport, and the second represent migration in response to the gradient in the electrostatic potential, $\phi$.

$$J_i = -D_i \frac{\partial C_i}{\partial x} - z_i \frac{F}{RT} D_i C_i \frac{\partial \phi}{\partial x} \tag{3}$$

together with the continuity equation

$$\frac{\partial C_i}{\partial t} = -\nabla J_i \tag{4}$$

are solved to yield the flux and concentration of each species down the crack and in the external environment. Of course, this requires statements of the initial ($t = 0$) and boundary conditions at $x = 0$ (crack tip) and $x = L$ (crack mouth) for $t > 0$, which are given in the original publication by Macdonald and Urquidi-Macdonald [7–12] to which the reader is referred for details.

**Table 3.** Default parameter values for the CEFM as used for calculating CGR in sensitized Type 304 SS in simulated BWR primary coolant (T = 288 °C).

| | | |
|---|---|---|
| Atomic volume | $1.18 \times 10^{-23}$ cm$^3$ | Fundamental |
| Fracture strain at the crack tip | $8 \times 10^{-4}$ | Assumed |
| Young's Modulus (E) | $2 \times 10^5$ MPa | Typical of Type 304 SS |
| Dimensionless constant ($\beta$) | 5.08 | Refs. [61–63] |
| Density of the steel ($\rho$) | 8 g/cm$^3$ | Typical of Type 304 SS |
| Yield strength ($\sigma_y$) | 215 MPa | Typical of Type 304 SS |
| Hwang–Gao strain hardening exponent ($n_{HG}$) | 1.7 | Refs. [61–63] |
| Ramberg–Osgood strain hardening exponent ($n_{HG}$) | xxx | Refs. [61–63] |
| Dimensionless constant ($\lambda$) | 0.11 | Refs. [61–63] |
| Shear modulus (G) | $7.31 \times 10^{10}$ Pa | Typical of Type 304 SS |
| Grain boundary self-diffusion constant ($D_{b0}$) | $2.5 \times 10^{-4}$ m$^2$/s | Typical of stainless steels |
| Activation energy for diffusion ($E_{a,D}$) | 168 kJ/mol | Refs. [61–63] |
| Grain boundary diffusion width | $5 \times 10^{-10}$ m | Typical of stainless steels |
| Tafel constant for the HER | 0.065/V | Typical of Type 304 SS [40] |
| Exchange current density ($i_0$) for HER | $5 \times 10^{-4}$ A/cm$^2$ | Typical of Type 304 SS [40] |
| Tafel constant for the OER | 0.071/V | Typical of Type 304 SS [40] |
| Exchange current density ($i_0$) for OER | $5.05 \times 10^{-3}$ A/cm$^2$ | Typical of Type 304 SS [40] |
| Passive current density for the steel | $2.6 \times 10^{-3}$ A/cm$^2$ | Typical of Type 304 SS [40] |
| Standard potential ($E^0$) for steel electrodissolution at the crack tip. | $-0.47$ V$_{she}$ | Calculated from thermodynamics for Fe$^{2+}$/Fe |

The 11 species considered in the current CEFM include Na$^+$, Cl$^-$, SO$_4{}^{2-}$, Fe$^{2+}$, Fe(OH)$^+$, Ni$^{2+}$, Ni(OH)$^+$, Cr$^{3+}$, Cr(OH)$^{2+}$, H$^+$, and OH$^-$. Only the first hydrolysis products of the metal ions are considered, because the low pH at the crack tip ($-1 <$ pH $< 2$) probably precludes the formation of higher hydrolysis products. The set of 11 flux equations are coupled by Poisson's equation that describes the variation of the electrostatic potential in the system:

$$\frac{\partial^2 \phi}{\partial x^2} = -\frac{\rho}{\varepsilon \varepsilon_0} \tag{5}$$

where $\varepsilon$ is the dielectric constant of the medium, $\varepsilon_0$ is the permittivity of a vacuum, and $\rho$ is the charge density;

$$\rho = \sum_{i=1}^{11} z_i C_i \tag{6}$$

As the value of $\varepsilon_0$ is very small, $8.854187 \times 10^{-14}$ F/cm, any small deviation from electroneutrality results in a large gradient in potential, which, of course, activates a large restoring force to reduce the difference in $\rho$ from 0. Nevertheless, to simplify the problem when describing the internal crack environment, we assume that $\rho = 0$, so that Equation (5) collapses into the one-dimensional Laplace equation that predicts that the electrostatic potential, varies linearly with distance down the crack. A consequence of this assumption is that the distribution in potential is different from that obtained by using Poisson's equation [64,65]. However, a comparison of the predicted crack growth rates reveals insignificant differences, so the Laplace assumption was maintained because of the mathematical simplicity obtained. This is possibly due to CGR control residing with the kinetics of the cathodic depolarizing reactions (e.g., oxygen reduction) occurring on the external surface.

Once the distributions in species concentrations and potential have been calculated, the total current flowing in the crack from the crack tip to the external surface can be written as

$$I_{crack} = F\sum_{i=1}^{i=11} z_i a_i J_{i,x=0} = \Theta_I \tag{7}$$

where $a_i$ is the charge weighted fraction of Fe, Cr and Ni in the steel matrix at the crack tip and $J_{i,x=0}$ is the flux of a species at the crack tip. $J_{i,x=0}$ is only non-zero if the species is involved in a reaction at the crack tip. $F$ is Faraday's constant, and $F = 96,485$ C/mol $e^-$.

In the external environment, the potential distribution is obtained by solving the two-dimensional Laplace equation:

$$\frac{\partial^2 \phi_s}{\partial x^2} + \frac{\partial^2 \phi_s}{\partial y^2} = 0 \tag{8}$$

where $\phi_s$ is the electrostatic potential in the solution at the surface (Figure 10a). Justification of the use of Equation (8) is partly based on the fact that under active flow (well-mixing) conditions, the formation of concentration gradients is suppressed, and transport is dominated by migration and convection. The solution of Equation (8) using a 40-term series yields the electrostatic potential close to the surface ($y = 0$) at any distance x from the crack mouth, as [7,9–12]:

$$\phi_s(x,0) = [\phi_s(0,0) - \phi_s^c]exp\left[-B\frac{x}{X_T - x}\right] + \phi_s^c \tag{9}$$

where $X_T$ is the distance on the surface at which the effect of the crack is no longer apparent in the potential. This potential is $\phi_s^c$. Once the potential distribution is calculated (Figure 11), the rates of the redox reactions and the electrodissolution of the steel are calculated as a function of distance across the external surface from the crack mouth, using the generalized Butler–Volmer equation [7,9–12,40]

$$i_{O/R} = \frac{e^{-[\phi_s - \phi_s^e]/b_a} - e^{[\phi_s - \phi_s^e]/b_c}}{1/i_0 + (1/i_{l,f})e^{-[\phi_s - \phi_s^e]/b_a} - (1/i_{l,r})e^{[\phi_s - \phi_s^e]/b_c}} \tag{10}$$

where $i_0$ is the exchange current density, $i_{l,f}$ and $i_{l,r}$ are the mass-transfer limited current densities for the redox reaction $R \rightleftharpoons O + ne^-$ [see Figure 10b], $\phi_s^e$ is the equilibrium potential, and $b_a$ and $b_c$ are the Tafel constants for the forward and reverse directions of the reaction.

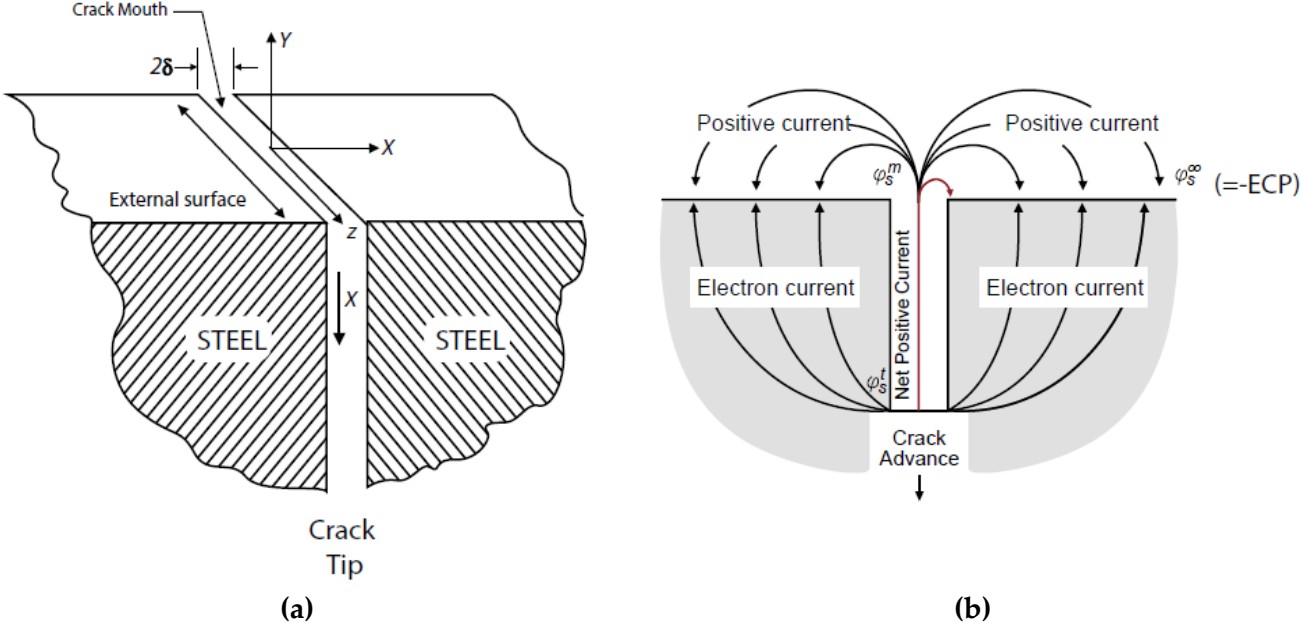

**Figure 11.** (**a**) Schematic of a crack in a metal, defining the coordinate system employed in developing the CEFM [7,9–12]. (**b**) CEFM showing the redox reactions occurring on the surface external to the crack and the flow of the electron current through the metal (the "Coupling Current"), and the ionic current through the solution from the crack tip to the external surfaces where they are annihilated by the charge transfer redox reactions.

In a later version of the CEFM, the exchange current density is calculated from the value on the hypothetical bare surface, knowing the thickness of the barrier oxide layer using quantum mechanical tunneling theory (QMT, see Section 4) [50–52]. The thicknesses of the barrier layer at the equilibrium potential for each of the redox reactions in the external environment are calculated using the Point Defect Model [66,67] from parameter values obtained on the alloys of interest (Types 304 and 316 SS, Alloy 600, and Alloy 690) in high temperature aqueous solutions [34]. This correction takes account of the fact that the barrier layer controls the kinetics of the redox reactions and provides a theoretical basis for interpreting the effect of the external surface condition on the CGR. In the case of the stainless steels, the correction is small and can be ignored to a good approximation.

Note that the potential $\phi_s^m$ (Figure 11) is the potential in the solution with respect to the metal or, more conventionally, with respect to a suitable reference electrode, and is opposite in sign to the normal definition of the potential of the metal with respect to the reference electrode. The calculation of the limiting current densities and the equilibrium potentials for the redox reactions and the data employed in the calculations are described in Macdonald and Urquidi-Macdonald [7,9–12].

The potential distribution displayed in Figure 12a predicts that the variation of $\phi_s - \phi_s^c$ with distance is strongly dependent upon the oxygen concentration, such that the decay with distance away from the crack mouth becomes less steep as [O$_2$] increases. This is a direct result of more oxygen being available at the surface for reduction as the bulk [O$_2$] increases, so that a lower oxygen electrode reaction overpotential is required to consume the current from the crack. Upon decreasing the conductivity or stress intensity factor, the distribution is found to flatten for similar reasons [10,11].

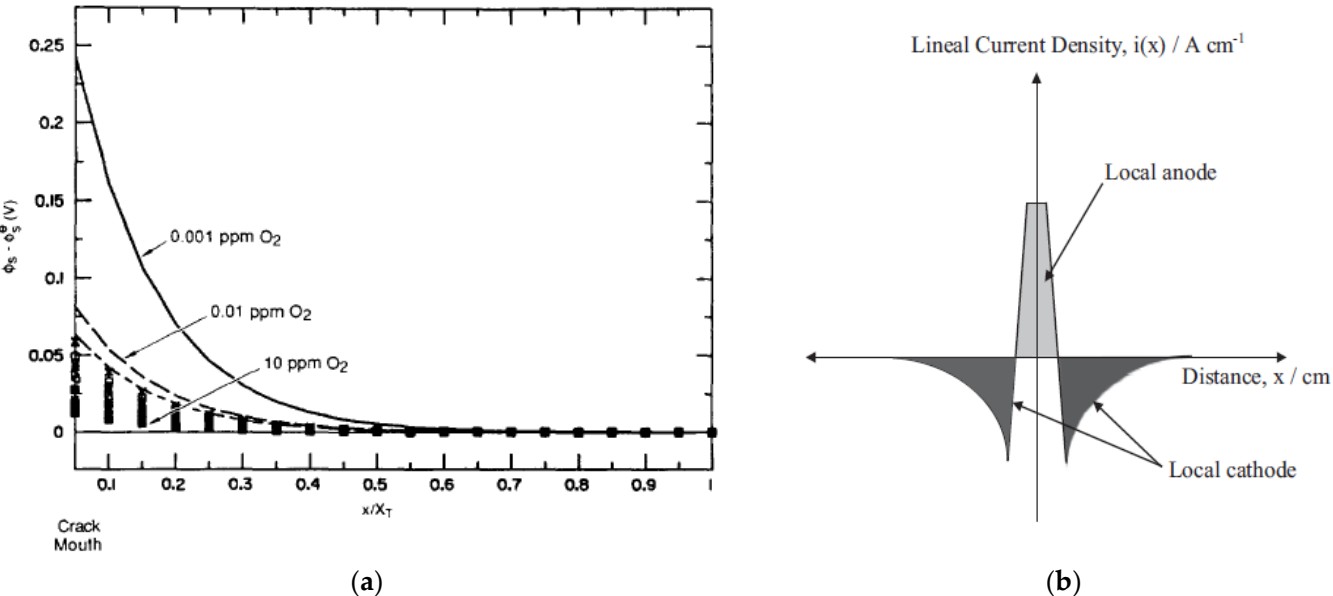

**(a)** **(b)**

**Figure 12.** (**a**) Calculated distribution of the potential in the solution on the external surface as a function of distance from the crack mouth as a function of the concentration of oxygen in the solution at 288 °C. Other parameter values employed in the calculation are given by Macdonald and Urquidi-Macdonald [7]. (**b**) Schematic of the current density across the crack at charge conservation (dark and light shaded area are equal). After [10,11].

The total current on the external surface is then given by

$$4\int_{w/2}^{Z_T}\int_{\delta}^{X_T}\frac{e^{-[\phi_s-\phi_s^e]/b_a}-e^{[\phi_s-\phi_s^e]/b_c}}{1/i_0+\left(1/i_{l,f}\right)e^{-[\phi_s-\phi_s^e]/b_a}-\left(1/i_{l,r}\right)e^{[\phi_s-\phi_s^e]/b_c}}dxdz+Ai_{ss}=I_E \qquad (11)$$

where $A$ is the area of the external surface and $i_{ss}$ is the current density for the electrodissolution of the steel. For stainless steels, $i_{ss}$ is independent of potential and hence is a constant [34,66,67].

Returning now to Figure 2, it is evident that an infinite number of solutions for the crack internal current exist depending upon the value chosen for the potential $\phi_s(0,0)$ at the crack mouth. Likewise, an infinite number of solutions exist for the current and potential distributions in the external environment, depending on the value chosen for that same potential. However, only one value for $\phi_s(0,0)$ exists for charge conservation in the system. Thus, the execution of the CEFM involves iterating on $\phi_s(0,0)$ until charge is conserved in the system [7,9–12]; i.e.,

$$\int i_N dS = 0 \qquad (12)$$

where the integration is carried out over the entire surface (including that within the crack), and $i_N$ is the net, local current density on the increment in area of $dS$. For the model developed in the early 1990s, that condition is expressed by $I_I = I_E$.

The algorithm used to obtain the value of $\phi_s(0,0)$ for $I_I = I_E$ involves two embedded loops, as outlined in Figure 13. The principal iteration is performed on $\phi_s(0,0)$ by first assuming $\phi_s(0,0) = E_{mouth}$ and then calculating the currents $I_I$ and $I_E$. If these are found not to be equal within a preselected difference, the potential $\phi_s(0,0)$ is changed and the cycle is repeated; within this loop is an embedded loop that is designed to converge on the crack tip potential, $\phi_{s,tip}$, using electro-neutrality at the crack tip as the physical condition that must be satisfied. The embedded loop converges for each cycle of the principal loop.

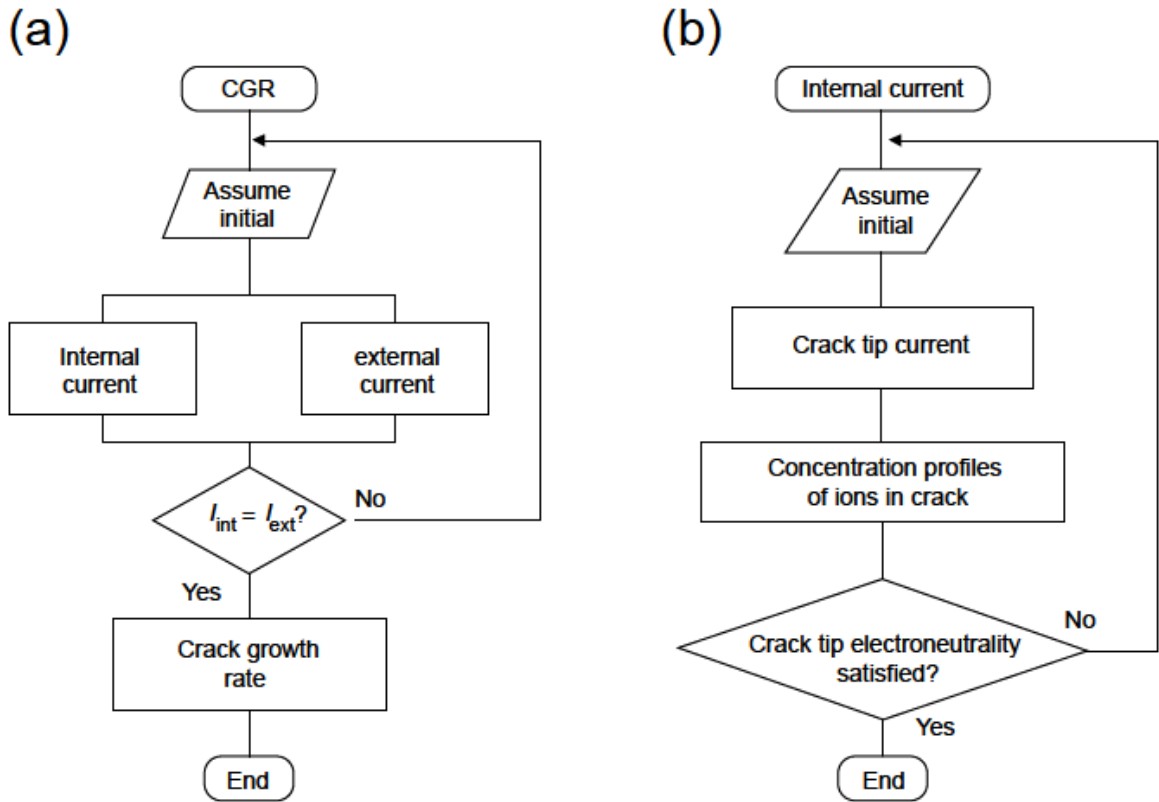

**Figure 13.** Flow diagram for the algorithm employed to calculate the potential at the crack mouth (**a**) and crack tip (**b**) such that charge is conserved in the system. Based on [7,9–12].

It is important to note that Condition (12) follows from the fundamental laws of chemistry, the conservation of charge, and mass-charge equivalency (Faraday's law) and that these laws confer determinism on the CEFM, yielding a solution for $\phi_s(0,0)$ that is "physically real". However, the reader will note that the CEFM, like most complex physico-electrochemical models contains parameters that cannot be determined by independent experiment. For example, the potential at the crack tip ($\phi_{s,tip}$) is related to the metal dissolution current ($\overline{I_0}$) by the expression [7,9–12]

$$\phi_{s,tip} = \phi_s^0 + \hat{b}_a ln\left\{ \frac{2i_0^0 A_t^0 (t_0/t_f)^{1/2}}{\overline{I_0}} \right\} \tag{13}$$

where $\hat{b}_a$ is the Tafel constant for metal electrodissolution, $\phi_s^0$ is a constant, $t_f = \frac{1}{MFF} = \varepsilon_f/\dot{\varepsilon}$ is the period of the cyclical fracture of the matrix at the crack tip, $\varepsilon_f$ is the fracture stain at the crack tip, $\dot{\varepsilon}$ is the crack tip strain rate that is calculated by various fracture mechanics models that are available in the literature [37,60–62,68,69] and are summarized in Table 4, $t_0$ is a constant, $A_t^0$ is the area of the fracture at the moment that it occurs, and $i_0^0$ is the standard exchange current density for metal electrodissolution at the crack tip at a moment just after the occurrence of fracture. All these parameters are poorly known, if known at all, so that it is necessary to calibrate the model on available data, with one calibration point being shown in Figure 4. However, while calibration is necessary to calculate the CGR under a given set of conditions, once that calibration has been performed the CEFM is found to accurately predict CGR over wide ranges of the independent variables, as discussed below.

**Table 4.** Expressions for the crack tip strain rate for estimating CGR in sensitized Type 304 SS.

| Model | Equation |
|---|---|
| Ford [37] | $\dot{\varepsilon}_{ct} = 4.11 \times 10^{-11} K_I^4$ |
| Congelton [68] | $\dot{\varepsilon}_{ct} = \frac{\dot{a}}{r} \left[ 63.65 \pi \alpha \frac{(1-v^2)}{\sigma_{yE}} + \beta \left( \frac{\sigma_y}{E} \right) ln \left( \frac{R_p}{r} \right) \right]$ |
| Shoji [61–63] | $\dot{\varepsilon}_{ct} = \beta_1 \left( \frac{\sigma_y}{E} \right) \left( \frac{n_{HG}}{n_{HG}-1} \right) \left\{ ln \left[ \left( \frac{\lambda}{r} \right) \left( \frac{K_I}{\sigma_y} \right)^2 \right] \right\}^{\frac{1}{n_{CH}-1}} \left( \frac{\dot{a}}{r} \right)$ |
| Hall [69] | $\dot{\varepsilon}_{ct} = \frac{n_R}{n_R+1} \frac{\sigma_y}{E} \left( \frac{\alpha_{ct} E J}{r \sigma_y^2} \right)^{\frac{n}{n+1}} \frac{\dot{J}}{J} + \frac{n_R}{n_R-1} \beta_{ct} \frac{\sigma_y}{E} \frac{\dot{a}}{r} \left\{ ln \left[ \left( \frac{\lambda}{r} \right) \left( \frac{K}{\sigma_y} \right)^2 \right] \right\}^{\frac{n_R+1}{n_R-1}}$ |
| Temperature Dependence | $\dot{\varepsilon}_{ct}(T) = \dot{\varepsilon}_{ct}(288\ ^{\circ}C) exp \left[ \frac{Q}{R} \left( \frac{1}{T} - \frac{1}{561.15} \right) \right]$ |

$\dot{\varepsilon}_{ct}$ = crack tip strain rate; $\alpha, \beta, \beta_1, \lambda$ = dimensionless constants in plastic strain calculation; $r$ distance from growing crack tip; $R_p$ = plastic zone size; $\sigma_y$ = yield strength; $E$ = Elastic (Young's) modulus; $n_{HG}$ = strain hardening exponent of Hwang and Gao ((see Refs. [61–63]); $Q$ = activation energy; $K_I$ = stress intensity factor; $\dot{a}$ = crack growth rate; $n_R$ = strain hardening exponent of Ramberg and Osgood (also see [61–63]); $J$ is the J-integral; $\dot{J}$ is the rate of change in $J$ with time.

### 5.3. Creep Crack Growth Rate

Experiment shows that, at a sufficiently negative ECP ($<-0.23$ V$_{she}$, Figure 4), the CGR becomes independent of the electrochemical properties of the system. Under these conditions, the crack propagates mechanically and the fracture morphology changes from IGSCC for ECP > $-0.23$ V$_{she}$ to a ductile fracture with evidence of micro-void nucleation and coalescence. The model adopted in the CEFM to describe this limit is the micro-void coalescence model of Wilkinson and Vitek [70], as shown in Figure 14. Briefly, the model proposes that voids nucleate at fixed distances ahead of the crack tip. Because of the local stress field the void nearest the crack tip grows the fastest, and when it reaches a critical size the ligament between it and the crack tip fractures, resulting in crack advance by a distance c, and the process repeats. The frequency of these events can be calculated, resulting in a creep crack growth rate (CCGR) of

$$\frac{dL}{dt} = fc \tag{14}$$

Further details can be found in Ref. [70].

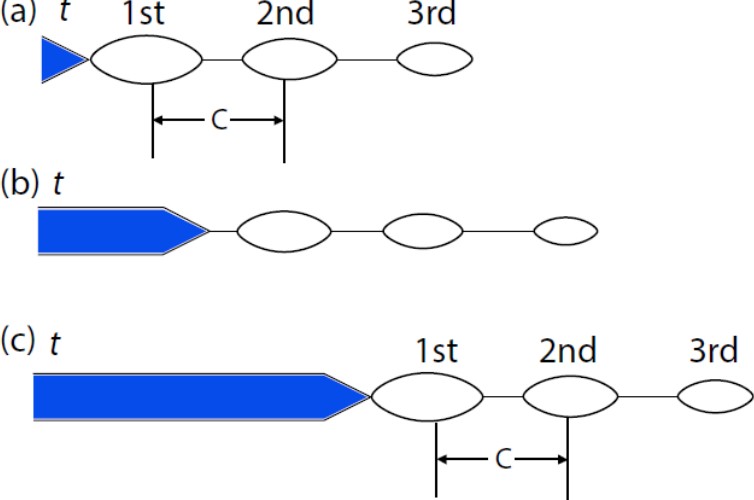

**Figure 14.** (**a–c**) Creep crack growth model of Wilkinson and Vitek illustrating crack advance via the nucleation and linking of voids ahead of the crack tip. Adapted from [70].

The impacts of temperature and stress intensity factor ($K_I$) on the CCGR according to the model of Wilkinson and Vitek [70] are displayed in Figure 15. Figure 15a displays the Arrhenius plot of log(CCGR) vs. reciprocal Kelvin temperature at a stress intensity factor of 27.5 MPa·m$^{1/2}$. Note that, below a CCGR of about $1 \times 10^{-11}$ cm/s, the CCGR is essentially unmeasurable using current techniques. Thus, for $T = 250$ °C, the CCGR is so low that it has no impact on the accumulation of cracking damage in Type 304 SS in reactor coolant circuits, because this value corresponds to a crack extension of 3.14 μm/a. Figure 15b displays the dependence of log(CCGR) on the stress intensity factor at 288 °C. The plot displays the expected dependence, with log (CCGR) increasing monotonically with $K_I$. The data are well-represented by log(CCGR) = $2 \times 10^{-13}$ $K_I{}^{2.0}$.

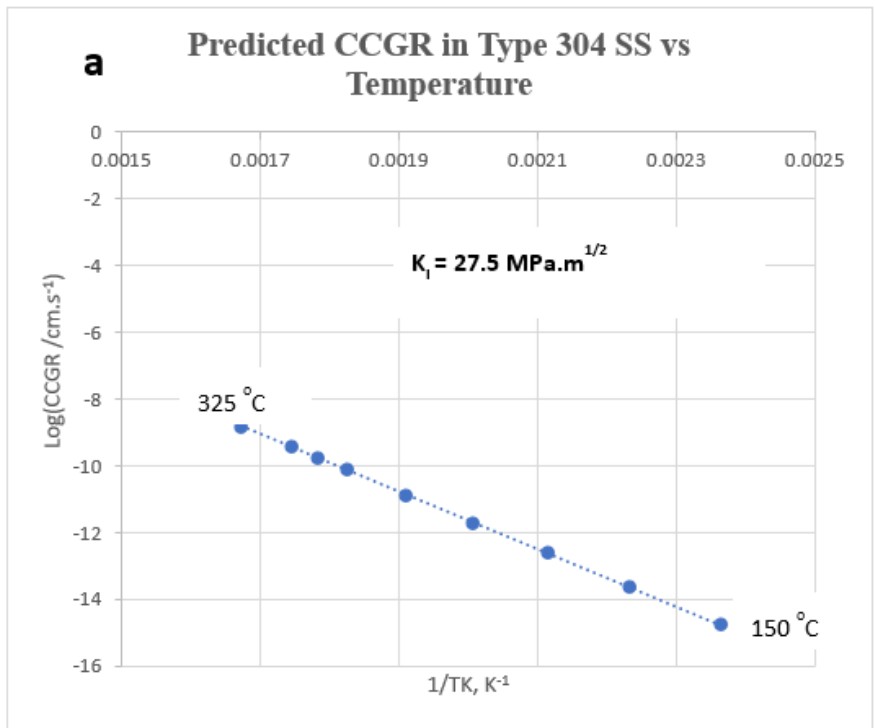

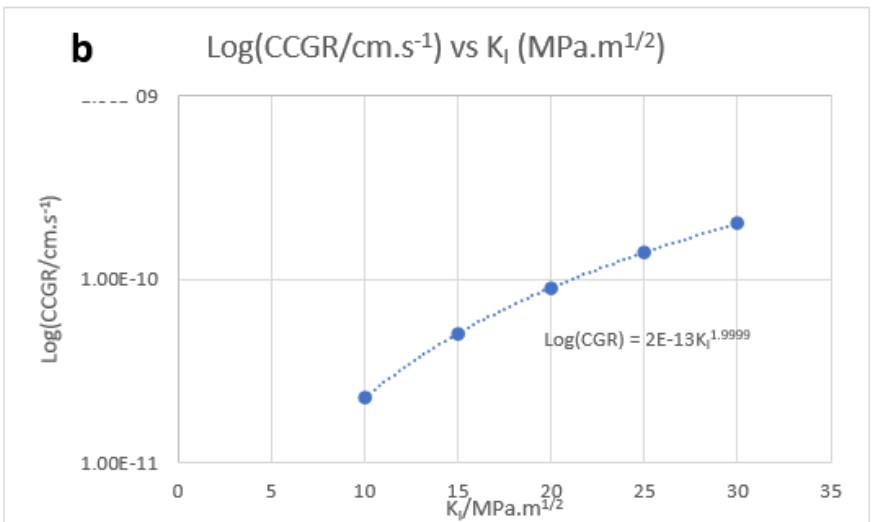

**Figure 15.** Plots of log(CCGR) vs. $1/T$ (**a**) and vs. $K_I$ (**b**) for Type 304 SS as calculated by the Wilkinson and Vitek model [70], using the Creep Crack Growth Rate algorithm.

It is important to note that very few data could be found in the literature to compare with the values calculated from the Wilkinson and Vitek [70] model using the Creep Crack Growth Rate module contained in the CEFM. While many data are available for fatigue loading on mechanical fracture, the translation of these data to the constant loading case has not been developed to the extent that it is readily possible to predict the CGR under constant load from the fatigue CGR, and vice versa. Accordingly, the data plotted in Figure 15 must be regarded as being unsubstantiated model calculations.

*5.4. CEFM Algorithm*

Although an analytical form of the CEFM (ACEFM) has been developed [64,65], it has not been largely used, primarily because its predictions, after calibration on the same data as the numerical CEFM, differ imperceptibly from those of the latter. A typical calculation of CGR vs. ECP is presented in Figure 16. Comparison with Figure 4 reveals negligible differences. One important conceptual difference between the CEFM and ACEFM is that the latter employs a trapezoidal crevice rather than a parallel-sided geometrical model of the crack. For large aspect ratios (AR > 10–20) the impact of crevice geometry is immaterial from a computational viewpoint.

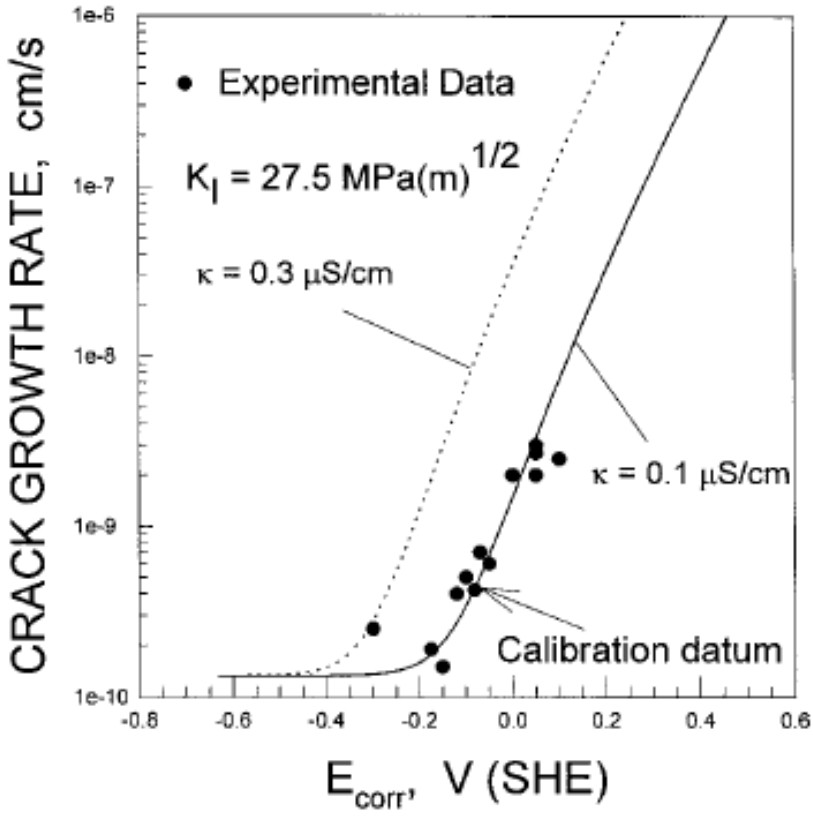

**Figure 16.** Calculated CGR vs. ECP for IGSCC in sensitized Type 304 SS in water at 288 °C using the analytical CEFM [64,65].

## 6. The Predictions of the CEFM

The objective of any model is to predict the dependent variable (CGR in the CEFM) under extended ranges of the independent variables, and some of these predictions of the CEFM have been discussed above. Below, some of the more important predictions of the CEFM are outlined together with brief discussions of the impact that the prediction has on the field of stress corrosion cracking. Recall that the metric of any model is its ability to predict the response of the dependent variable to values of the independent variables that lie outside of any calibrating range and, especially, its ability to predict new phenomena. In

this regard, the CEFM has only ever been calibrated on two CGR vs. ECP data at different temperatures to obtain the activation energy for the crack tip strain rate, as noted above.

### 6.1. External Polarization

Previous models that were developed to account for SCC generally assume (implicitly or explicitly) that the electrostatic potential in the solution at the crack mouth is a constant and is equal to the negative of the free corrosion potential, such that it is independent of the CGR (e.g., [71]). If so, then no potential gradient exists in the external environment, and hence the external coupling current, via Ohm's law, must be zero. Since this current must be equal and opposite in sign to the coupling current (CC) that flows through the metal (Figure 2), the CC must also be zero and should not have been detected in the experiments of Manahan et al. [45]. Furthermore, a zero CC requires that the dissolution rate at the crack tip must also be zero, and hence by Faraday's law the electrochemical CGR must be zero. In other words, these models predict that SCC cannot occur!

Using the CEFM, it is possible to calculate the polarizations within the crack enclave and in the external environment as a function of the various independent variables, and such a calculation as a function of temperature is presented in Figure 17 [13]. At ambient temperature (25 °C), and for the values selected for the independent variables [13], the polarization in the external environment is predicted to be 170 mV while that within the crack enclave is 280 mV. As the temperature increases the polarization across the external surface decreases, while that within the crack enclave increases, such that at 300 °C they are about 2 mV and 420 mV, respectively. Indeed, the sum of these polarizations, which represents the electrochemical driving force for crack propagation, passes through a small maximum between 200 and 250 °C. The small polarization across the external surface at 300 °C would, seemingly, justify the assumption that the potential in the solution at the crack mouth is equal to −ECP [72]. However, the polarization is still finite and sufficient that the redox reactions that occur on the external surface can annihilate the electronic CC flowing through the metal from the crack tip. Thus, the fundamental reason for the failure of these previous models is not so much due to their arbitrary assumption of a potential at the crack mouth as it is in ignoring the role played by the redox reactions on the external surface that precludes determinism.

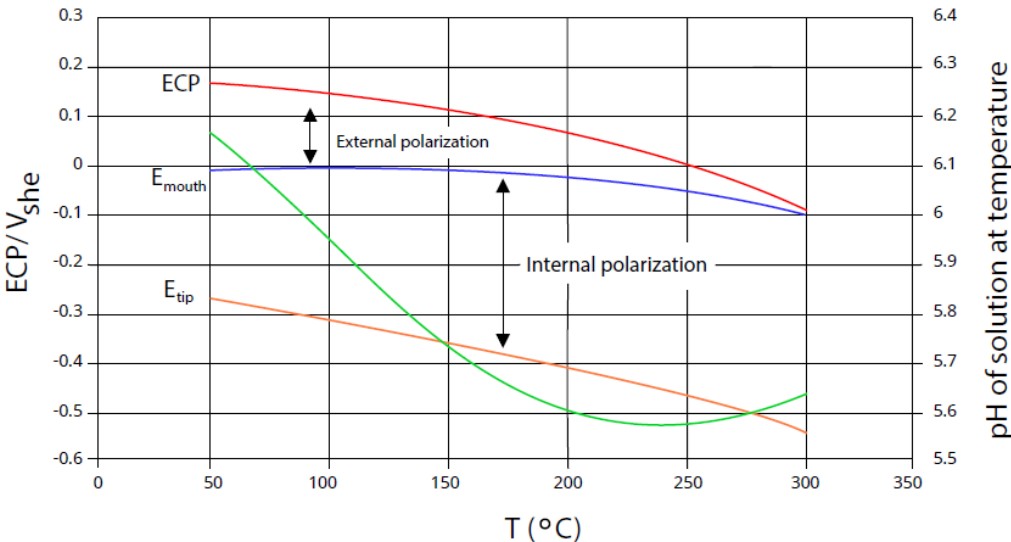

**Figure 17.** Effect of temperature and pH on the electrostatic potential at the crack tip, crack mouth, and at an effectively infinite distance on the external surface away from the crack mouth (-ECP), together with the variation of the pH for IGSCC in sensitized Type 304 SS in dilute $H_2SO_4$ solution having an ambient temperature conductivity of 0.27 μS/cm and a dissolved oxygen concentration of 200 ppb. Adapted from [13].

*6.2. Role of the Reactions on the External Surface*

Occasionally, anecdotal reports are made that the state of the external surface (e.g., the composition and nature of the passive film) appears to affect the CGR. For example, more than 30 years ago the author observed that many chromium-containing steels under ostensibly identical conditions display similar CGRs, even though their mechanical and metallurgical parameters are significantly different. Recently, Que [72] reported that the initiation and propagation of IGSCC in Alloy 182 is sensitive to the finishing of the external surface, but they made no attempt to relate the differences to the electrochemistry of the system, even though the need to do so had been established decades earlier. After much analysis of the possible factors involved, it became apparent that the important parameters are the values of the exchange current densities and Tafel constants of the redox reactions that occur on the external surface, because they determine the ability of the external surface to annihilate the electronic CC, which is linearly related to the CGR. Figure 18 plots the CGR vs. $[O_2]$ for different values of the standard exchange current density multiplier (SECDM) for the OER, HER, and HPER, in a scenario described as general catalysis/inhibition. SECDM = 1 corresponds to uncatalyzed stainless steel, SECDM > 1 describes general catalysis, and SECDM < 1 corresponds to general inhibition. For SECDM = $10^{-4}$, the crack growth is predicted to be completely inhibited and the CGR is calculated to be reduced to the creep crack growth rate (CCGR) limit of $1.69 \times 10^{-10}$ cm/s. For SECDM = $10^{-2}$ complete inhibition is predicted except for $[O_2] > 1$ ppm, but for SECDM = 1 and $10^2$ the CGR is predicted to increase (general catalysis) by one and two orders in magnitude, respectively, for low $[O_2]$ and by more than an order of magnitude at high $[O_2]$. Thus, we conclude that the kinetics of the redox reactions on the external surfaces exert powerful influences on the CGR, and it follows the original dictum for localized corrosion be the case of a "large cathode driving a small anode" [28]. Thus, the important finding is that, under normal operating conditions in a BWR primary coolant circuit, the CGR is controlled by the redox reactions on the external surface. Therefore, previous models that ignored the role played by the external surface arguably missed the most important aspect in CGR control.

The experimental evidence for these predictions comes in two forms. First, recall that the CC and the CGR are linearly related by Faraday's law so that any change in the CC must be reflected in a proportionate change in the CGR. Thus, comparing Figures 8 and 9 shows that, for the same $K_I$ value (33 MPa·m$^{1/2}$), the CC from the Pt-catalyzed Ni cathodes (Figure 8) is 500 μA compared with 10 μA for the uncatalyzed cathodes, a difference of a factor of 50. Accordingly, the CGR should be a factor of 50 higher in the former compared with that in the latter. Although we do not know the standard exchange current densities, these results are generally in agreement with the predictions of Figure 18.

The second set of evidence comes from the work of Zhou et al. [57]. Their experiment daisy-chained two C(T) fracture mechanics specimens together in an autoclave in a recirculating flow loop operating at 250 °C, $K_I$ = 25 MPa·m$^{1/2}$, with dilute $Na_2SO_4$ solution (conductivity at 25 °C of 220 μS/cm, and $[O_2]$ = 40 ppm (pure $O_2$). One specimen had been coated with an electrophoretically deposited $ZrO_2$ layer, which was then cured at 250 °C for 48 h resulting in an impervious coating that was about 150 μm thick. The crack opening displacement (COD) of each specimen was monitored over 400 h and the COD was converted into crack length using standard compliance methods, while simultaneously monitoring of the ECP using an Ag/AgCl external pressure-balanced reference electrode. The specific impedance of the coating was measured at ambient temperature using a fast $Fe(CN)_6^{3/4-}$ redox couple to monitor the coating integrity before and after the experiment [57].

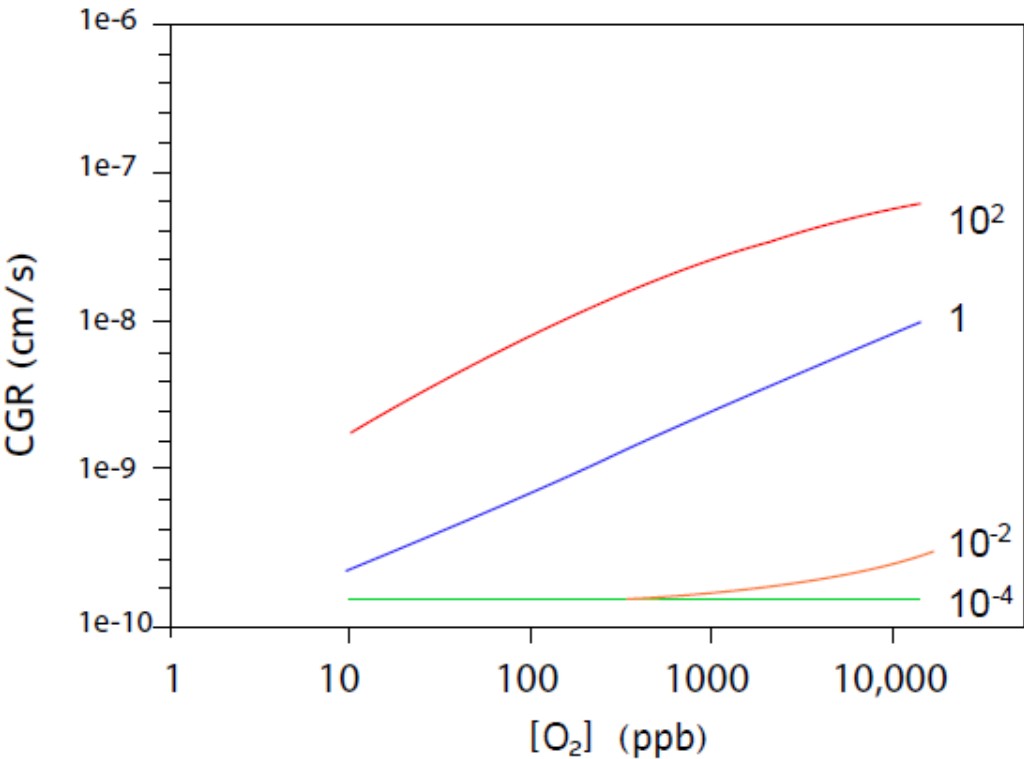

**Figure 18.** Impact of exchange current density multiplier on the IGSCC in sensitized Type 304 SS in water at 288 °C as a function of oxygen concentration. The number next to each curve is the multiplier of the standard exchange current density (MSECD). Parameter values are given by Lu et al. [13]. Adapted from [13].

The experiment shows (Figure 19a) that the crack grows in the uncoated specimen but not in the coated specimen, because the coating prevents the occurrence of the redox reactions on the external surface, thereby reducing the CC, and hence the CGR, to zero. The impedance analysis shows that after the exposure of the coated sample for 400 h, the specific impedance of the coated specimen was a factor of 100 to 1000 higher than that of the uncoated specimen. Because the exchange current density of a redox reaction $(Fe(CN)_6^{3/4-})$ varies inversely with the specific impedance, we estimated that the exchange current density of the oxygen electrode reaction in the actual experiment (Figure 19a) was reduced by a factor of $10^2$ to $10^3$ by the coating. Accordingly, as calculated from the CEFM, the ECP should have been reduced from about 100 mV$_{she}$ to $-200$ to $-400$ mV$_{she}$ and the crack growth rate should have been reduced by factor 100 to 1000. The CGR calculated from the uncoated specimen (Figure 19a) was $3.4 \times 10^{-7}$ cm/s, which reflects the high $[O_2]$ and conductivity used in the experiment. Thus, the CGR of the coated specimen should have been reduced to between $3.4 \times 10^{-9}$ and $3.4 \times 10^{-10}$. Since the sensitivity of the experiment to CGR does not allow CGR to be determined about $2 \times 10^{-8}$ cm/s, the finding that the observed crack growth rate is below this limit was consistent with the calculation. Regarding the ECP, the observed value for the uncoated specimen was 200 mV$_{she}$, while that of the coated specimen was $-200 \pm 100$ mV$_{she}$; again, in reasonable agreement with calculation (Figure 19b).

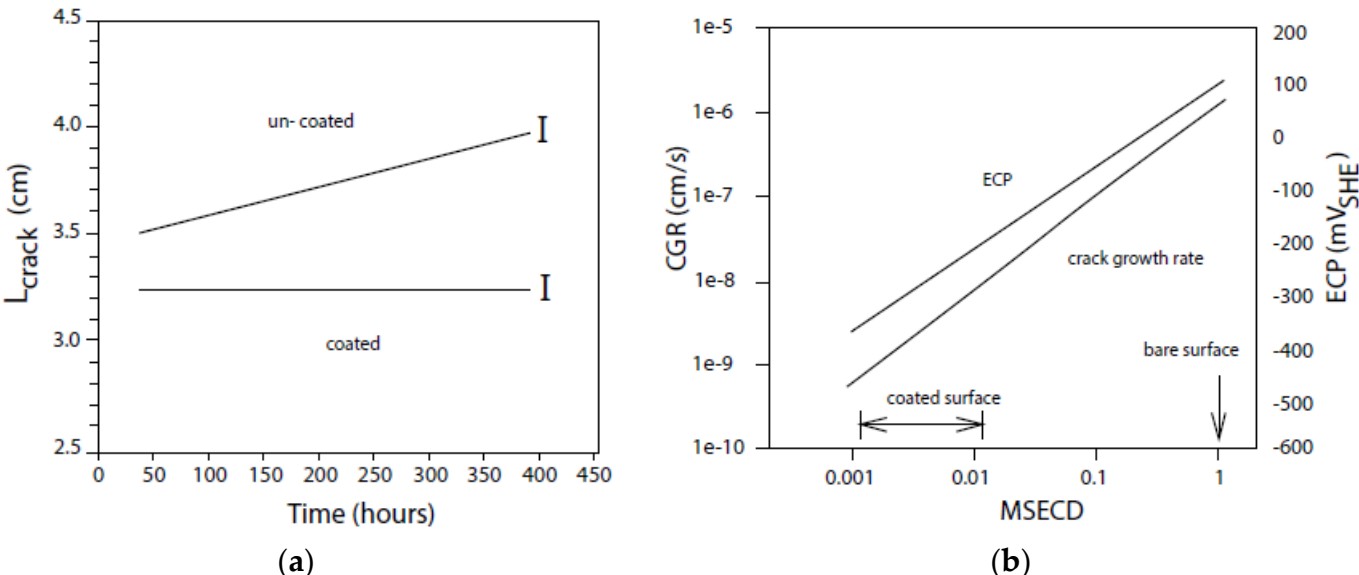

**Figure 19.** (**a**) Crack length vs. time for the ZrO$_2$-coated and uncoated specimens. (**b**) CGR and ECP as a function of the SECDM. IGSCC in sensitized Type 304 SS in dilute Na$_2$SO$_4$ solution at 250 °C (conductivity at 25 °C of 220 μS/cm, and [O$_2$] = 40 ppm (pure O$_2$)). Adapted from [57].

### 6.3. Crack Growth Rate/Coupling Current Relationship

As stated above, the CEFM predicts a linear relationship between the CGR and the CC. Since both may be measured in the same experiment (Figures 8 and 9), this relationship can be tested experimentally. This relationship has been explored by Wuensche and Macdonald [56], and their findings are given in Figure 20. Although the data are meager, because of the difficulty of performing this experiment, the linear relationship is apparent and in accordance with Equation (15).

$$\frac{dL}{dt} = \frac{\overline{M}}{\overline{z}\rho FA} I \tag{15}$$

where $L$ is the crack length, $\overline{M}$ is the composition averaged atomic weight of the steel, $\overline{z}$ is the composition averaged electron number, $\rho$ is the steel density, and $A$ is the dissolution area at the crack tip. Noting that $\frac{\overline{M}}{\overline{z}\rho F}$ is about $3.7 \times 10^{-5}$ cm$^3$/C and that the slope of the line is 0.17, we calculated the crack tip area as $2 \times 10^{-4}$ cm$^2$. While we do not have an independent measurement of the crack tip area, the value obtained from Figure 20 is not unrealistic. For example, from Table 2, the crack tip area is assumed arbitrarily to be $10^{-3}$ cm$^2$, which is within a factor of 5 given by the CGR/CC analysis presented above. Given that the COD and crack width parameters given in Table 2 were arbitrarily chosen, the level of agreement must be judged to be acceptable. However, some equivocation exists regarding how the crack tip area is defined, since it appears that crack advance is not entirely due to electrodissolution at the crack tip, but that hydrogen-induced fracture (HIC) plays a significant role in determining the microfracture dimension (MFD) and, hence, also the CGR. We do not currently have data to resolve this issue.

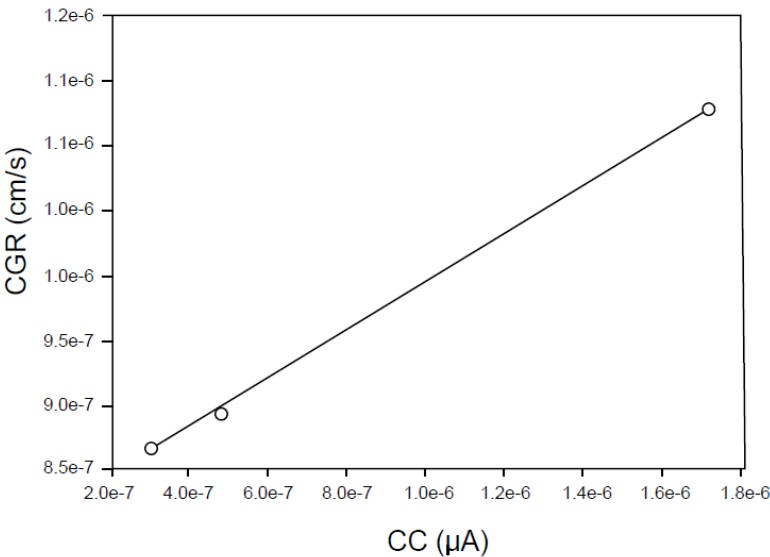

**Figure 20.** Relationship between CGR and CC for IGSCC in sensitized Type 304 SS in dilute NaCl solution (50 ppm), [$O_2$] = 7.6 ppm at 250 °C. From [56].

*6.4. Crack Tip Conditions*

The solution of the Nernst–Planck equations for the species in the crack [$H^+$, $OH^-$, $Na^+$, $Cl^-$, $Fe^{2+}$, $Ni^{2+}$, $Cr^{3+}$, $Fe(OH)^+$, $Ni(OH)^+$, $Cr(OH)^{2+}$] allows the concentrations of these species to be calculated at any position within the crack as a function of the various independent variables [10–12,64,70]. To illustrate these calculations, the examples given are restricted to pH ([$H^+$]), [$Na^+$], and [$Cl^-$] at the crack tip as a function of the ECP, which controls the CGR (see Figure 4), for two different bulk environment conductivities as set by the bulk [$Na^+$]$_b$ and [$Cl^-$]$_b$. Thus, Figure 21 shows that $Na^+$ is rejected from the crack, while $Cl^-$ concentrates within the crack enclave as the ECP is made more positive, and hence as both the CC and CGR increase.

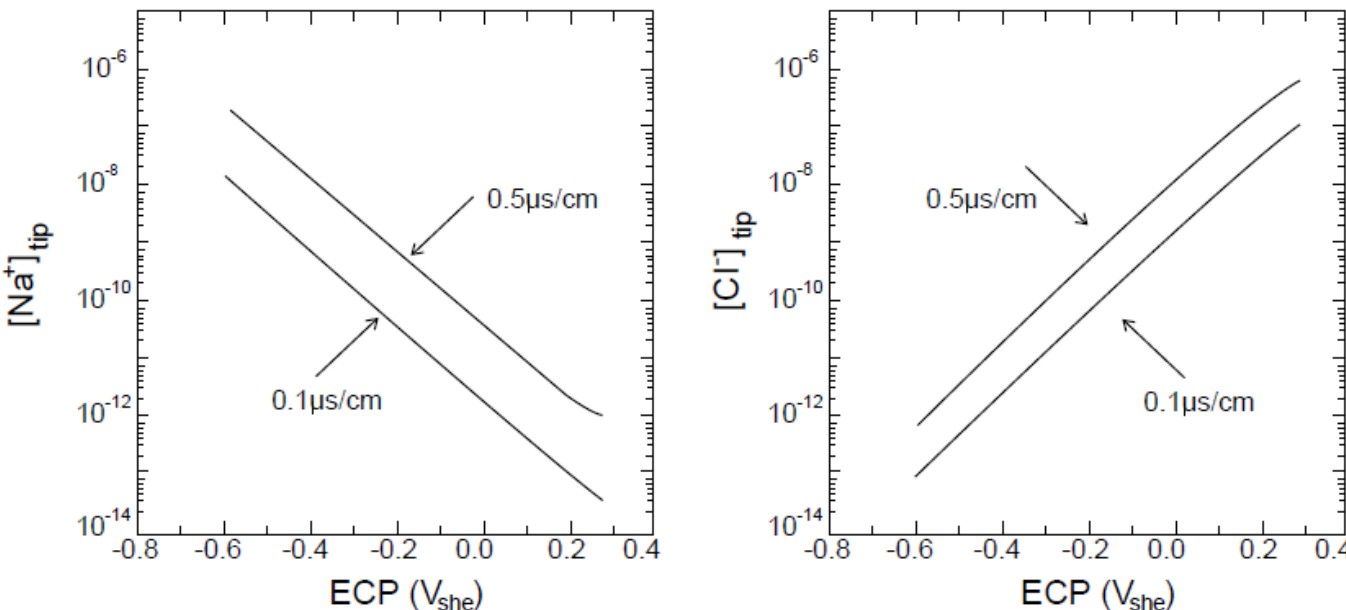

**Figure 21.** Calculated [$Na^+$], and [$Cl^-$] at the crack tip for a standard crack (Table 3) in sensitized Type 304 SS in dilute NaCl solution as a function of ECP for different values of ambient temperature conductivity. Adapted from [12].

Likewise, the pH at the crack tip is predicted to strongly decrease ([H$^+$] increase) with increasing ECP (Figure 22). These are the expected behaviors corresponding to the migration of positively charged ions (Na$^+$) down the potential gradient and negatively charged ions (Cl$^-$, OH$^-$) up the potential gradient from the crack tip to the crack mouth, resulting in the flow of positive current out of the crack through the solution to the external surface. In the case of H$^+$, protons are produced by the hydrolysis of metal cations at the crack tip at a sufficient rate to overcome the migration out of the crack, so that the pH at the crack tip remains low.

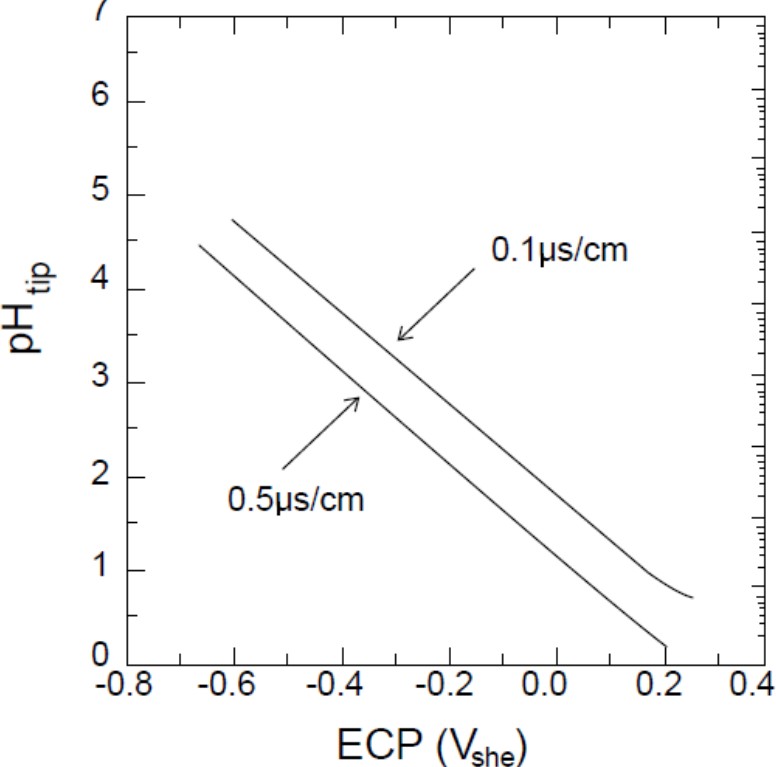

**Figure 22.** Calculated pH at the crack tip for a standard crack (Table 3) in sensitized Type 304 SS in dilute NaCl solution as a function of ECP for different values of ambient temperature conductivity. Adapted from [12].

From Figures 21 and 22, we see that for given ECP the crack tip [H$^+$] and [Na$^+$] increase and the crack tip [Cl$^-$] decreases as the ambient temperature conductivity increases. This is due to the more intense migration within the crack reflecting a higher CC, and hence CGR.

*6.5. Effect of Crack Length*

Perhaps the most important prediction of the CEFM is the dependence of the CGR on the crack length over and above that which sets the stress intensity factor under constant load. To the author's knowledge, this dependence had not been previously reported when the prediction was first made (in 1995, [12]). An example of such a prediction for constant $K_I$ is shown in Figure 23 where the CGR is plotted vs. the ECP for crack lengths varying from 10 μm to 10 cm. It is seen that the crack growth rate varies over almost three orders of magnitude for the assumed range in crack length and is reduced with increasing crack length.

It is important to recognized that the crack length referred to here is the shortest path taken by the current flowing down the crack to the nearest external surface, and is termed the "electrochemical crack length" (ECL). This is generally different from the mechanical crack length (MCL) that, together with the load, sets the stress intensity factor,

as is illustrated in Figure 24 for a C(T) fracture mechanics specimen. As seen, the ECL, which is the shortest path from the crack front to the external surface, is quite different to the MCL. It is clear, therefore, that in describing SCC it is necessary to specify two crack lengths: the MCL that establishes $K_I$, and the ECL that determines the electrochemical activity. As we will see below, the dependence of CGR on the ECL has a profound impact on the accumulation of corrosion damage due to SCC in practical systems.

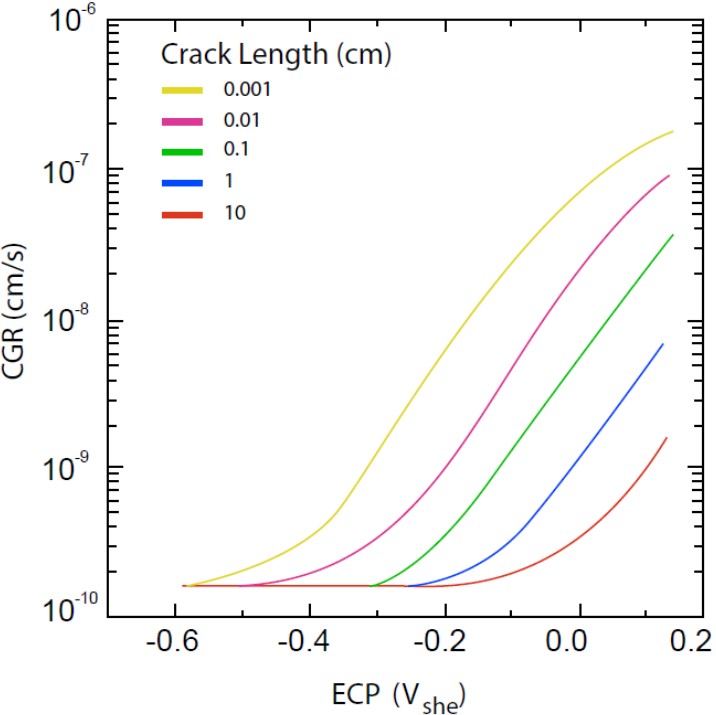

**Figure 23.** Dependence of CGR in sensitized Type 304 SS in water at 288 °C as a function of ECL. $K_I = 27.5$ MPa·m$^{1/2}$. Adapted from [12].

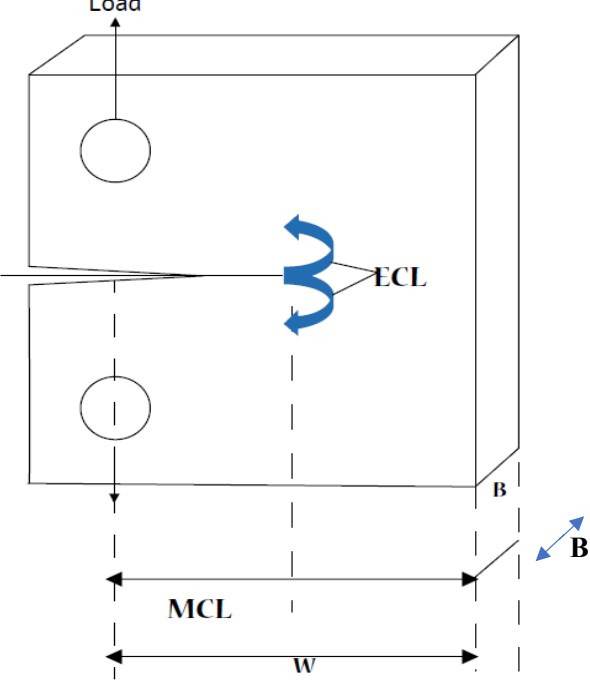

**Figure 24.** Illustration of the difference between the ECL and MCL.

As noted previously (Section 5.3), the CGR and the CC are related by Faraday's law and a plot of calculated CC vs. CGR is displayed in Figure 20. It is evident that the relationship follows two arms; the SCC arm for CC > $10^{-9}$ µA, and the CCGR (creep crack growth rate) arm in which the CCGR is independent of the CC. A critical CC of $10^{-9}$ µA exists below which crack advance is entirely due to creep. Elsewhere in this paper, I propose that the critical CC is more fundamental than the ECP in determining the conditions at which the crack transitions from propagating mechanically (via creep) to propagating by SCC. It is also important to note that the CC is finite within the CCGR arm, so that SCC still occurs but is "outrun" by creep, which controls the observed CGR. The existence of a CC for ECP < $-0.23$ $V_{she}$, the potential at which the CGR becomes dominated by SCC, also accounts for the variations of the crack tip [$Na^+$], [$Cl^-$], and pH shown in Figures 21 and 22 under conditions for which CCGR dominates.

From Figure 25, it is noted that the CC/CGR data for all ECL fall on a common locus, with the distribution shifting to the left (reduced CGR) as ECL increases, which is consistent with the data plotted in Figure 23 (CGR vs. ECP for different ECL). This is, of course, expected, because the CGR and the CC are related by a law (Faraday's law) that is not a function of the independent variables of the system.

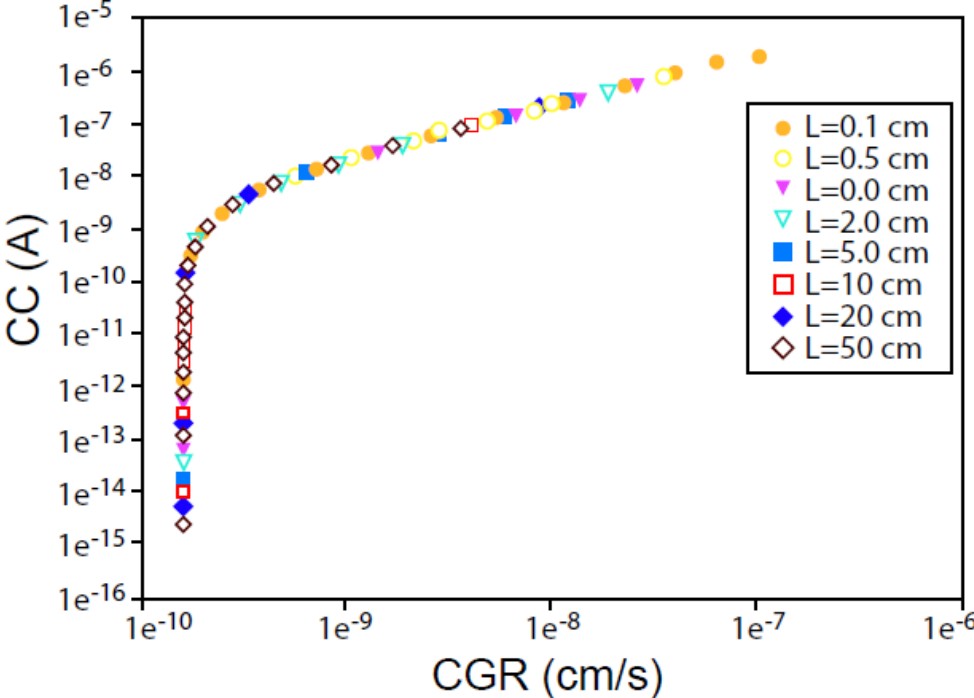

**Figure 25.** Predicted coupling current vs. crack growth rate for the growth of a standard crack (Table 3) for IGSCC in sensitized Type 304 SS in dilute $H_2SO_4$/NaCl solution at 288 °C for different values of ECL. COD = $5 \times 10^{-4}$ cm, crack width = 1.0 cm, $K_I$ = 27.5 MPa.m$^{1/2}$, $V_f$ = 100 cm/s, $d$ = 50 cm, $T$ = 288 °C, [$H_2$] = $10^{-4}$ ppb, [$H_2O_2$] = $10^{-4}$ ppb, [$Na^+$] = 1.35 ppb, [$H_2SO_4$] $10^{-6}$ ppbS, [$O_2$] = 1 ppb − $2.33 \times 10^6$ ppb, $\kappa_{288}$ = 2.69 uS/cm, $\kappa_{25}$ = 0.0618 µS/cm, creep crack growth rate = $1.61 \times 10^{-10}$ cm/s. Adapted from [12].

The dependence of CGR on ECL has important consequences for the accumulation of SCC damage in reactor coolant circuits, as shown in Figure 26 [73]. These plots of crack length vs. time, which are based upon the operating history of a BWR in Taiwan, show that as time proceeds the CGR (the gradients of the curves in the figure) decreases, with the result that the accumulated damage is much less than would be the case if no dependence of CGR on ECL was recognized (as indicated by the broken line). The calculation assumed an initial crack of 0.5 cm in length and the CEFM predicted an initial CGR of about $3 \times 10^{-9}$ cm/s under the prevailing conditions. The figure also demonstrates an important

issue that arises in the adoption of hydrogen water chemistry (HWC); in this case via the addition of 1 ppm of $H_2$ to the reactor feedwater. Thus, if no HWC is adopted, the crack is predicted to grow by 2.2 cm over the ten-year operating period. If HWC is adopted immediately, the crack is predicted to grow by only 0.6 cm. However, if HWC is adopted after 5 years of operation, the crack will be 1.7 cm long: only 0.5 cm less than that under normal water chemistry (NWC). This clearly follows the law of decreasing returns, in that the longer one waits to introduce HWC the lower the benefit will be. Since the installation of HWC is expensive, because of the need to store large amounts of flammable hydrogen on-site and the need to strengthen and shield the turbine hall to from irradiation from $^{16}N_7$ and $^{17}N_7$ in the form of volatile ammonia, a cost/benefit analysis of the type presented above is necessary.

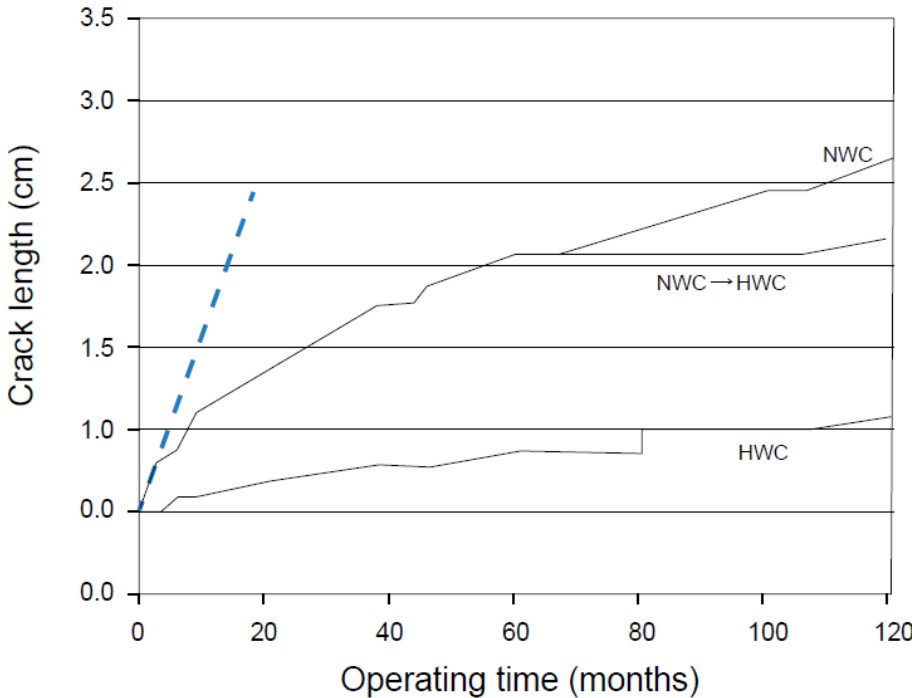

**Figure 26.** Predicted damage accumulation (crack length) in the heat affected zone adjacent to the H3 weld in a BWR core shroud over 10 years of operation for different hydrogen water chemistry (HWC) regimes. Note that the discontinuities in the CGR are due to reactor outages. The broken line represents the crack length in the case where no dependence of the CGR on crack length is recognized. Adapted from [73].

*6.6. Microfracture Frequency and Dimension*

As noted above [45], IGSCC in sensitized Type 304 SS in high temperature aqueous solutions propagates via the occurrence of microfracture events on the crack front, with the crack advancing by a distance $r$ on average for every $B/2r$ events occurring on the crack front, assuming that the events are semi-circular in geometry. $B$ is the thickness of the specimen (see Figure 24), typically 1.27 cm (0.5 inch). The CGR is given by Equation (1) and it is evident that the events are characterized by a frequency ($f$) and a dimension ($r$). These two quantities are readily calculated by the CEFM and are plotted as a function of CGR in Figure 27a,b. The MFF ($f$) was calculated for three models for estimating the crack tip strain rate. Ford et al.'s [37] empirical formula (Table 4) predicts that $f$ is independent of CGR, whereas the two small-scale yielding fracture mechanics expressions by Congleton [68] and Shoji [61–63] predict a linear log ($f$) vs. log (CGR) relationship for CGR that is clearly dominated by SCC (CGR > $4 \times 10^{-10}$ cm/s). At lower CGR, the MFF becomes independent of CGR to yield a correlation that is reminiscent of that between the CC and CGR. A significant difference exists between the crack tip strain rate models of Congleton [68]

and Shoji [61–63], even though both predict similar forms of the log (*f*) vs. log (CGR) correlation. In any event, for the conditions relevant to Figure 9, the Congleton CTSR expression predicts *f* = 1 Hz whereas Shoji's expression predicts *f* = 10 Hz. Both are within an order of magnitude of the experimental value of 2 Hz (Figure 8), which is acceptable agreement. Likewise, the MFD (*r*) is predicted to be about 3 μm for the conditions that are relevant (Figures 8 and 9), which is in excellent agreement with experimentation (2–3 μm). Finally, the two fracture mechanics-based models (Congleton et al. [68] and Shoji et al. [61–63]) predict that the MFF increases strongly with increasing CGR, as expected from the dependence of fracture frequency on $K_I$ (Figure 10) and from the relationship CGR = $Gfc^2$, where $G = 2/B$, if *c* (i.e., *r*) is only weakly dependent on the CGR. Because *c* is expected to be determined by the diffusion length of hydrogen in the matrix ahead of the crack, or by the spacing of $C_{23}C_6$ precipitates on the grain boundaries, it is evident that it should not depend on $K_I$ or on any other environmental variable. As shown in Figure 27b, the microfracture dimension (MFD) at low CGR values decreases sharply in the mechanical fracture region, but in the SCC region (CGR > $4 \times 10^{-10}$ cm/s) the microfracture dimension is predicted to be almost independent of CGR. Note that, in the case of the Ford et al. [37] model, the crack tip strain rate is a function of $K_I$ only so that for a fixed fracture strain the MFF is independent of the CGR.

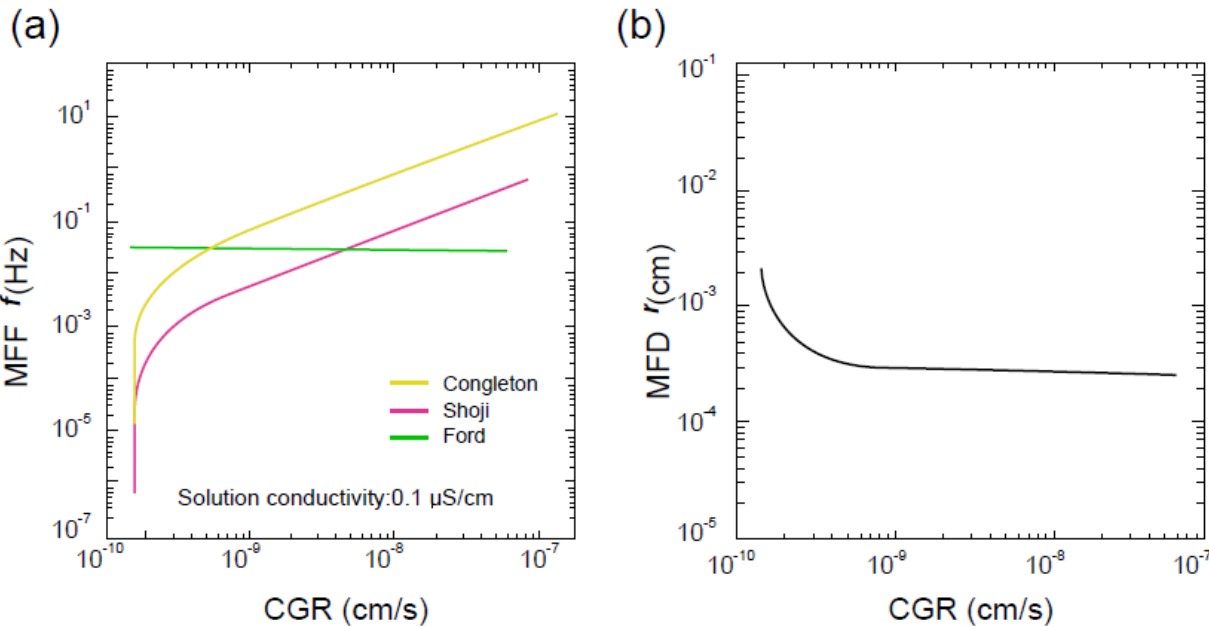

**Figure 27.** Calculated MFF (**a**) for three CTSR models and MFD (**b**) for IGSCC in Type 304 SS in water at 288 °C.

*6.7. Effect of Temperature*

Andresen [59] reported an interesting dependence on temperature of the CGR in sensitized Type 304 SS in simulated BWR primary coolant as shown in Figure 28. The experimental data (open circles) were determined for a stress intensity factor of 33 MPa·m$^{1/2}$, [O$_2$] = 200 ppb, and a sulfate concentration of 9.62 ppb. The CGR was observed to pass through a maximum at temperatures between 175 °C and 225 °C. The CEFM, with all three CTSR models, reproduced the maximum with a crack tip strain rate activation energy of 1000 kJ/mole [8]. The maximum arose from a competition between the thermal activation of the CTSR, which causes the CGR to increase with increasing temperature, and the decrease in the ECP, that results in a decrease in the CGR, noting the exponential relationship between the CGR and ECP (Figure 3).

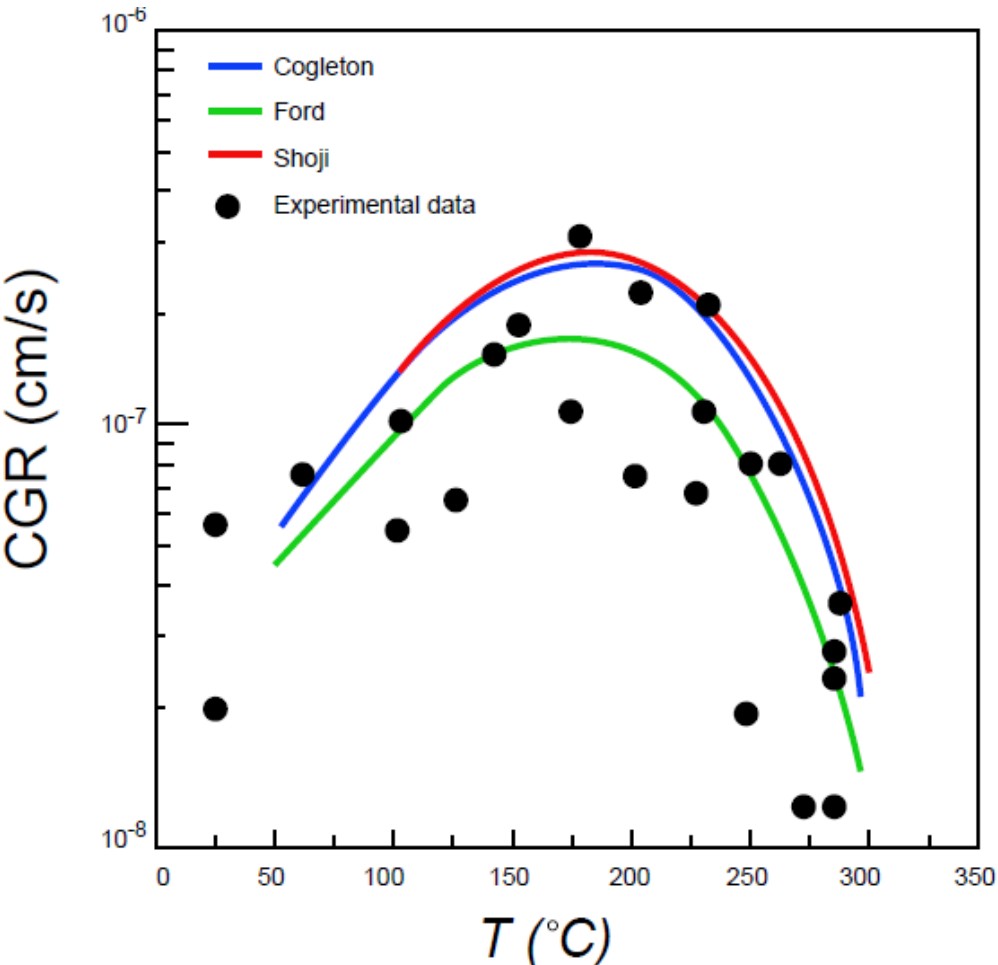

**Figure 28.** Plot of CGR for constant $[O_2]$, $K_I$, and sulfate concentration as a function of temperature calculated using different crack tip strain rate models by Ford [37], Congleton [68], and Shoji [61–63]. The ambient temperature conductivity was 0.27 µS/cm. The experimental data are from Andresen [59].

The CGR vs. temperature data plotted in Figure 28 have important implications for operating BWRs that undergo start-up and shutdown of the reactor [73,74]. Thus, for the stated conditions, the CGR at 288 °C is $2 \times 10^{-8}$ cm/s, corresponding to a crack extension of 0.63 cm/a, but during start-up and shut-down the CGR is about $2 \times 10^{-7}$ cm/s (6.3 cm/a) during the time that the reactor is at 125–225 °C. Of course, a reactor undergoing start-up and shutdown spends little time in the susceptible temperature region compared with the time spent under normal operation at 288 °C. Thus, for each hour spent in this 175–225 °C region the crack is predicted to advance by 7.2 µm, whereas for each hour spent at 288 °C (or 25 °C) the crack grows by about 0.72 µm. Nevertheless, Balachov et al. [73,74] predict that a significant fraction of the stress corrosion cracking damage incurred by a BWR under normal operating conditions accumulates during typical start-ups and shutdowns (Figure 29). It should also be noted that operational transients, such as start-ups and shutdowns, are accompanied by transients in conductivity due to hide-out return, which also affects the CGR.

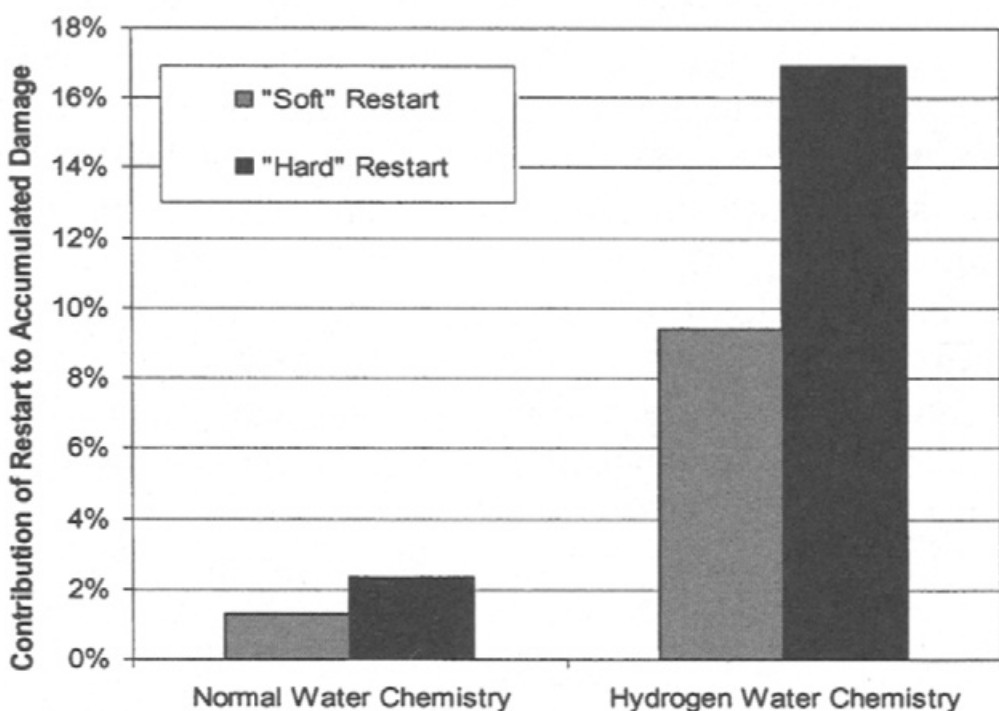

**Figure 29.** Predicted contribution of start-ups and shut downs to the total stress corrosion cracking damage incurred in the outer surface of the core shroud during normal operation of a BWR. Adapted from [74]. A soft restart corresponds to low conductivity excursion, whereas a hard restart is characterized by a large conductivity excursion.

The examination of Figure 29 indicates that the contribution of shutdowns and subsequent startups (i.e., a "restart") is much larger under HWC operating conditions than under NWC conditions. This is attributed to the effect of solution conductivity on the CGR, which partially negates the impact of HWC and to the fact that hydrogen is not typically added to the reactor feedwater until the reactor reaches operating temperature. In both cases (NWC and HWC), the impact of a "hard restart" during which there is a large conductivity excursion due to hideout return is predicted to be almost twice as much as that for a "soft restart", during which the conductivity excursion is much lower. Soft restarts can be affected by maximizing the flow of coolant through the ion exchange columns of the Reactor Water Cleanup Unit (RWCU), while restarting the reactor. This issue is discussed further in Section 6, in which the role of the corrosion evolutionary path (CEP) on the prediction of corrosion damage in an actual, operating BWR is presented.

### 6.8. Effect of Flow Velocity

Little systematic work has been reported on the effect of fluid flow rate on the CGR in Type 304 SS (or any other alloy for that matter) in reactor coolant systems, even though the coolant flow rates through various piping systems can be many m/s. It is of great importance, also, that the crack growth rates measured in laboratory studies are commonly carried out in quiescent autoclave environments and hence do not accurately replicate the conditions that exist in operating reactors. The CEFM permits this shortcoming to be corrected.

The role played by flow velocity is two-fold, depending critically on the aspect ratio (AR defined as length/width) of the crack. Thus, for a tight crack (AR > 1000), the crack tip and indeed most of the crack length is hydrodynamically isolated from the fluid flow in the external environment. In the case of a crack of moderate AR, the outer part of the crack is coupled to the hydrodynamics of the external environment, and turbulence-induced mixing destroys the concentration gradients and converts this region into being part of the "external surface" with the result that the ECL is reduced (Figure 30). Thus, fluid flow past

the mouth of the crack induces a clockwise, circular rotating eddy that in turn induces a counterclockwise rotating eddy below.

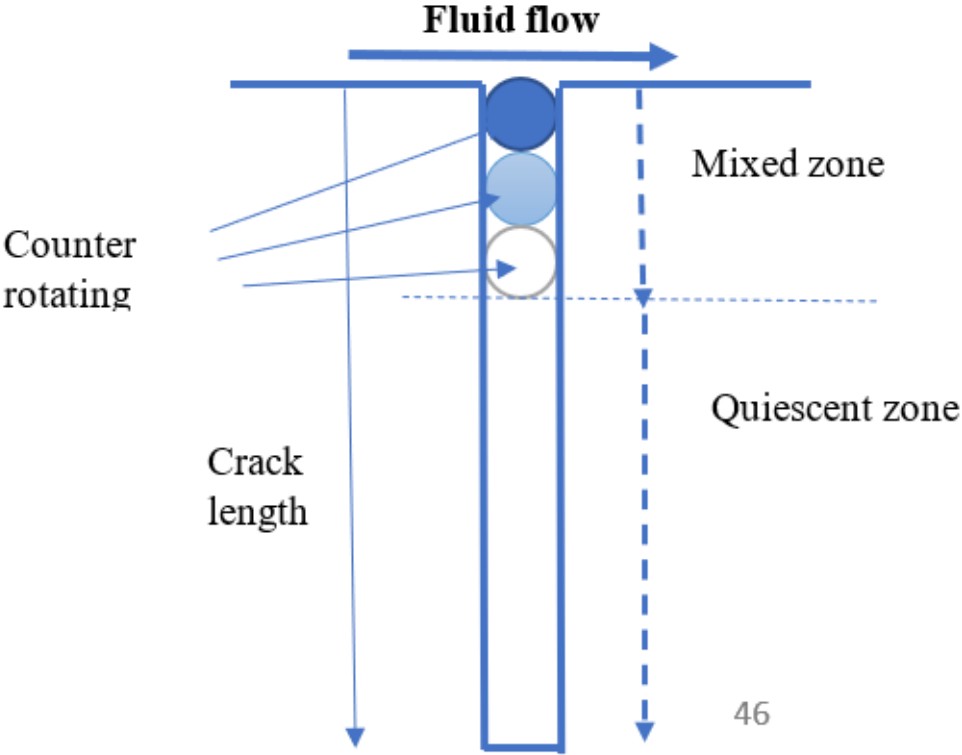

**Figure 30.** Impact of fluid flow on the electrochemical crack length (ECL).

Because of the limited transfer of momentum between neighboring eddies, the intensity of the eddies decreases with distance into the crack, as indicated by the change in color in Figure 30. This effect is illustrated experimentally in Figure 31 for the flow of water containing aluminum powder past a surface cavity of AR = 3. The flow was visualized optically. At some distance into the crevice, the induced mixing becomes negligible and the solution is quiescent. Thus, the surface area in the crack within the mixing zone can support the mass transfer of a cathodic depolarizer (e.g., $O_2$) and, hence, should be countered as being part of the external surface. Accordingly, the external area is increased (slightly) but the ECL decreases, resulting in an increase in the CGR. A similar approach was adopted to define the impact of loading frequency on the corrosion fatigue CGR in AA5083 in NaCl solution [75]. In that case, increasing frequency not only increases the rate of fracture at the crack tip but also enhances the exchange of solution between the crack and the external environment, thereby leading to a decrease in the ECL and, hence, to an increase in the CGR.

A somewhat simplistic expression can be derived by using the "compound interest" expression to describe the decay of eddy intensity (velocity) with distance into the crevice:

$$V_n = V_0(1 + \varsigma)^n \tag{16}$$

where $V_n$ is the velocity of the nth eddy in the crevice, $V_0$ is the velocity of the first eddy, $n$ is the eddy number, and $\varsigma$ describes the loss of momentum transferred from one eddy to another (neighboring) eddy. This equation can be rearranged to yield

$$n = ln(V_n/V_0)/ln(1 + \varsigma) \tag{17}$$

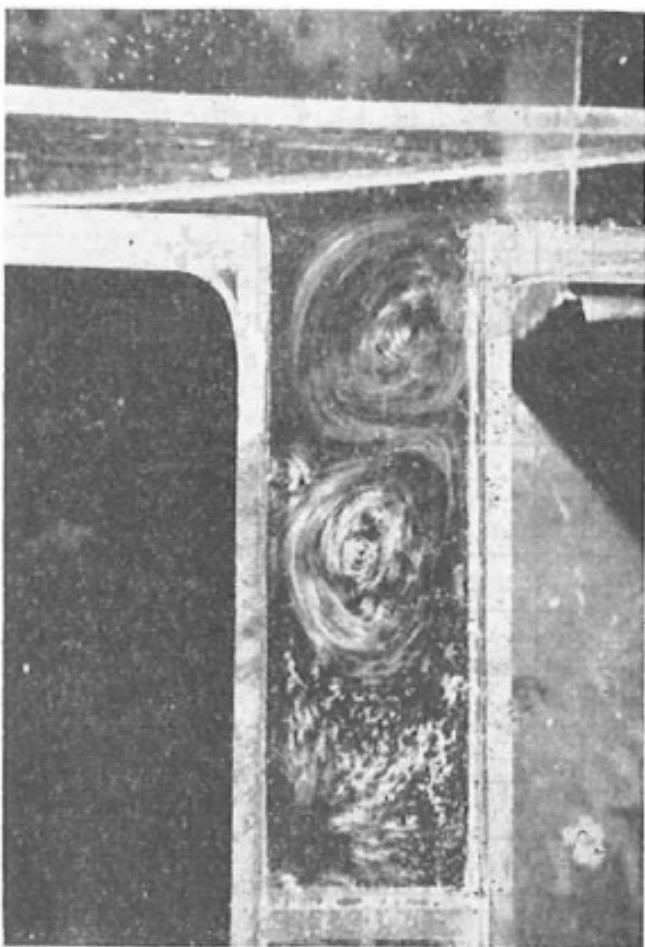

**Figure 31.** Experimental illustration of flow-induced rotating eddies in a cavity of AR = 3. [75], ©1966 ASME. This image is hereby used courtesy of the ASME.

For illustrative purposes, we assume that a system is quiescent when $V/V_0 < 0.001$ and that the loss of momentum between neighboring eddies is $\varsigma = -0.1$ (i.e., a 10% loss in momentum), which yields $n = 66$. Thus, according to this simplistic model, 66 eddies exist in the crevice for the fluid velocity to decrease by a factor of 1000. Accordingly, if the flow velocity across the surface was 1 m/s, the velocity of the 66th eddy into the crevice would be 1 mm/s. Now, for a parallel sided slot, the diameter of each eddy may be taken as the COD of the crack, or 0.001 cm (Table 2). Accordingly, the mixed zone (Figure 30) has a length of 0.066 cm, or about 13.2% of the 0.5 cm crack length assumed for a standard crack (Table 2). Thus, the ECL is reduced from 0.5 cm to 0.434 cm. The CGR is predicted by the CEFM to increase by a factor of about a factor of three.

The second contribution that flow makes to the crack growth rate is by increasing the rate of transport of a cathodic depolarizer (e.g., $O_2$ or $H_2O_2$) to the external surface. This has the effect of increasing the ECP (Figure 32a) and in supporting a higher CC, and hence CGR (Figure 32b). Figure 32a,b show that the impact of increased $[O_2]$ becomes increasingly muted as the flow velocity is increased.

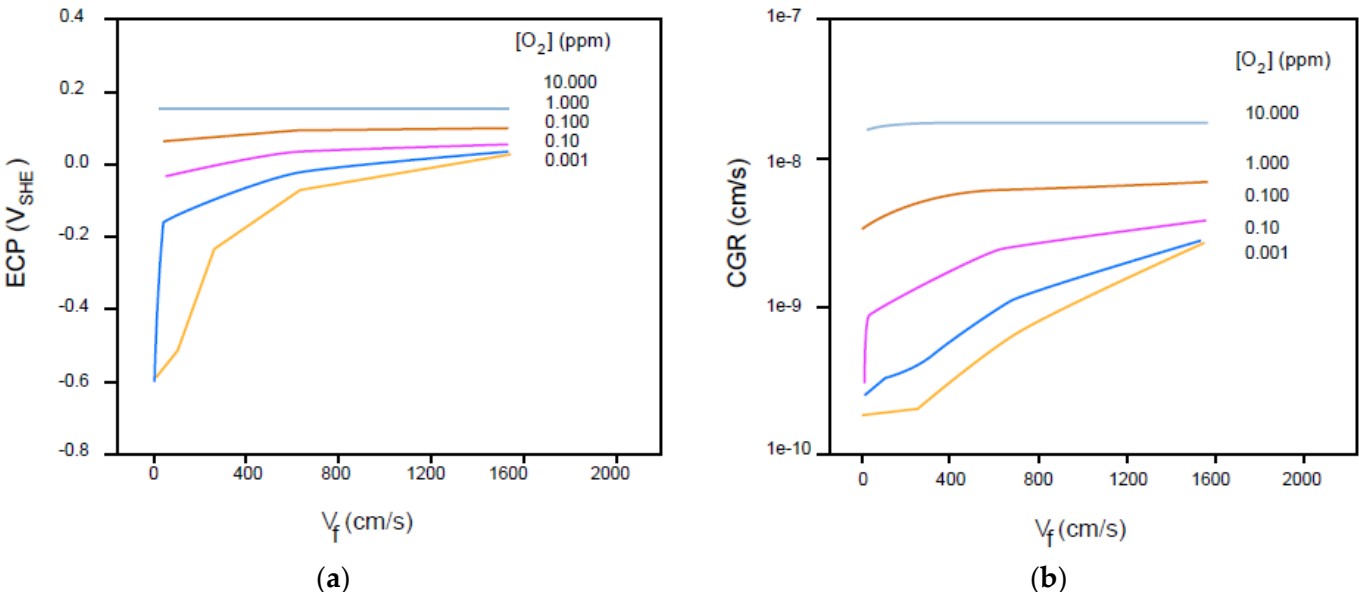

**(a)**

**(b)**

**Figure 32.** Effect of flow velocity on the ECP (**a**) and IGSCC CGR in sensitized Type 304 SS as a function of [O$_2$] in simulated BWR coolant (**b**) at 288 °C [12]. The aspect ratio (AR) = 500. Adapted from [12].

The impact of flow velocity and solution conductivity on the ECP and CGR are displayed in Figure 33a,b, respectively. Because the ECP does not involve the flow of a current through the external environment, but rather is established by the partial anodic and cathodic reactions acting on the same location [40], the ECP is independent of the solution conductivity (Figure 33a). In the case of the CGR (Figure 33b), the sensitivity of the CGR to flow velocity increases with increasing conductivity, particularly at low flow velocities.

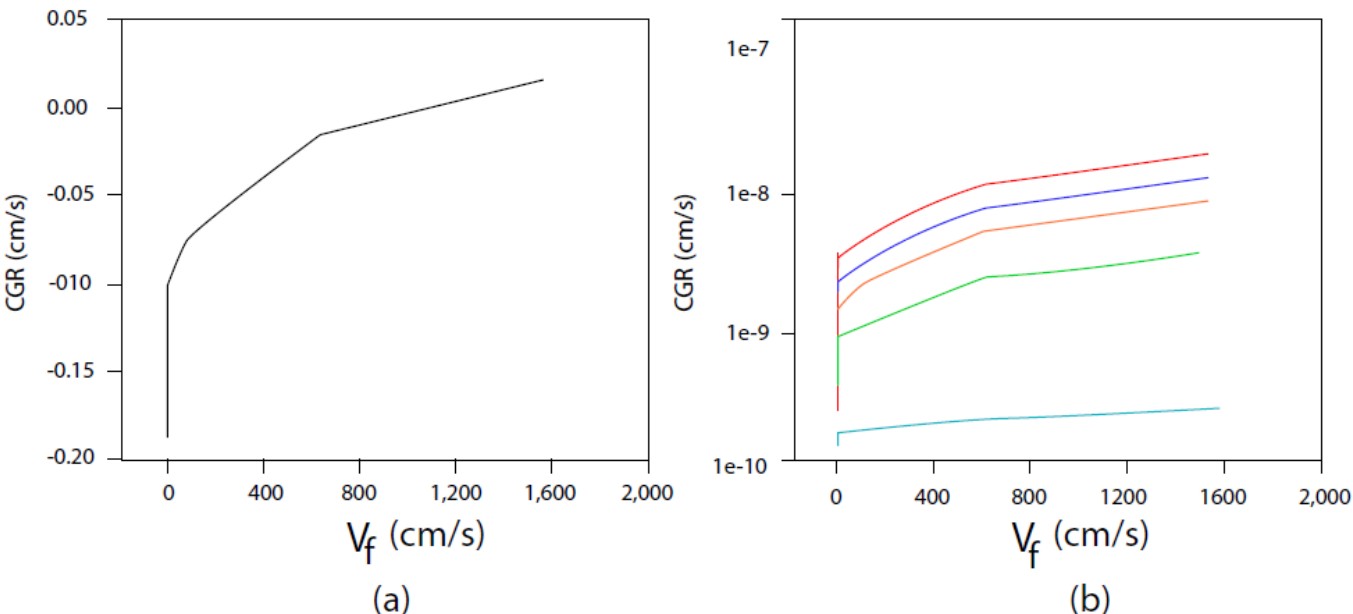

(a)

(b)

**Figure 33.** Effect of flow velocity on the ECP (**a**) and IGSCC CGR in sensitized Type 304 SS as a function of ambient temperature conductivity in simulated BWR coolant (**b**) at 288 °C, [O$_2$] = 200 ppb [12]. The aspect ratio (AR) = 500. Adapted from [12].

### 6.9. Effect of Solution Conductivity

The impact of conductivity on CGR for systems containing oxygen and hydrogen peroxide are displayed in Figure 34a,b, respectively. The conductivity determines the throwing power of the current exiting the crack mouth across the external surface. The synergistic relationship displayed in Figure 34 between conductivity and oxidant concentration arises because increasing conductivity exposes a greater external surface area upon which oxygen reduction can occur, the rate of which also depends upon the oxidant concentration. Accordingly, a larger CGR is induced.

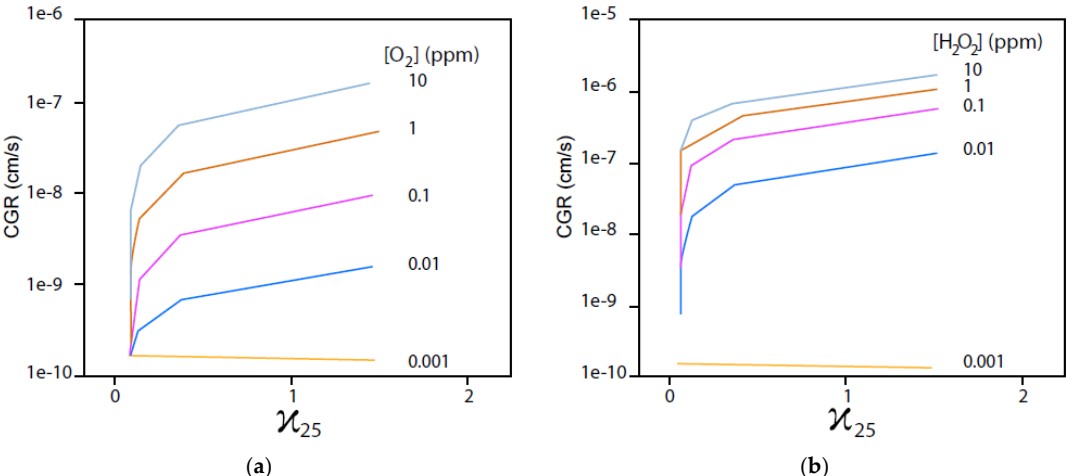

**Figure 34.** Plots of calculated CGR vs. ambient temperature conductivity as a function of [$O_2$] (**a**) and [$H_2O_2$] (**b**) for IGSCC CGR in sensitized Type 304 SS as a function of ambient temperature conductivity in simulated BWR coolant at 288 °C [12]. Other parameters as in Table 2. Adapted from [12].

Finally, we have listed the ambient temperature conductivity in the above figures but, of course, the operative conductivity is that at the operating temperature of the system (experiment or reactor at 288 °C). The ambient temperature values were used because that is reported by the reactor operator. The two are related, as shown in Figure 35a,b for a dilute NaCl solution, but it is important to note that these relationships depend upon the identities of the ionic species in the solution. Of course, the conductivity at the specified temperature was used in the calculation.

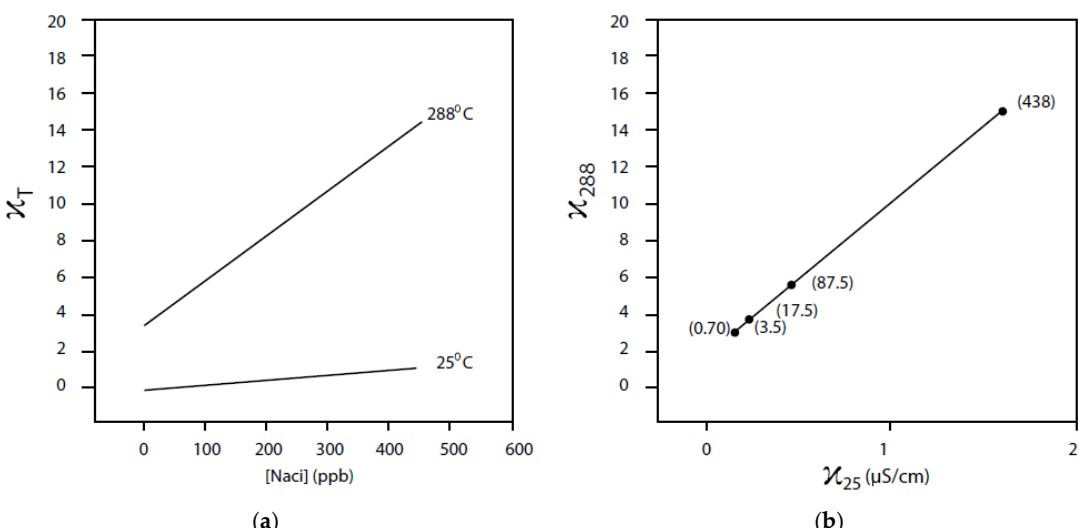

**Figure 35.** Dependence of solution conductivities at 25 °C and 288 °C as a function of [NaCl] (**a**) and correlation of conductivities at 25 °C and 288 °C (**b**). Adapted from [12].

*6.10. Development of Semi-Elliptical Cracks*

One of the most important accomplishments of the CEFM is its prediction of the formation of semi-elliptical cracks that are observed to develop in surfaces (e.g., pipe walls). Such a crack obtained under fatigue loading conditions is shown in Figure 36.

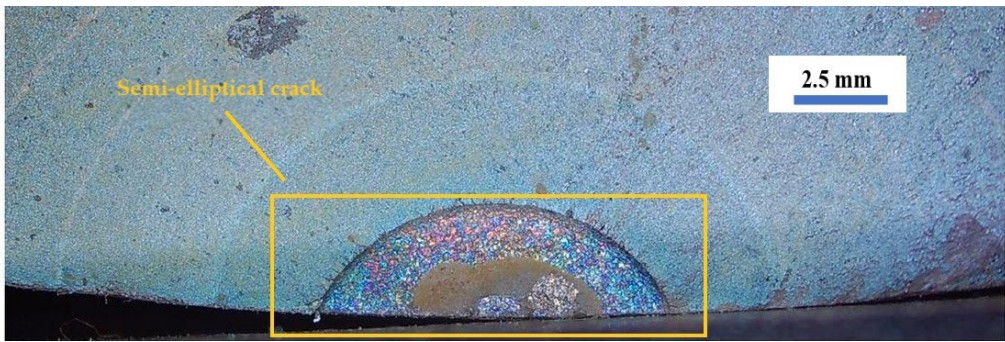

**Figure 36.** Semi-elliptical crack in the surface of a pipe [76].

Extensive effort has been expended by the fracture mechanics community to explain the formation of semi-elliptical cracks in terms of mechanical phenomena (e.g., creep crack growth). The approach has been to calculate the stress intensity factor at various locations along the crack front by initially starting with a semi-circular nucleus, and then progressing to an elliptical shape as the crack develops [76–78]. If $K_I$ at 90° from the origin is greater than $K_I$ at 0° and 180°, and if the creep crack growth rate increases with $K_I$ (Figure 15), the crack develops as a prolate ellipse with the major axis perpendicular to the surface (Figure 37). On the other hand, if $K_I$ at 90° from the origin is less than $K_I$ at 0° and 180°, the crack develops as an oblate ellipse with the major axis coincident with the surface. While both forms are predicted theoretically from mechanics, stress corrosion cracks are invariably oblate in form and this author is not aware of any reports of prolate surface stress corrosion cracks. An important point in this regard is that cracks grow at constant load; the CGR for purely mechanical creep is so low for Type 304 SS ($1.69 \times 10^{-10}$ cm/s), as indicated in Figures 4 and 15, that the crack would advance by only 2.1 mm over forty years of operation. Of course, under fatigue loading conditions the crack can advance much more rapidly. According to James and Schwenk [79], the fatigue crack growth rate (FCGR) in air at 600 °F (315 °C) for Type 304 SS can be written as $\frac{da}{dt} = 9.802 \times 10 - 16xfx(\Delta K_I)^{2.237}$, where $\Delta K_I$ is the stress intensity range in psi.in$^{1/2}$ (note that 1 psi·in$^{1/2}$ = 0.9100477 $\times 10^{-3}$ MPa·m$^{1/2}$). Choosing $\Delta K_I = 1.2 \times 10^4$ psi.in$^{1/2}$ (10.92 MPa.m$^{1/2}$), the cycle-based FCGR is $7.62 \times 10^{-6}$ cm/cycle or for a loading frequency of 1000 Hz, a FCGR is estimated to be $7.62 \times 10^{-3}$ cm/s. After 40 years of service, such a crack is predicted to grow to an impossible length of more than 9 m! Thus, while fatigue loading may well account for the growth of oblate and prolate surface cracks, creep does not appear to be a viable mechanism for producing cracks that threaten the integrity of the coolant piping system in a BWR or the correlation indicated above in inapplicable. Of course, in assessing the fatigue loading case, the comparison should be made between FCGR and the corrosion fatigue crack growth rate (CFCGR), but that comparison is beyond the scope of this paper.

According to the CEFM, semi-elliptical cracks develop because the CGR decreases with increasing ECL (Figure 23), since the increasing IR potential drop down the crack in the minor axis direction (perpendicular to the surface) subtracts from the electrochemical driving force of the crack. Thus, the crack growth rates along the surface and perpendicular to the surface can be calculated as a function of time using the recursive formulae [80,81].

$$L_{surface}(j+1) = L_{surface}(j) + \left(\frac{dL}{dt}\right)_{L=L^0} dt \qquad (18)$$

$$L_{perpendicular}(j+1) = L_{perpendicular}(j) + \left(\frac{dL}{dt}\right)_{L=L(j)} dt \tag{19}$$

where $L_{surface}(j)$ and $L_{perpendicular}(j)$ are the crack lengths along the surface (x-xis) and perpendicular to the surface (y-axis), respectively (Figure 37), $\left(\frac{dL}{dt}\right)_{L=L^0}$ and $\left(\frac{dL}{dt}\right)_{L=L(j)}$ are the corresponding CGRs, and $dt$ is the increment in time ($=t_{j+1} - t_j$). Because of the existence of the IR potential drop down the crack, $\left(\frac{dL}{dt}\right)_{L=L^0} > \left(\frac{dL}{dt}\right)_{L=L(j)}$ and becomes increasingly so as time proceeds (i.e., as the crack length normal to the surface increases). Note that $\left(\frac{dL}{dt}\right)_{L=L^0}$ is a constant because the ECL ($L_0$), which is the length of the least resistant path from the bottom of the crack nucleus at the crack edge (intersection of the crack with the surface) to the external surface, is very small for all times and remains approximately invariant with time. The equation for the oblate case is given by

$$\left(\frac{x}{L_{surface}}\right)^2 + \left(\frac{y}{L_{perpendicular}}\right)^2 = 1 \tag{20}$$

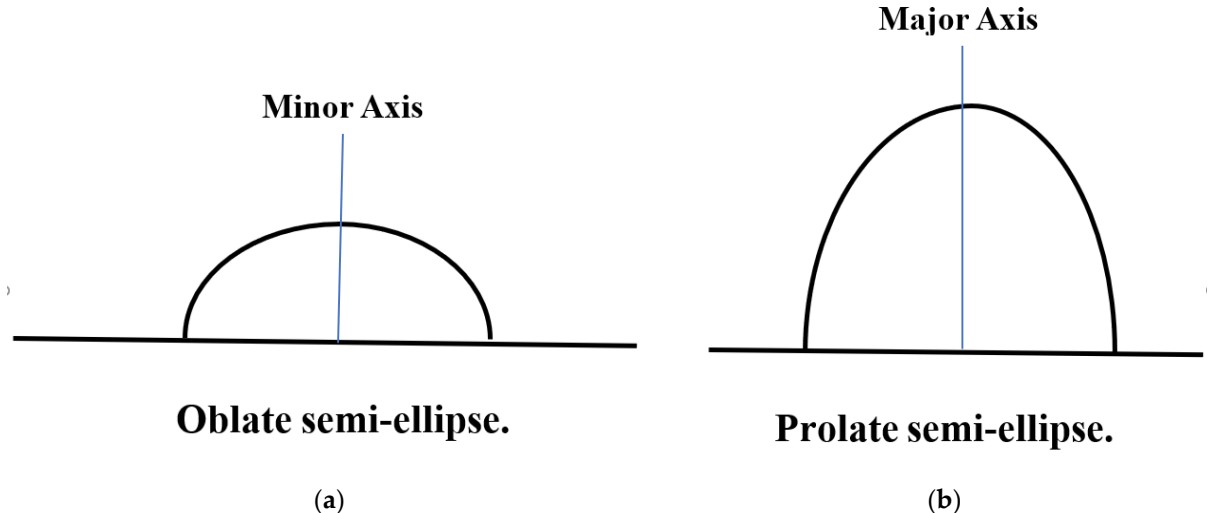

**Figure 37.** Oblate (**a**) and prolate (**b**) ellipses.

    Thus, the crack length along the surface becomes progressively greater than that normal to the surface, resulting in the development of an oblate semi-circular surface stress corrosion crack. For the crack to be prolate, somehow the CGR would have to be larger in the perpendicular direction than along the surface. We have not been able to simulate that possibility using the current theory. In performing these calculations, the stress intensity factor along the crack front was evaluated using Equation (21) [80,81]:

$$K_I = \frac{\sigma_0 \sqrt{\pi L}}{E(k)} \left(sin^2 \theta + \alpha^2 cos^2 \theta\right)^{1/4} f\left(\alpha, \frac{a}{B}, \frac{b}{W}, \theta\right) \tag{21}$$

where the various geometrical parameters are defined in Figure 38, where $f(\alpha, \theta) = (1.13 - 0.09x)$ $\left[1 + 0.1(1 - sin^2 \theta)^2\right]$, $k = 1 - \alpha^2$, $0 \leq \alpha \leq 1$, $0 \leq \theta \leq \pi$, $\alpha = L_{perpendicular}/L_{surface}$, and

$E(k)$ is the complete elliptical integral of the second kind. Accordingly, the value of $K_I$ at the edge and center of a surface crack are

$$K_I(\theta = \pi/2) = \frac{\sigma_0\sqrt{\pi L}}{E(k)}\left(1.13 - 0.09\frac{L_{perpendicular}}{L_{surface}}\right) \tag{22}$$

and

$$K_I(\theta = 0, \pi/2) = \frac{1.1\sigma_0\sqrt{\pi L}}{E(k)}\sqrt{\frac{L_{perpendicular}}{L_{surface}}}\left(1.13 - 0.09\frac{L_{perpendicular}}{L_{surface}}\right) \tag{23}$$

where $\theta$ is defined in Figure 38.

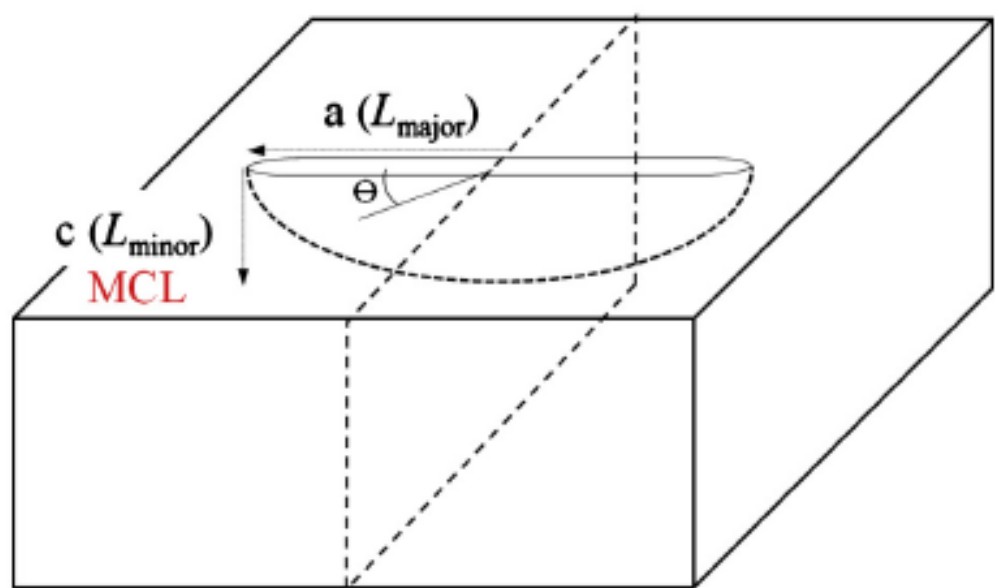

**Figure 38.** Semi-elliptical oblate surface crack in a plate. Adapted from [80].

The predicted evolution of an oblate semi-elliptical stress corrosion crack is presented in Figure 39 [80]. The major and minor axes were continually updated with respect to $K_I$ (black lines) or were plotted without updating the stress intensity factor (red line); one sees that updating KI has virtually no effect on the crack shape or dimension. The lack of impact of increasing $K_I$ on the evolution in the shape and dimension of surface stress corrosion cracks is a general finding in our work, but this finding should not be extended to cracks that grow by purely mechanical means (i.e., under creep conditions) where it is also possible that prolate creep cracks might develop according to the solution for the stress intensity factors along the major and minor axes that are adopted. The lack of the dependence of shape crack on the increase in $K_I$ is a manifestation of the weak dependence of the CGR on $K_I$ in the Stage II region of the CGR vs. $K_I$ correlation [80].

The predicted evolution of the shapes of oblate stress corrosion cracks in a stainless-steel surface as a function of [O$_2$], and hence ECP, are displayed in Figure 40. The corresponding ECP for [O$_2$] = 1 ppb, 10 ppb, 100 ppb, 1 ppm and 10 ppm are estimated using the mixed potential model (MPM) to be $-0.6025$, $-0.1987$, $-0.0818$, $-0.024$ and $0.1259$ V$_{she}$, respectively, for sensitized Type 304 SS in BWR primary coolant (water) at 288 °C [81]. Note that the stress intensity factor was updated with the progression of the crack, and note also the different scales on the axes for each [O$_2$]/ECP combination. The crack nucleus was assumed to be semi-circular with a 10 μm radius, corresponding to a small pit, for example. The nucleus grows by creep alone, because at 1 ppb O$_2$ the ECP < $E_{crit}$ ($-0.23$ V$_{she}$) and SCC is inactive. For all other [O$_2$], the ECP > $E_{crit}$, so that the crack advances by stress corrosion cracking, the rate of which depends on the ECL.

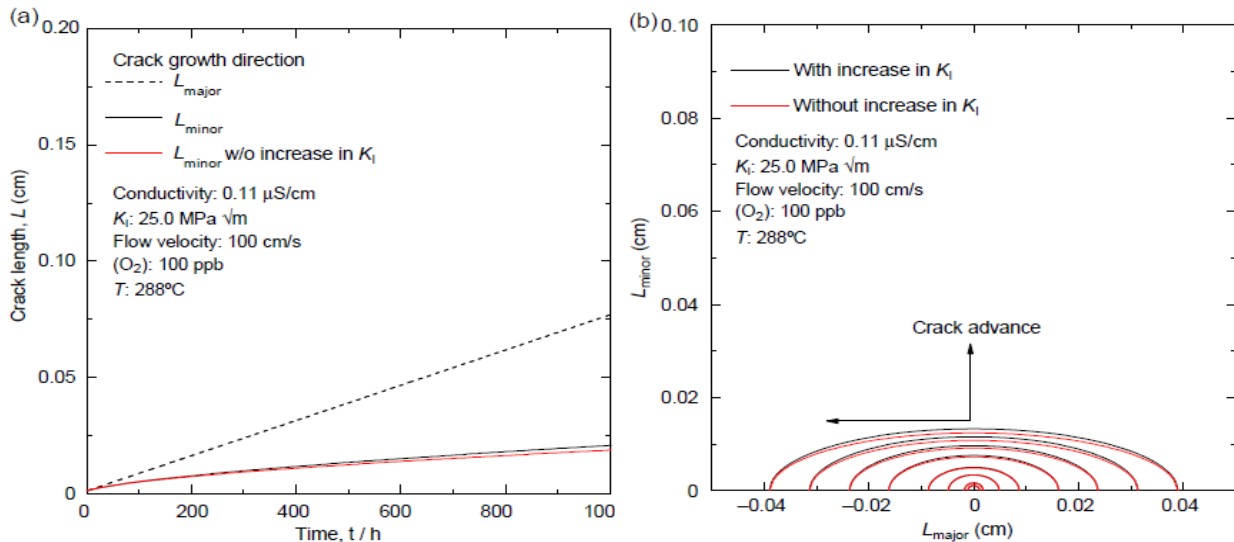

**Figure 39.** Evolution of an oblate semi-elliptical stress corrosion crack for the non-default parameter values given in the figure. (**a**) Minor and major axes vs. time. (**b**) Oblate surface stress corrosion cracks vs. time. Adapted from [80].

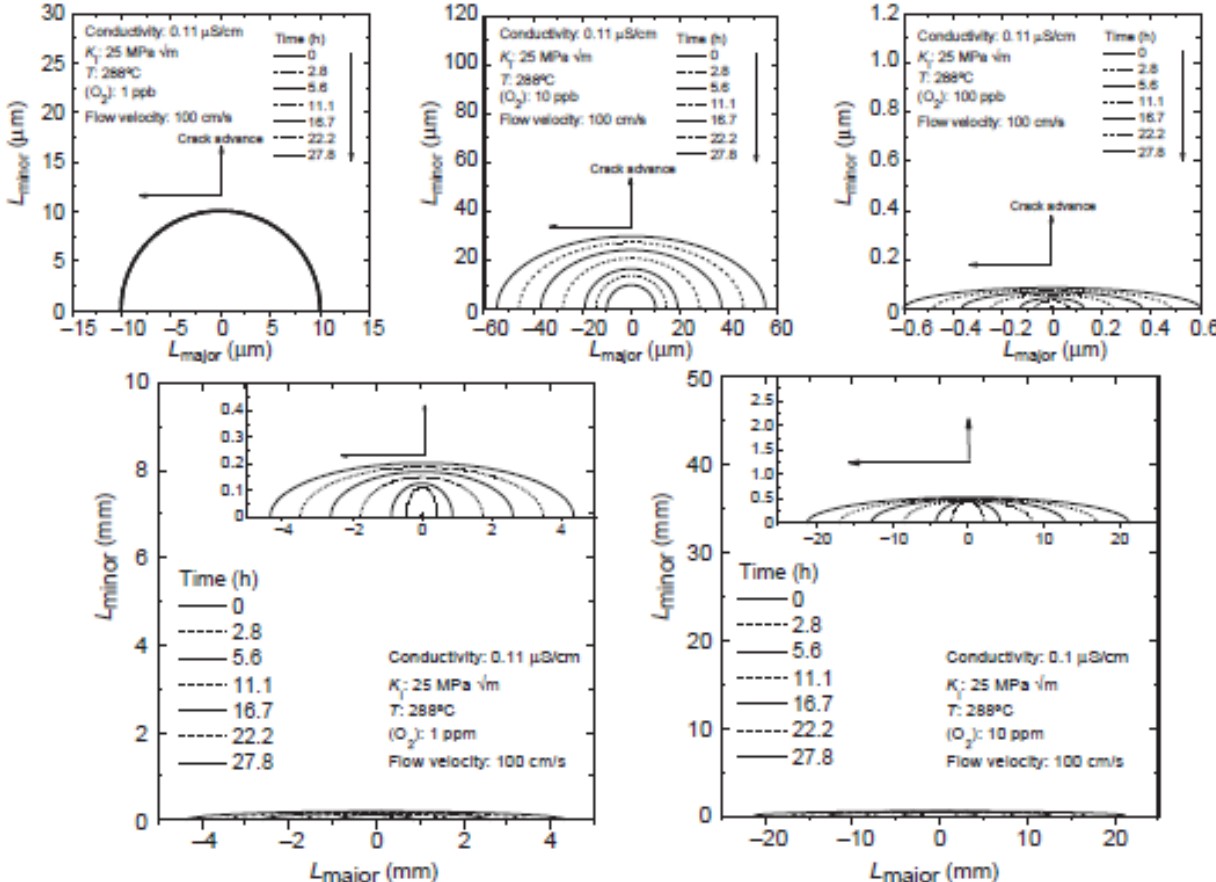

**Figure 40.** Predicted crack shapes vs. time as a function of [$O_2$], and hence ECP, that the ECP for [$O_2$] = 1 ppb, 10 ppb, 100 ppb, 1 ppm and 10 ppm are estimated to be $-0.6025$, $-0.1987$, $-0.0818$, $-0.024$ and $0.1259$ V$_{she}$, respectively, for sensitized Type 304 SS in BWR primary coolant (water) at 288 °C. Adapted from [81]. Note that the stress intensity factor was updated with the progression of the crack. Note also the different scales on the axes for each [$O_2$]/ECP combination.

The predicted impacts of conductivity and stress intensity factor, both of which affect the stress corrosion CGR, on the evolution of oblate semi-elliptical cracks in the surface of stainless steel are illustrated in Figure 41. As expected, both parameters exert an influence of the evolution of the shape, particularly the conductivity, which is consistent with the dependence of this independent variable on CGR, as discussed in Section 6.9 above.

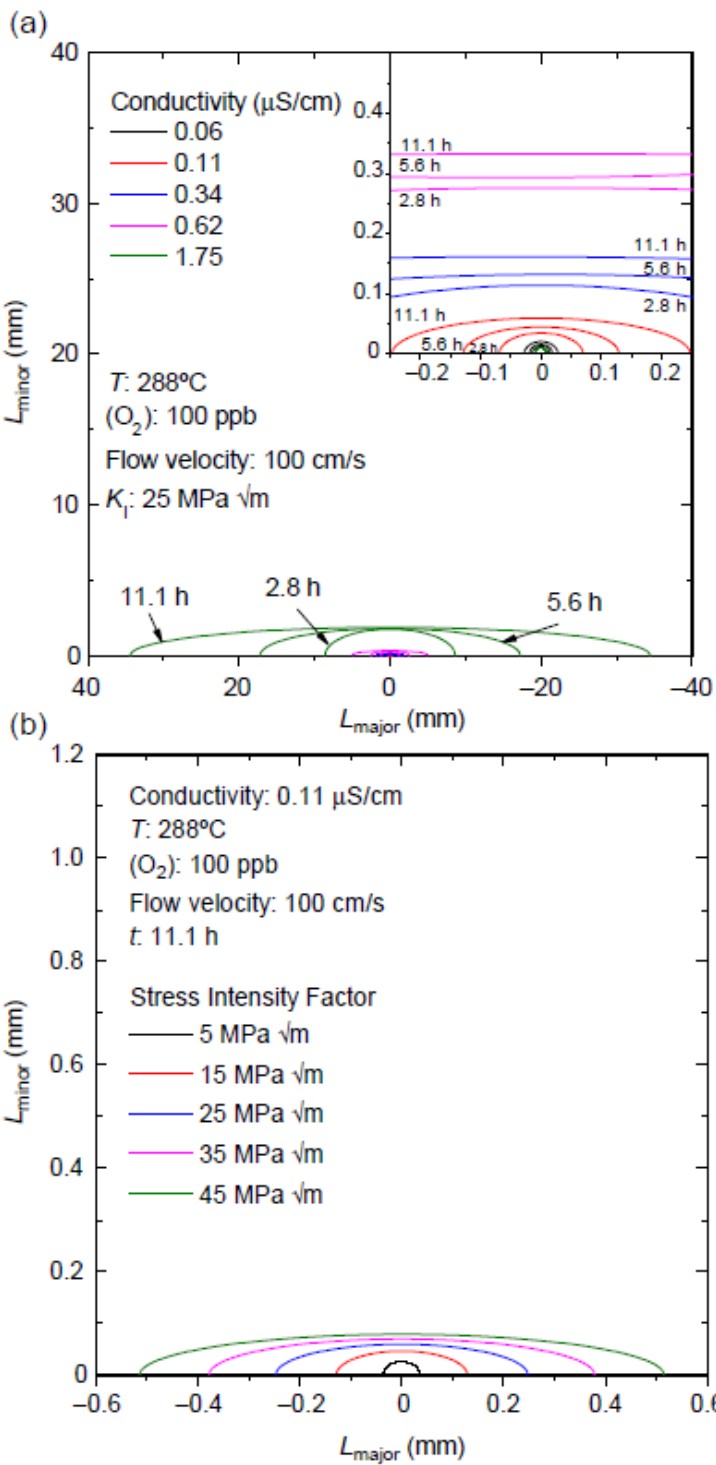

**Figure 41.** Evolution of an oblate semi-elliptical surface stress corrosion crack in sensitized Type 304 SS in BWR primary coolant (water at 288 °C) as a function of solution conductivity (**a**) and stress intensity factor (**b**). Adapted from [81].

Flow velocity is also predicted to either increase the CGR (tight cracks) or decrease the rate (open cracks), and hence is predicted to impact the evolution of oblate, semi-empirical surface cracks. Theoretically, at short times, when the crack nucleus is open to the environment, flow is expected to inhibit the SCC CGR because flow induced turbulence within the crack enclave prevents the establishment and maintenance of the aggressive conditions at the crack tip that are necessary to maintain pit/crack growth. As a sharp crack develops from the nucleus, this "washing out" effect becomes progressively muted and the CGR displays an increasingly positive relationship with flow velocity. By the time the crack aspect ratio (length/width) reaches 100, the crack tip is effectively "decoupled" from the external hydrodynamics and the impact of flow can be attributed to the enhanced transport of a cathodic depolarizer to the external surface. However, flow may have a major impact on the initiation time for SCC.

### 6.11. Global Assessment of the Accuracy of the CEFM

While the discussion presented above explored the veracity of the CEFM in calculating the CGR in response to changes in individual independent variables, the issue remains as to its ability to calculate the CGR when multiple independent variables change simultaneously. This issue was explored by using the CEFM to calculate the CGR for each set of independent variables (ECP, $T$, $K_I$, conductivity, flow velocity, etc.) in the experimental database that was used to train the ANN (Figure 7). This "CEFM" dataset was then used to train the ANN in the same manner that was used in training the net on the experimental database. A plot was then made of the CEFM-calculated CGR vs the CGR calculated using the net trained on the experimental database for the same set of independent variables and the plot is shown in Figure 42.

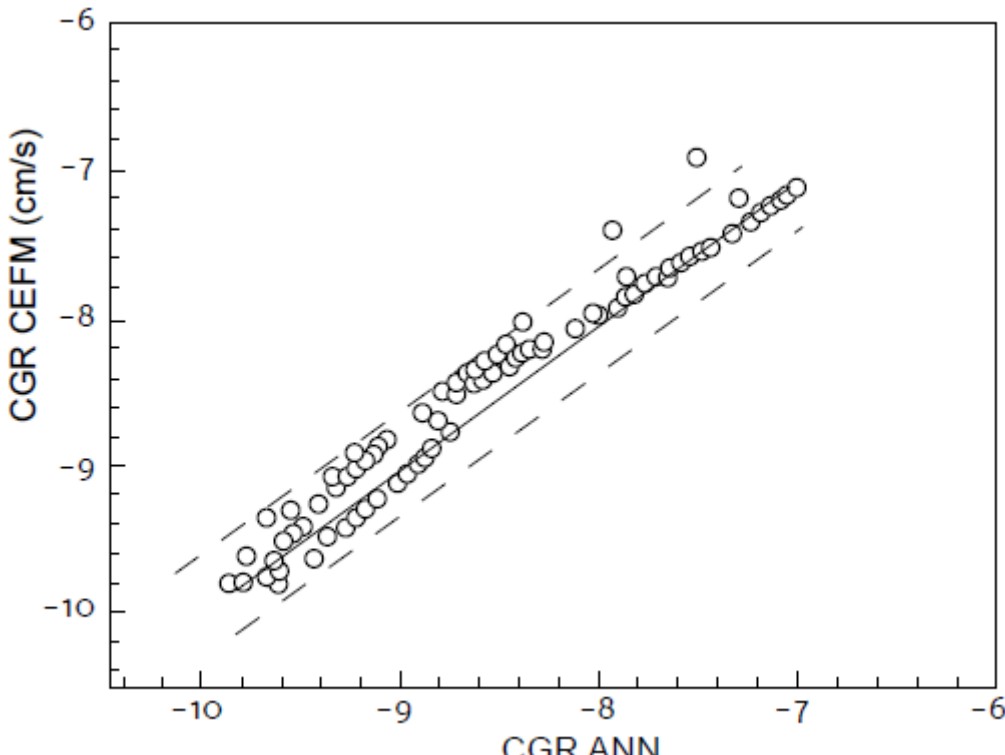

**Figure 42.** Comparison between CGR for IGSCC in simulated BWR primary coolant (pure water at 288 °C, calculated via the trained ANN, and that calculated using the CEFM for identical sets of independent variables [38]. The broken lines define the 95th percentile of ±0.4Log units. Adapted from [38].

Figure 42 shows that the CEFM and the ANN agree within a 95th percentile of ±0.4Log units (± a factor of 2.5) over wide ranges in the independent variables, which matches the uncertainty in the original database (Figure 7). This is about the accuracy with which CGR can be measured under optimal conditions and this has led to the opinion that the CEFM can calculate the CGR about as accurately as it can measured for the ranges of the independent variables considered in this study.

## 7. Corrosion Evolutionary Path

The final issue that is addressed in this paper is the Corrosion Evolutionary Path (CEP), which is the path taken by the system in terms of the independent variables that have a significant impact on the damage accumulation rate (CGR, in this case) as SCC damage accumulates. As noted earlier in this paper, this is frequently the most difficult aspect of the deterministic prediction of damage because there are no natural laws that fully govern future behavior. Two general approaches have evolved to address this issue:

1.  Assume that the system will behave in the future as it has in the recent past for which a record exists on the evolution of the independent variables. This is a viable approach for modeling nuclear reactors because of the wealth of information that is recorded during operation, although the data are not always of the type or in the form that are readily incorporated into predictive models. For example, power plants record ambient temperature conductivity, not the conductivity at the operating temperature that is employed in the CEFM. Likewise, to the author's knowledge no nuclear plants regularly monitor the ECP at any point in the coolant circuit. Fortunately, these parameters can be calculated with sufficient accuracy to permit their inclusion in the models.

2.  Assume a future operating history (CEP) that is designed to probe the impact of specific operating issues, such as HWC, reduced conductivity, stress relief, etc. These "what if" scenarios are one of the great benefits of the modeling described in this paper, because they allow issues to be addressed in a computer that are not practical in an operating reactor. For example, if one sought to define the cost/benefit of operating a reactor with ultralow conductivity, that is more easily done, and at much lower cost, with programs developed by the author and his colleagues, such as REMAIN, ALERT, FOCUS and MASTER_BWR, than by installing the additional ion-exchange columns in the RWCU system that would be required to achieve the desired low conductivity.

The primary coolant circuit of a typical BWR with external coolant pumps is displayed in Figure 43. Starting at the bottom of the core, the coolant flows upward through the core while boiling to produce steam and undergoes radiolysis to form the various radiolysis products. The steam/water mixture exits the top of the core and transfers into the upper plenum along with the water flowing through the core bypass (where it does not boil). The steam and water are separated via "steam dryers" and the dry steam is sent to the turbines to generate electrical power. The separated water flows to the mixing plenum, where it is combined with feedwater, which comprises returning water from the condensers and make-up water as needed. The coolant then flows into the upper downcomer, where it partitions into streams into the jet pumps and into the lower downcomer. The water entering the jet pumps draws water from the upper downcomer where the combined flow enters the recirculation system, which is activated by the main coolant pumps. The flow from the jet pumps empties into the bottom of the lower plenum and then flows into the bottom of the core via the top of the lower plenum, as shown in Figure 43.

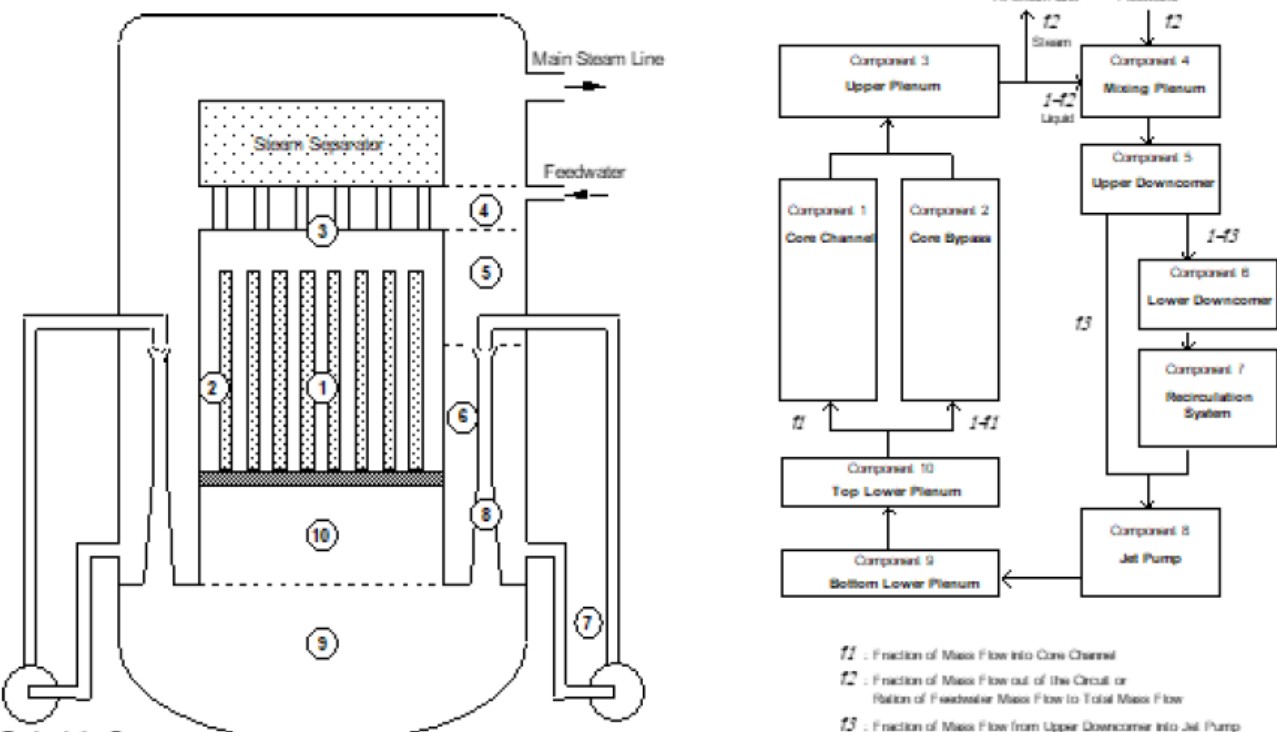

**Figure 43.** Schematic of the primary coolant circuit of a BWR. 1. Core; 2. Core bypass; 3. Upper plenum; 4. Mixing plenum; 5. Upper downcomer; 6. Lower downcomer; 7. Recirculation system; 8. Jet pump; 9. Bottom of lower plenum; 10. Top of lower plenum. Adapted from [73,74].

Over the past 25 years, the author and his colleagues have developed a series of programs to model the coolant chemistry and electrochemistry, including calculation of the CGR and integrated damage in the primary coolant circuits of BWRs, including DAMAGE-PREDICTOR, REMAIN, ALERT, FOCUS, and MASTER_BWR. The codes have evolved in an evolutionary manner, with each extending the capability of the previous code in some important manner, with the final code MASTER_BWR [34] predicting IGSCC damage in any location in the coolant circuit over any specified corrosion evolutionary path.

All the codes have a common structure, as summarized in Figure 44 except for DAMAGE PREDICTOR [13–18], which does not predict the integrated damage but was limited to predictions at a single state point. The other codes calculate the various output parameters (*T*, ECP, flow velocity, conductivity, etc.) at closely spaced state points along the CEP, and then integrate the CGR to determine the integrated damage over the CEP. The algorithm employed to facilitate the calculations is summarized in Figure 44.

The algorithm begins with the input of the CEP, as formed by listing those parameters of the plant and its operation as a function of time as a continuous set of closely spaced state points. Included in the CEP are the relevant void fractions in the core, the geometric, thermohydraulic parameters (temperature, flow velocity), and the dose rates for neutron and $\gamma$-photon irradiation as a function of elevation in the core. The thermohydraulic and dose rate data are obtained by RELAP and [15] and similar codes, respectively. These data are then used to model the radiolysis of water in a sub-code called RADIOCHEM [15] that describes the radiolytic generation of primary radiolysis products (H, O, OH, $e^-$.aq, $O_2^-$, etc.), which then react amongst themselves to produce even more radiolysis species (e.g., $O_2$, $H_2$, $H_2O_2$; 11 in all). These species are transported by convection, simultaneously with the undergoing reaction, throughout the coolant system by the flowing coolant. The solution of the set of stiff differential equations yields the concentrations of all species at 1 cm increments around the circuit, except in the recirculation system where the increment is 10 cm. These concentrations are then used in the CEFM to calculate the ECP and, hence,

the CGR, with temperature, conductivity, stress intensity factor and flow velocity also being employed in the calculation. Thus, the codes yield the ECP and CGR at 1 cm spaced points around the entire coolant circuit (10 cm in the recirculation system). The CGR is then integrated over time along the CEP at each location of the coolant circuit to yield the crack length at any point along the path.

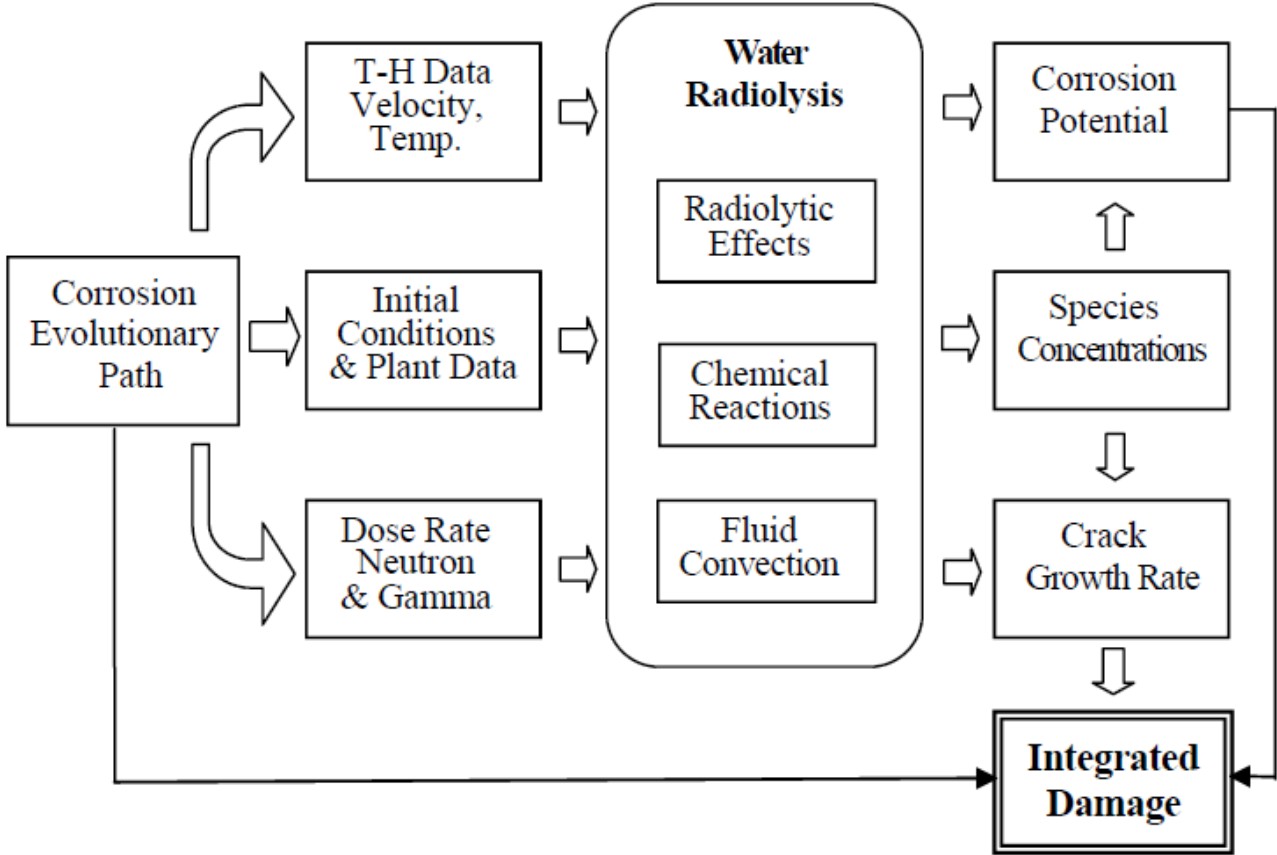

**Figure 44.** Algorithm of the damage prediction codes for BWR primary coolant circuits. Adapted from [13,14].

Because a viable crack initiation model was not available at the times of code development, all the codes listed above assume the presence of a preexisting "standard" crack at all locations. This crack has a length of 0.5 cm, a COD of 0.001 cm, a width of 1 cm, and a stress intensity factor of 25 MPa·m$^{1/2}$. Of course, this standard crack may be replaced by an actual crack having characteristics selected by the use, as is shown below. Accordingly, the current codes are particularly useful for predicting the damage that may evolve from a crack that has already been detected by an inspection during a planned outage. However, crack initiation models have now been developed [34] and these models will be inserted into BWR_MASTER soon. An important feature of the codes developed later than but including ALERT is the inclusion of the impact on CGR of increasing ECL, because without that feature all CGR codes greatly over-predict the evolution of IGSCC damage (Figure 27). Thus, in Figure 27, if the impact of ECL on the CGR was not included, the predicted crack length after 20 months would be about 2.6 cm rather than about 1.4 cm, which could have important consequences for any cost/benefit analysis. This is a matter of the utmost importance because the over-prediction of damage may have serious economic impact for a plant operator who might be tempted to install expensive remedial measures that are not warranted by the actual evolution of the damage.

The application of these codes is illustrated below by the application of ALERT to predict the evolution in the HAZ of the H3 weld on the top and inside surface of the core shroud in the Chinshan BWR in Taiwan [82]. The technical data for input into ALERT were taken from the NRC. The input data are displayed in Figure 45 for the reactor core starting at the core bottom (note that the inside of the core shroud is part of the core of the reactor). The columns contain data for (in succession from left to right) the distance into the core (from the bottom); the temperature; the liquid coolant flow velocity; the steam flow velocity; the void fraction (the fraction of the coolant that is in the steam phase); the section hydrodynamic diameter; the neutron and γ-photon dose rates (in Rads/s); a parameter that specifies if specific measures are in place to catalyze the destruction of hydrogen peroxide (a value of 1 indicates none); the standard exchange current density multiplier (SECDM) for the hydrogen electrode reaction (HER), the oxygen electrode reaction (OER), and the hydrogen peroxide electrode reaction (HPER) on the steel surfaces as specified in the CEFM (see Section 5.2); the stress intensity factor for the crack; the initial crack depth; the critical stress intensity factor (if desired); and a critical crack length, again if desired. Note that, in this case, the initial crack length was specified as 0.3 cm and the stress intensity factor as 15 MPa·m$^{1/2}$ rather than as specified in the standard crack. As given in the original publication, a table of this type is developed for each region of the coolant circuit, as specified in Figure 43.

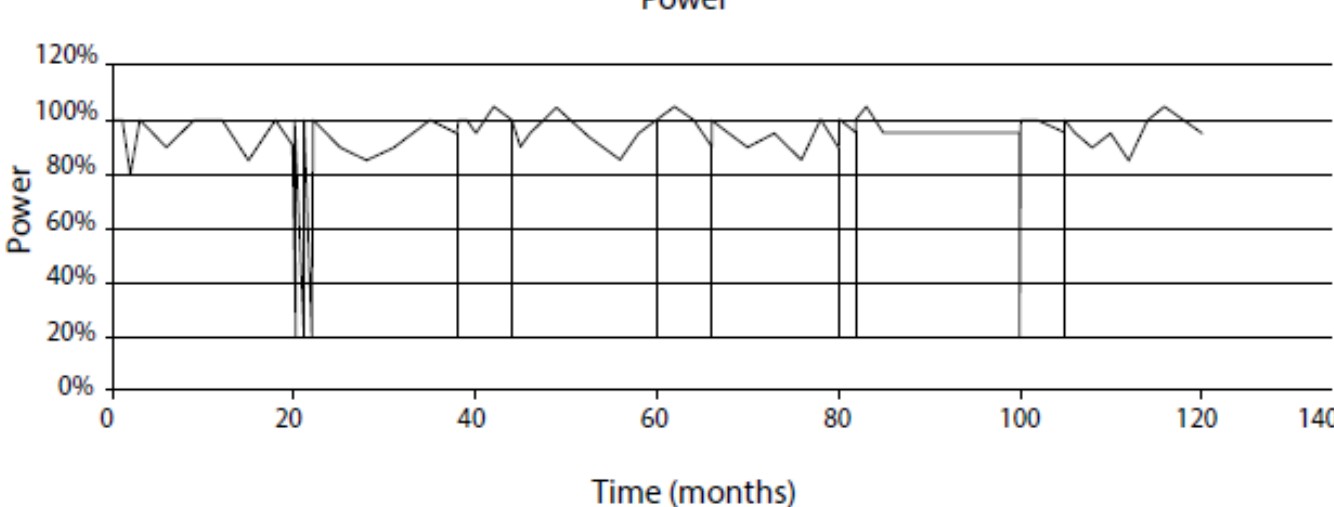

**Figure 45.** CEP of the Chinshan BWR with respect to reactor power [82].

The next step involves specifying the CEP (Figure 46). The inputs are the time of operation under any given set of conditions (including outages); the power level of the reactor; the hydrogen and oxygen concentrations in the feedwater (0 [H$_2$] indicates NWC, 1 ppm indicates a specified level of HWC); the flow rate (assumed to be proportional to the power level); the % of rated full-power temperature; and the concentrations of the three common anionic impurities (Cl$^-$, SO$_4^{2-}$ and CO$_3^{2-}$) with the counter cation assumed to be Na$^+$. CEP contains data for the full 120 months or for whatever operating period is desired. Note that the data specifying the CEP displays power-downs and power-ups and that during these periods HWC may be discontinued. It is CEPs of this kind that may be used to explore operational "what-if" scenarios, which is a particularly valuable facility of the codes.

The output file from ALERT (for the core channel) obtained in modeling the Chinshan BWR in Taiwan includes the distance from the core bottom, the temperature, the pH, and the concentrations of the principal radiolysis products of OH, $H_2O_2$, $HO_2$, $HO_2^-$, $O_2$, $O_2^-$ and $H_2$ [73,74]. It is found that $H_2$, $H_2O_2$ and $O_2$ are the dominant species and that, according to the Mixed Potential Model (MPM) [40], they are the only species that need to be considered in establishing the ECP and hence the CGR.

The CEP of the reactor over the 120 month operating period with respect to reactor power level, feedwater hydrogen, coolant oxygen, coolant hydrogen, coolant hydrogen peroxide, ECP, CGR, stress intensity factor for constant load but corrected for the increase in crack length, and the integrated damage is shown in Figures 45–53, respectively.

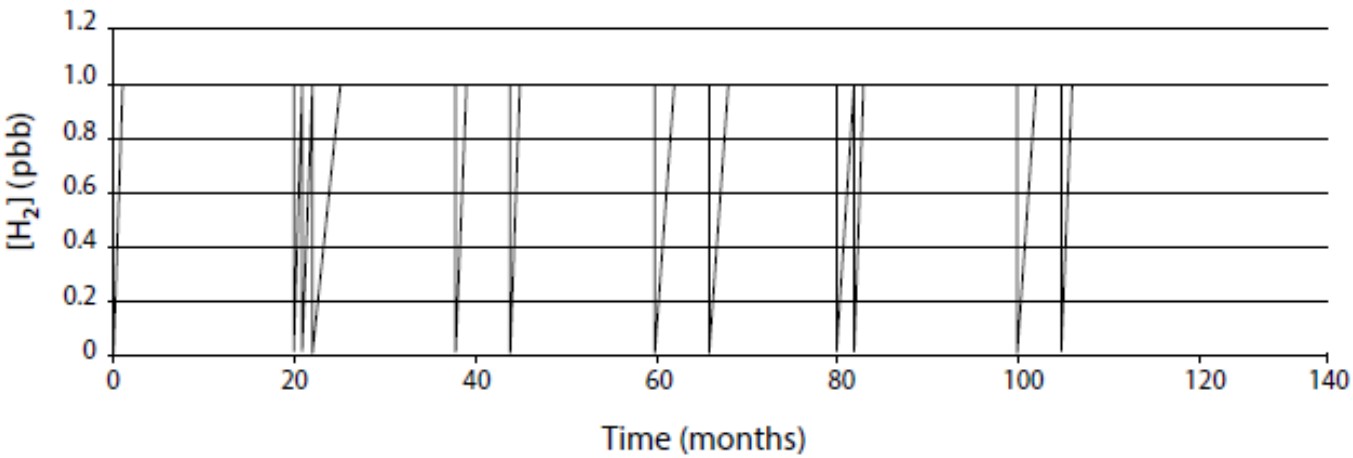

**Figure 46.** CEP of the Chinshan BWR with respect to feedwater hydrogen injection [82].

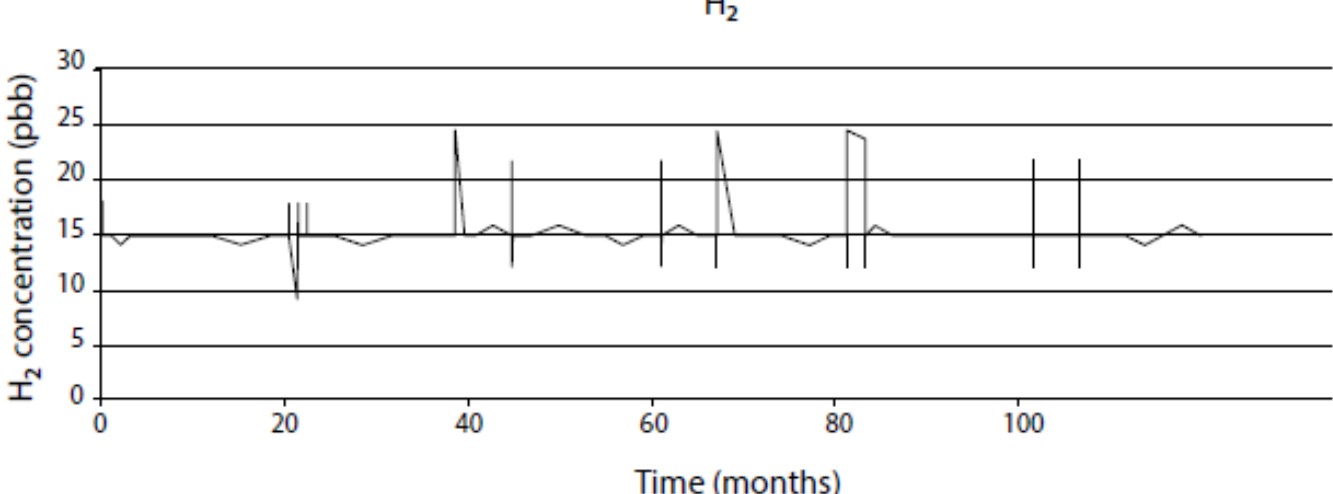

**Figure 47.** CEP with respect to hydrogen in the coolant at the HAZ adjacent to the H3 weld in the outer core shroud surface of the Chinshan BWR in Taiwan [82].

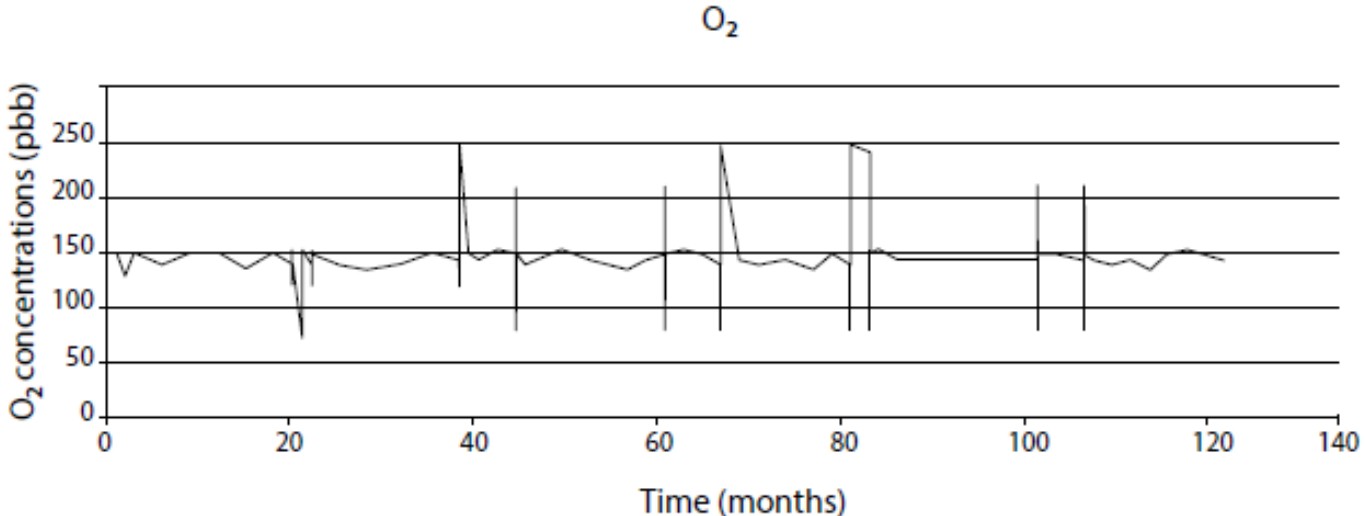

**Figure 48.** CEP with respect to oxygen in the coolant at the HAZ adjacent to the H3 weld in the outer core shroud surface of the Chinshan BWR in Taiwan [82].

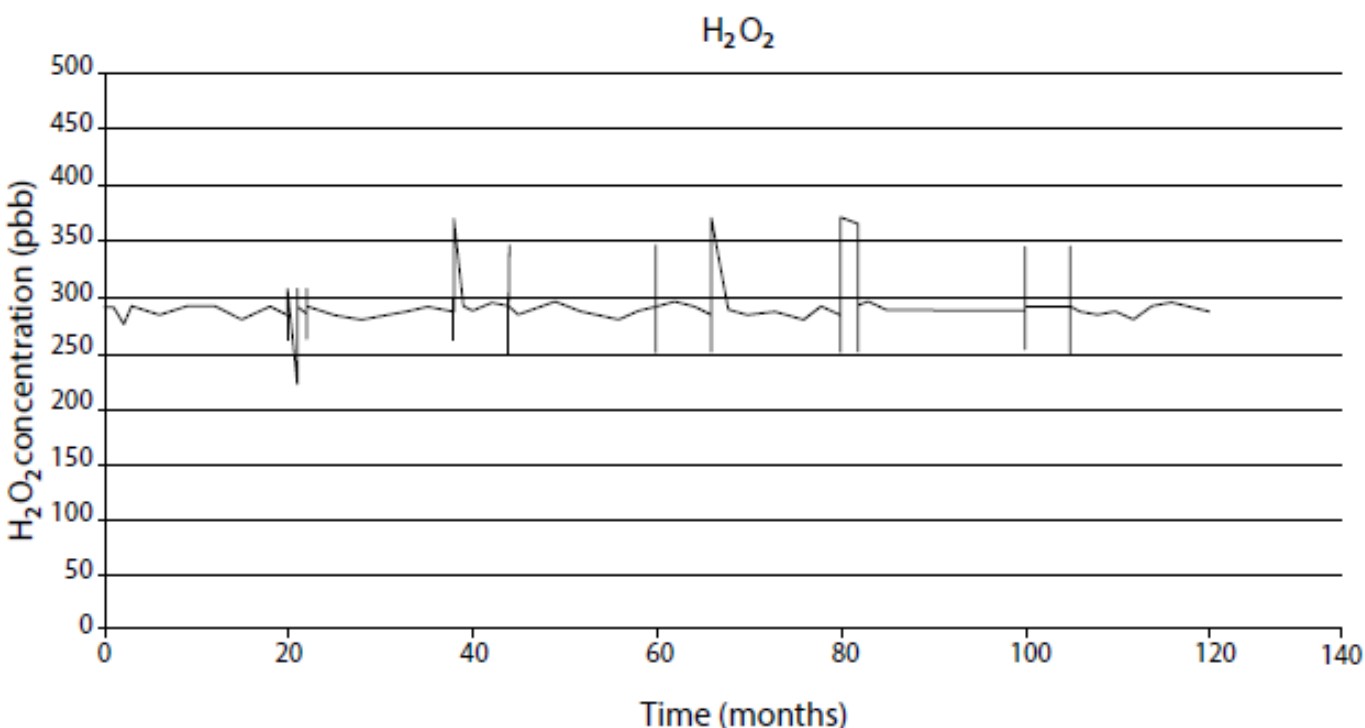

**Figure 49.** CEP with respect to hydrogen peroxide in the coolant at the HAZ adjacent to the H3 weld in the outer core shroud surface of the Chinshan BWR in Taiwan [82].

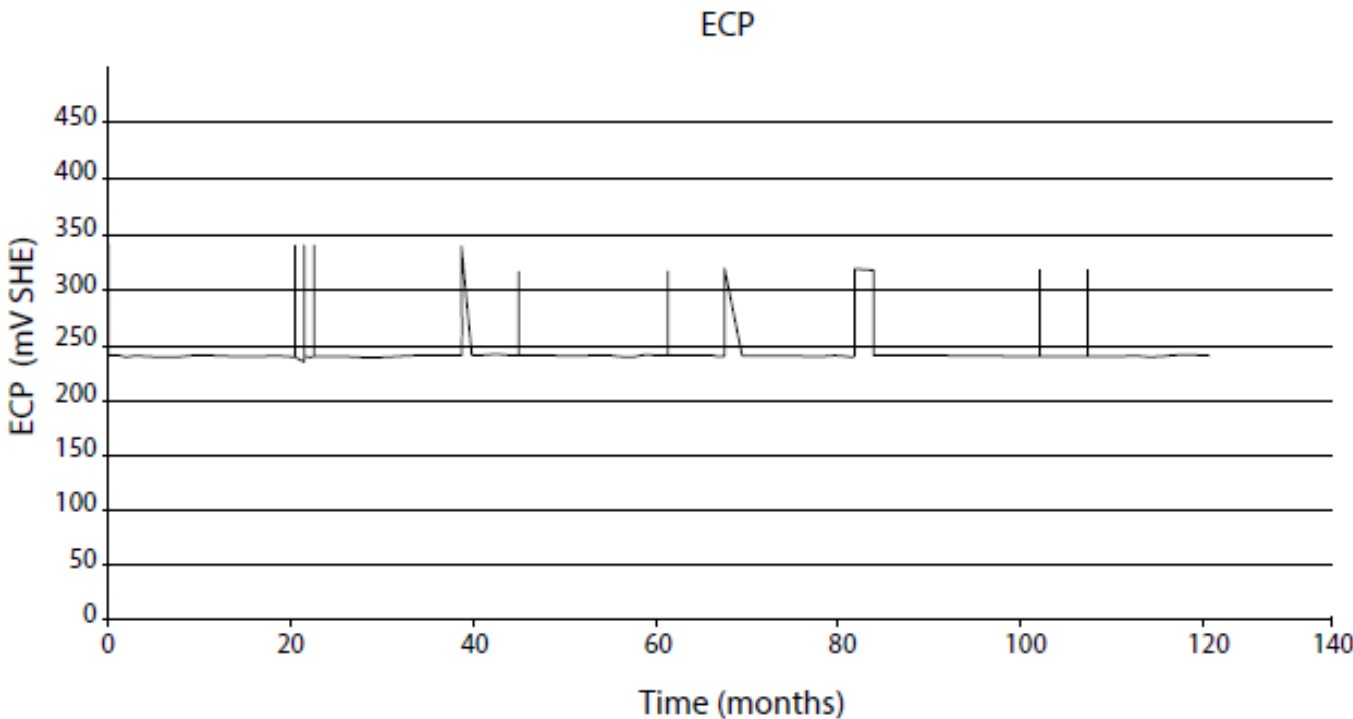

**Figure 50.** CEP of the Chinshan BWR with respect to ECP at the HAZ adjacent to the H3 weld in the outer core shroud surface of the Chinshan BWR in Taiwan [82].

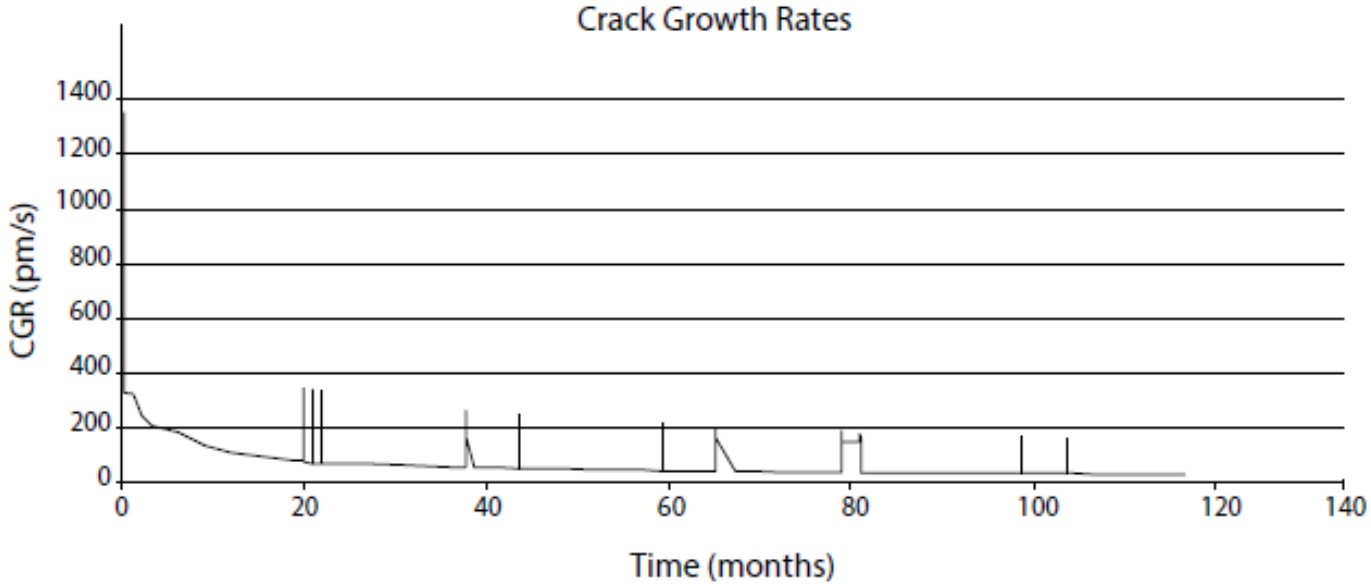

**Figure 51.** CEP of a crack in the HAZ adjacent to the H3 weld in the outer core shroud surface of the Chinshan BWR in Taiwan with respect to the CGR [82].

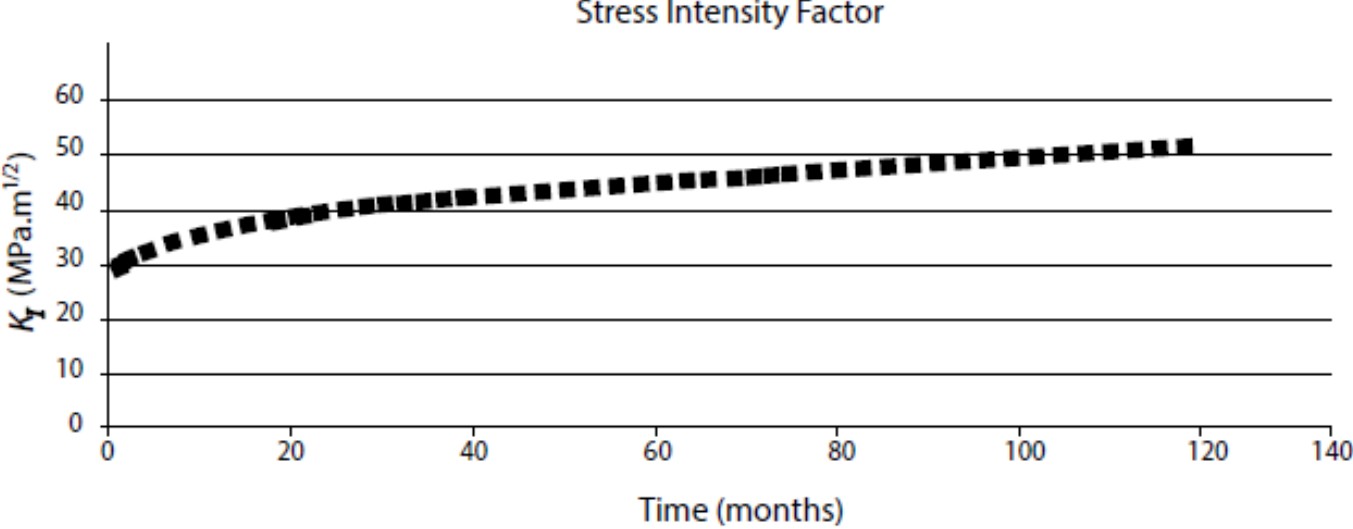

**Figure 52.** CEP of the Chinshan BWR with respect to stress intensity factor of a crack in the HAZ adjacent to the H3 weld in the outer core shroud surface of the Chinshan BWR in Taiwan [82].

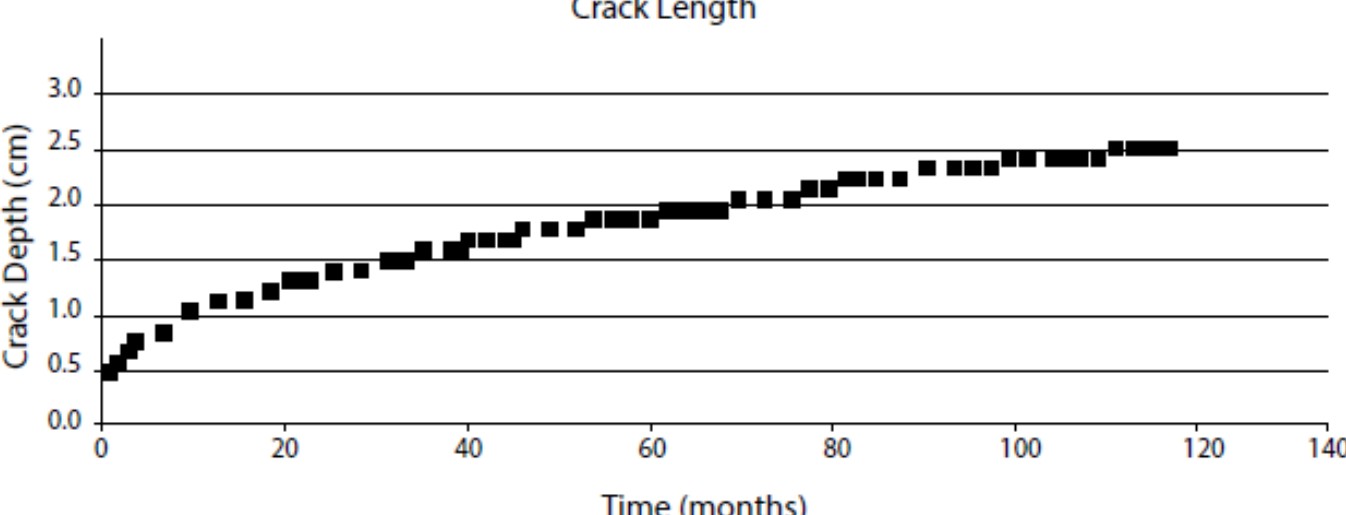

**Figure 53.** CEP of the Chinshan BWR with respect to the integrated damage (crack depth) of a crack in the HAZ adjacent to the H3 weld in the outer core shroud surface of the Chinshan BWR in Taiwan [82].

The reader will note that the spikes in various quantities, such as feed water hydrogen; coolant [$O_2$], [$H_2$], and [$H_2O_2$]; ECP; and CGR, coincide with outages of the reactor and will appreciate that these outages make a significant contribution to the damage incurred (see also Figure 29). The temporal increase in CGR during the outages is due to excursions in temperature ECP, conductivity and flow velocity during cool-down and subsequent heat-up, some of which might be controlled to mitigate the damage. For example, the implementation of HWC during outages might significantly mute the excursion in ECP, which is important because of the exponential relationship between the CGR and ECP. As noted above, the control of coolant conductivity by full-flow deionization of the coolant during excursions in the reactor power might be an option, although an expensive one.

It is necessary to assess the accuracy of the prediction for an operating BWR, in this case typified by the Chinshan BWR in Taiwan that is operated by the Taiwan Power Co. Data for this assessment are provided by Tang et al. [82], as shown in Figure 54. Thus, these workers give the depth of a circumferential crack in the inner surface of the core shroud

adjacent to the H3 weld as 1.45 cm (38% of the shroud thickness of 3.81 cm) at Month 10 after Outage 11. During the next refueling outage, the crack depth was determined to be 1.9 cm (Figure 54). The value predicted by ALERT is 1.8 cm, which is in excellent agreement with the measured value because some of the operating data had to be adopted from similar plants since they were not available to us at the time of our study. The CGR was reported by Tang et al. [82] as being $1.98 \times 10^{-8}$ cm/s (198 pm/s), which is in reasonable accord with the calculated value of approximately 50 pm/s (Figure 51). It is important to note that only the comparison at Month 20 has probative value as the lack of a crack initiation time forced us to adjust the calculated curve of crack length (equivalent to adjusting the crack initiation time) to coincide with the measured crack depth at Month 10 (Figure 54), so that the comparison is between the calculated and measured increase in crack length from Month 10 to Month 20.

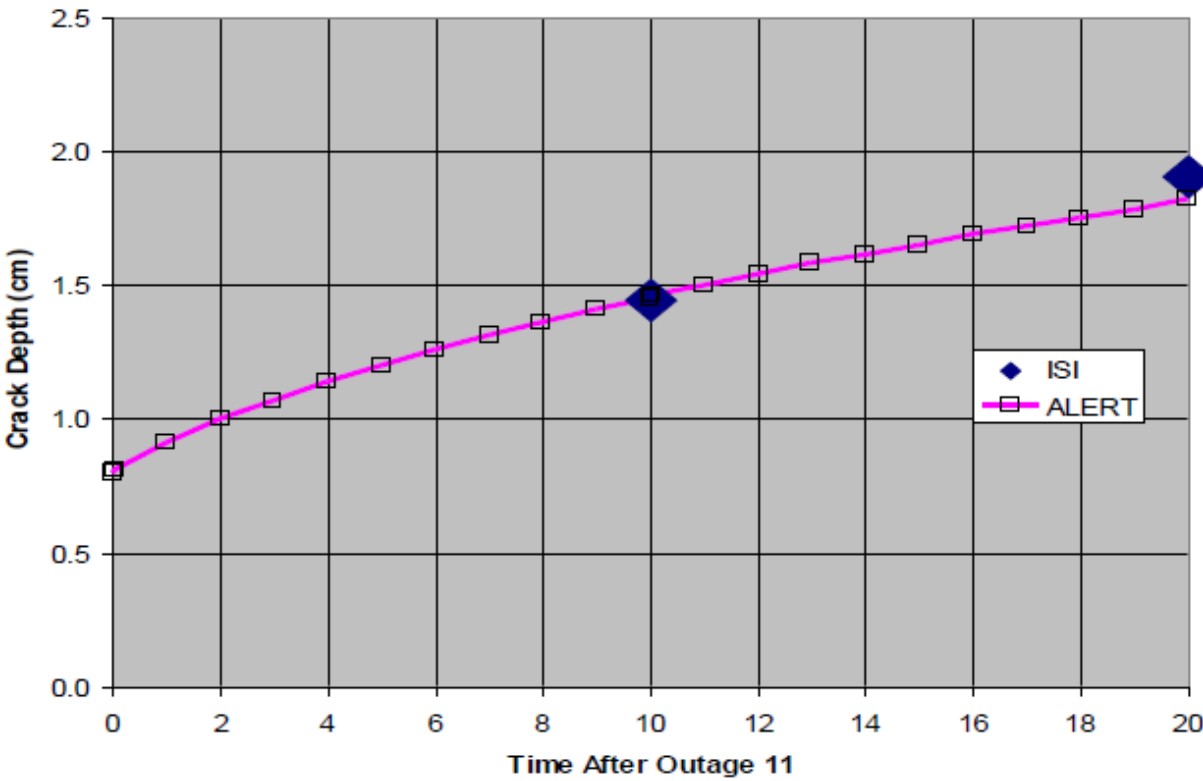

**Figure 54.** Comparison of predicted crack length in the heat-affected zone of the H3 weld in a BWR core shroud as a function of time after Outage 11. ◆ Tang et al. [82].

Finally, Tang et al. [82] describe the crack as being "circumferential", corresponding to total length of the circumferential crack in the H3 weld, which was 370.84 cm or 29% of the total circumference. This represents a highly semi-elliptical crack of the type predicted in Figures 38 and 39, for example. Thus, not only are these codes able to accurately predict the crack depth perpendicular to the surface, but they also yield realistic surface crack geometries.

## 8. Summary and Conclusions

"Science" is portrayed as the evolution of empiricism (observation) to determinism, where the future can be predicted from the past upon the basis of the natural laws which are the condensation of all previous scientific knowledge. However, this oft-expressed view must be tempered with the caveat that the system evolutionary path must be defined in terms of those independent variables that have a significant impact on the dependent variable. This is commonly the most unpredictable problem in predicting future behav-

ior. Nevertheless, predictions are made by using deterministic models, which are the calculational arms of theories. The salient issues in this process are summarized as follows.

- All theories and models are inherently incorrect because they are figments of the modeler's imagination as conceived via imperfect senses and intellect, so that they can never describe "reality". However, they are nudged toward that ideal goal by the "scientific method" of cyclical prediction and evaluation. Because of the inherent defects, models ultimately fail (i.e., are "falsified") and must be replaced by a new model that addresses the shortcomings of the old model.
- The models and theories themselves must be based upon empirical observations and must be employ postulates that are consistent with those observation and upon assumptions that, while not necessarily being demonstratably true, are reasonable expectations of current knowledge.
- In the "scientific method", the model must not be evaluated against the same data and postulates that were used in formulating the theory and calibrating the model;
- The theory itself should be "global", in that it accounts for all known observations about the system. "Local" theories are discouraged because they are based upon incomplete information, often being only based upon observations made by a single researcher;
- Importantly, all deterministic models must possess a theoretical basis but not all theories need to calculate;
- All deterministic models must contain a feedback loop that facilitates the enactment of the "scientific method", in which the model is continually tested against new observations. If discrepancies are observed, the model is modified within the bounds of observation and the prediction is repeated;
- A deterministic model generally comprises constitutive equations that describe the operation of the model and constraints, the latter commonly being the natural conservation laws. The number of constitutive equations and the number of constraints must be at least equal to the number of unknown parameters in the model;
- If no amount of change within the bounds of observation can resolve the problem, the model and the theory must be discarded ("i.e., the model is "falsified").
- It is important to note that no amount of successful prediction can "prove" a model, and its underlying theory to be correct, but only one instance of disagreement is necessary to prove the model and theory incorrect.

Model building is illustrated by describing the development of the Coupled Environment Fracture Model (CEFM) that was developed by the author and his colleagues for predicting crack growth rate due to stress corrosion cracking (SCC) in structural alloys in the primary coolant circuits of water-cooled nuclear power reactors, particularly Boiling Water Reactors. The theoretical basis of the CEFM is the Differential Aeration Hypothesis (DAH), that has stood the test of time in describing localized corrosion processes such as SCC, pitting corrosion and crevice corrosion. The model is deterministic because the predictions are constrained by the five relevant natural laws: the Law of Definite Proportions (Proust), Lavoisier's Law of the Conservation of Mass, the Law of Multiple Proportions (Dalton), the Law of the Conservation of Charge and the Law of Mass–Charge Equivalency (Faraday's Law). The first three are the fundamental laws of chemistry, while the latter two are fundamental laws of physics and electrochemistry, respectively. The CEFM, after the calibration on two CGR data at different temperatures, is found to accurately predict CGR as a function of a wide range of independent variables, including temperature, stress intensity factor, ECP, flow velocity and solution conductivity. In addition, the model successfully predicts the coupling current (CC) and the electrochemical crack length (ECL) and emphasizes the need to specify the ECL in addition to the mechanical crack length (MCL) in describing SCC, predicts the development of increasingly aggressive conditions (low pH, high $[Cl^-]$) at the crack tip with increasing CGR, and successfully predicts the micro-fracture frequency (MFF) and dimension (MFD). Furthermore, as the result of training an artificial neural network (ANN) on a large body of experimental data, the CEFM successfully accounts

for the mechanical/metallurgical/electrochemical character of IGSCC in sensitized Type 304 SS, with the fracture process being primarily electrochemical in nature, augmented by mechanics and metallurgy. The CEFM also successfully accounts for the development of oblate, semi-elliptical surface cracks that are found to develop in components in the primary coolant circuits of water-cooled nuclear power reactors.

**Funding:** This research received no external funding.

**Data Availability Statement:** Data sharing is not applicable to this article.

**Conflicts of Interest:** The authors declare no conflict of interest.

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
