# Peer review of "The Role of Determinism in the Prediction of Corrosion Damage"

_cmd, doi:10.3390/cmd4020013_

Round 1

Reviewer 1 Report

The article well reviewed the Coupled Environment Fracture Model (CEFM) which was well proved in structural alloys in the primary coolant circuits of water-cooled nuclear power reactors, particularly Boiling Water Reactors, furthermore, ANN model was used for IGSCC. However, it is confused the title is about “corrosion damage” which is seemingly not related with applied stress. Besides that, it is not clear what parameters were used to assess corrosion damage. Some minor questions are as follows:

  1. Typesetting of papers should be improved, for example, the figure and figure caption were separated in pages 7 and 8. Fig.44 is unclear, please change it.
  2. What’s the temperature when the corrosion potential was tested?
  3. The format of references is not uniform, for example, 64, 67, 69, 70, 76, etc.
  4. Please clearly point out what parameters were used to assess corrosion damage. How to present the effect of mechanical damage?

Author Response

I wish to thank the Reviewer for the time that he/she invested in reviewing this lengthy paper.  My responses to the specific questions and comments are as follows:

  1. Any typesetting issues will be resolved in the final production of the paper.
  2. I am not sure which test the Reviewer is referring to but it is generally 250 oC when measured by us and 288 oC when measured by others (e.g., by Andersen) as indicated in the captions of the figures.
  3. The format issues with the references have been resolved.
  4. Corrosion damage is assessed via the calculated corrosion rate that in turn is a function of all of the independent variables (IVs, including, ECP, T, conductivity, KI,, etc).  The IVs define the corrosion evolutionary path and the damage (the crack length) is determined by integrating the crack growth rate along that path to the desired future state.  This is described in the paper.

Reviewer 2 Report

This paper laid out very nicely and thoroughly one good example for the process of developing a deterministic model related to corrosion damage. Please consider comments as below for the revision of the paper.

  • Line 52-53: Please add reference for the cost of corrosion.
  • Line 76-84: Author only mentioned the Gumbel distribution, but Type 3 distribution (Weibull) also has been widely used in the analysis of failure due to stress corrosion cracking. Although two parameters determine the overall shape, the parameter can be a function of various independent variables. For example, Arrhenius form, which has Physical background, can be combined into the parameter to address the temperature effect. Proportional hazard model is also possible to address the effects of covariates although this model doesn’t have strong physical background. Overall author seems to classify statistical modeling approach as empiricism. Considering the crack initiation time, statistical approach might be better than deterministic approach because of inherent nature of the crack initiation. Does author believe that deterministic approach should be considered first whenever possible according to whatever corrosion damage?
  • Line 92: “Phyics” must be typo.
  • Line 469: In the 4th column of Table 1, clarify the numbers for Conductivity: “1.7 1116.”
  • Line 492-496: The PWR primary water conditions in actual plants, especially dissolved hydrogen, pH, and conductivity, are assumed to be pretty stable. That might be the reason why electrochemistry was “ignored” in many experimental studies for PWR materials. They didn’t have to consider them as variables.
  • Line 591: Typo in legend: “csthodes”
  • Line 697: In Table 3, R-O exponent must be specified.
  • Line 872: “at the crack, tip,” => “crack tip,”
  • Line 1150: “This is hue to” => “due to”
  • Line 1168: “MC” => “MCL”
  • Line 1177: In Fig. 25, MCL is identical to W, which is wrong. Specify the MCL correctly.
  • Line 1555: Figure 43 is identical to Fig. 17 or different?
  • Line 1610: The resolution of Fig. 44 is pretty low.
  • Line 1622: Figure 41 means Figure 45?
  • Line 1738: “Crack Growth Rates” located above the Figure 52 is partially covered up.
  • Line 1750: “to the to damage” => “to the damage”
  • Line 1859: Ref [6] is identical to Ref [8].
  • At the beginning the author mentioned that the definition of determinism vs. empiricism will be addressed, but it is difficult to identify which part is addressing this.
  • What is the main purpose of comparing model prediction with ANN results? ANN models belong to determinism or empiricism? If ANN models belong to determinism, what is the natural laws behind them?

Author Response

  • Line 52-53: Please add reference for the cost of corrosion. Done.
  • Line 76-84: Author only mentioned the Gumbel distribution, but Type 3 distribution (Weibull) also has been widely used in the analysis of failure due to stress corrosion cracking. Although two parameters determine the overall shape, the parameter can be a function of various independent variables. For example, Arrhenius form, which has Physical background, can be combined into the parameter to address the temperature effect. Proportional hazard model is also possible to address the effects of covariates although this model doesn’t have strong physical background. Overall author seems to classify statistical modeling approach as empiricism. Considering the crack initiation time, statistical approach might be better than deterministic approach because of inherent nature of the crack initiation. Does author believe that deterministic approach should be considered first whenever possible according to whatever corrosion damage? Yes, if insufficient data are available to devise an empirical model, such as an ANN. For example, in the case of SCC, as discussed in this paper, many hundreds if not thousands of CGR data versus whatever independent variables were considered were employed in setting up the training dataset and even then, the resulting matrix was very sparse. This is because many studies did not measure the values of important independent variables that were not the subject of the study. To establish a viable database for an empirical model can be a very expensive undertaking.  With regard to the statistical nature of physical phenomena, such as crack initiation, fundamentally it is deterministic because the processes must obey the laws of nature.  Then fact that the result appears to be distributed simply reflects real (in model parameters) or artefactual distributions (e.g., measurement error) in contributing parameters.  Thus, the Point Defect Model (PDM) predicts, upon the basis of the number of passivity breakdown sites on a surface being normally distributed with respect to the cation vacancy diffusivity, that the breakdown voltage is “near-normally” distributed but the induction time is a “left acute” distribution.  These distributions are observed experimentally.  Thus, for the initiation of cracks at pits, these mechanistically-based distributions, modified by distributions in the pit growth rate, carry over the crack initiation time.  I did not include this important subject in the paper as doing it justice would have required an additional thirty pages.  The Reviewer is correct in that the parameters contained in the Gumbel distribution is correct by adopting the mechanistically based Arrhenius equation but that simply introduces two additional unknowns; the activation energy and the value of the parameter at the reference temperature.  Unless these can be calculated from fundamental theory (e.g., DFT), it appears to me that to stop at a statistical representation appears to be somewhat pointless.  Since this paper is intended to outline a previous application of one statistical method, and is not intended to be a treatise on statistical methods as applied to corrosion science in general, and recognizing the limited space available, I respectfully decline to expand the paper further to accommodate a discussion of statistical methods 
  • Line 92: “Phyics” must be typo. Corrected.
  • Line 469: In the 4th column of Table 1, clarify the numbers for Conductivity: “1.7 1116.” Corrected.
  • Line 492-496: The PWR primary water conditions in actual plants, especially dissolved hydrogen, pH, and conductivity, are assumed to be pretty stable. That might be the reason why electrochemistry was “ignored” in many experimental studies for PWR materials. They didn’t have to consider them as variables. No, for example many hydrogen levels have been explored over the years and the pH was known to change with [B]/[Li] so at least those were understood to be variables and they were controlled during the fuel cycle. Ater looking into this issue very carefully, I concluded that he problem was the lack of any electrochemistry in mechanical engineering and nuclear engineering curricula in Universities.
  • Line 591: Typo in legend: “csthodes” Corrected.
  • Line 697: In Table 3, R-O exponent must be specified. No relevant value available.
  • Line 872: “at the crack, tip,” => “crack tip,” Corrected.
  • Line 1150: “This is hue to” => “due to” Corrected.
  • Line 1168: “MC” => “MCL” Corrected.
  • Line 1177: In Fig. 25, MCL is identical to W, which is wrong. Specify the MCL correctly. Corrected.
  • Line 1555: Figure 43 is identical to Fig. 17 or different? Figures were inadvertently mixed up. Now corrected.
  • Line 1610: The resolution of Fig. 44 is pretty low. I have redrawn the figure.
  • Line 1622: Figure 41 means Figure 45? Corrected.
  • Line 1738: “Crack Growth Rates” located above the Figure 52 is partially covered up. Corrected.
  • Line 1750: “to the to damage” => “to the damage” Corrected.
  • Line 1859: Ref [6] is identical to Ref [8]. Corrected.
  • At the beginning the author mentioned that the definition of determinism vs. empiricism will be addressed, but it is difficult to identify which part is addressing this. Much of the Introduction is devoted to a discussion of “determinism” vs “Empiricism” and further discussion occurs throughout the paper.
  • What is the main purpose of comparing model prediction with ANN results? ANN models belong to determinism or empiricism? If ANN models belong to determinism, what is the natural laws behind them? As stated, an ANN has no mechanistic or model basis so it is a purely empirical construct. A paragraph has been added to the bottom of p.10 to explain this and to compare an ANN to a deterministic model.